# H3K4me3 regulates RNA polymerase II promoter-proximal pause-release

Hua Wang[1,2], Zheng Fan[3,4,5], Pavel V. Shliaha[6], Matthew Miele[6], Ronald C. Hendrickson[6], Xuejun Jiang[1] & Kristian Helin[1,2,3,4,5] ✉

Trimethylation of histone H3 lysine 4 (H3K4me3) is associated with transcriptional start sites and has been proposed to regulate transcription initiation[1,2]. However, redundant functions of the H3K4 SET1/COMPASS methyltransferase complexes complicate the elucidation of the specific role of H3K4me3 in transcriptional regulation[3,4]. Here, using mouse embryonic stem cells as a model system, we show that acute ablation of shared subunits of the SET1/COMPASS complexes leads to a complete loss of all H3K4 methylation. Turnover of H3K4me3 occurs more rapidly than that of H3K4me1 and H3K4me2 and is dependent on KDM5 demethylases. Notably, acute loss of H3K4me3 does not have detectable effects on transcriptional initiation but leads to a widespread decrease in transcriptional output, an increase in RNA polymerase II (RNAPII) pausing and slower elongation. We show that H3K4me3 is required for the recruitment of the integrator complex subunit 11 (INTS11), which is essential for the eviction of paused RNAPII and transcriptional elongation. Thus, our study demonstrates a distinct role for H3K4me3 in transcriptional pause-release and elongation rather than transcriptional initiation.

Histone modifications are closely linked to transcription regulation and are involved in processes that determine cell fate, development and disease[5,6]. Methylation of H3 lysine 4 (H3K4) is one of the most studied modifications owing to its association with gene expression and cancer[2,7]. H3K4 methylation is deposited by SET1/COMPASS complexes that contain different lysine methyltransferases and several essential subunits, including the six catalytic subunits SETD1A, SETD1B and MLL1–4. H3K4me3 is enriched at transcription start sites (TSSs) and is believed to promote transcription through the recruitment of PHD-domain-containing proteins involved in transcription initiation, such as TATA-box-binding protein associated factor 3 (TAF3)[1,8]. Moreover, H3K4me3 has been reported to counteract DNA methylation[9] and repressive histone modifications such as H3K9me3 and H3K27me3[10,11]. Furthermore, the broad H3K4me3 domains, prominently found in preimplantation embryos[10] and somatic cells[12], have been proposed to ensure transcriptional consistency of essential genes to maintain cell identity.

Previous studies have addressed the roles of various components of the SET1/COMPASS complexes, such as CFP1[13], WDR5[14], DPY30[15], MLL1/2[16], MLL3/4[17] and SETD1A/B[18], in mammalian cells by inhibiting their expression or deleting their respective genes. However, observations of the effects on gene expression in these studies have not produced uniform results. These inconsistencies may in part be due to redundant functions of the components of the SET1/COMPASS complexes. Moreover, inadequate temporal resolution of previous knockdown or knockout studies has rendered it difficult to establish a direct role for H3K4 methylation in gene expression[19]. In this study, we determined the transcriptional effects of acute loss of H3K4me3 by targeted degradation of core components of the SET1/COMPASS complexes. Our data show that H3K4me3 has a key role in regulating RNAPII pause-release; however, notably, we did not detect a role for H3K4me3 in regulating transcription initiation.

## Models to study H3K4 methylation

To study the role of H3K4me3 in transcription, we developed model systems for the acute depletion of either DPY30 or RBBP5, two core components that have been reported to be integral and shared components of all the SET1/COMPASS family of H3K4 methyltransferase complexes, in mouse embryonic stem (mES) cell lines (Fig. 1a and Extended Data Fig. 1a,b). To do this, we generated isogenic mES cells expressing DPY30 fused to the miniAID degron tag[20] (DPY30–mAID) or RBBP5 fused to the FKBP12(F36V) degron tag[21] (RBBP5–FKBP) (Extended Data Fig. 1c). Auxin treatment led to undetectable levels of DPY30 within 1 h, and treatment with auxin for 24 h caused a substantial decrease in mono-, di- and trimethylation of H3K4 (Fig. 1b and Extended Data Fig. 1d). Similarly, treatment of RBBP5–FKBP cells with dTAG-13 also caused acute loss of H3K4 methylation after the depletion of RBBP5 (Fig. 1c). Thus, these results show that targeting DPY30 or RBBP5 of the SET1/COMPASS complex leads to a rapid loss of H3K4 methylation.

Notably, H3K4me3 levels were strongly decreased within 2 h of DPY30- and RBBP5-induced degradation, whereas a substantial decrease in H3K4me1 and H3K4me2 levels was achieved only after 24 h (Fig. 1b,c). The binding of DPY30 and RBBP5 was completely lost genome-wide, as determined by chromatin immunoprecipitation with sequencing (ChIP–seq) analysis after treatment for 2 h with auxin or dTAG-13, respectively.

[1]Cell Biology Program, Memorial Sloan Kettering Cancer Center, New York, NY, USA. [2]Center for Epigenetics Research, Memorial Sloan Kettering Cancer Center, New York, NY, USA. [3]The Institute of Cancer Research, London, United Kingdom. [4]Biotech Research and Innovation Centre (BRIC), University of Copenhagen, Copenhagen, Denmark. [5]The Novo Nordisk Foundation Center for Stem Cell Biology (Danstem), University of Copenhagen, Copenhagen, Denmark. [6]Microchemistry and Proteomics Core Facility, Memorial Sloan Kettering Cancer Center, New York, NY, USA. ✉e-mail: kristian.helin@icr.ac.uk

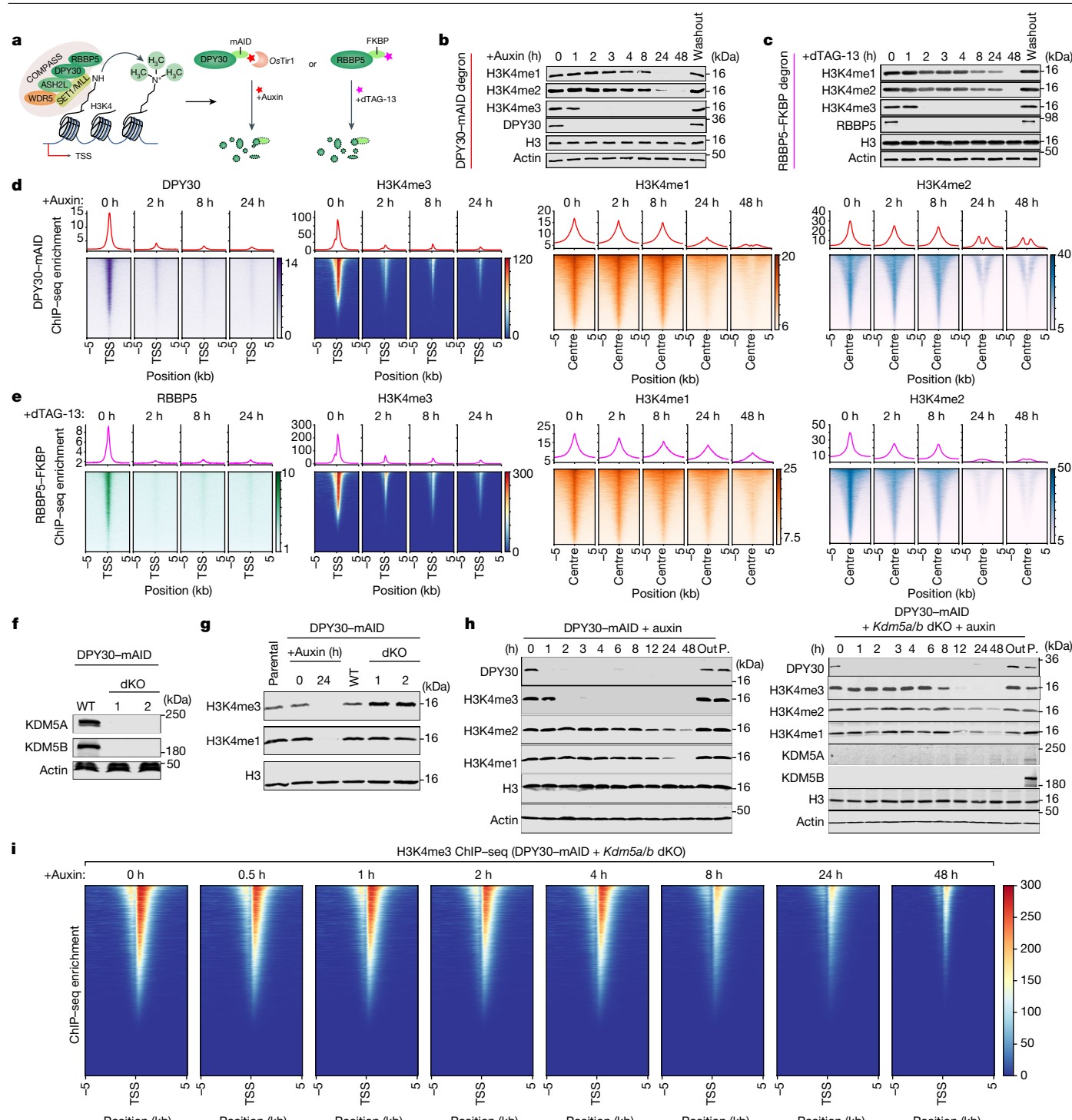

**Fig. 1 | Acute depletion of SET1/COMPASS core subunits reveals rapid turnover of H3K4me3. a**, Schematic of the degron systems for the targeted degradation of DPY30 and RBBP5. **b**,**c**, Immunoblot analysis of DPY30, RBBP5 and H3K4me1–3 levels at the indicated times after treatment with 500 nM auxin (**b**) or 500 nM dTAG-13 (**c**). Washout, degron ligand was washed out for 48 h. **d**,**e**, ChIP–seq heat maps and profiles were generated from control and auxin-treated DPY30–mAID cells (**d**) and dTAG-13-treated RBBP5–FKBP cells (**e**). For DPY30, RBBP5 and H3K4me3 ChIP–seq, the signal was plotted over the TSSs (TSS ± 5 kb) of protein-coding genes. For H3K4me1 and H3K4me2 ChIP–seq, the signal was plotted over their centre peaks (peak centre ± 5 kb), which are called from steady-state mES cells. Sites were sorted by the ChIP–seq signals at 0 h.

**f**, Immunoblot analysis of KDM5A and KDM5B in DPY30−mAID cells and two independently isolated dKO cell lines. β-Actin was used as the loading control. **g**, Immunoblot analysis of H3K4me3 and H3K4me1 levels in DPY30−mAID, control and *Kdm5a/b*-dKO cells. Histone H3 was used as the loading control. **h**, Immunoblot analysis of DPY30, H3K4me1–3, KDM5A and KDM5B at the indicated times after auxin treatment. Out, degron ligand was washed out for 48 h; P, parental cells. **i**, H3K4me3 ChIP–seq heat maps in DPY30−mAID *Kdm5a/b*-dKO cells. The signal was plotted over the TSSs (TSS ± 5 kb) of protein-coding genes. Rows are sorted by decreasing ChIP–seq occupancy in the auxin 0 h cells.

Moreover, all three methylated forms of H3K4 were strongly decreased genome-wide after treatment for 24 h with auxin or dTAG-13 (Fig. 1d,e and Extended Data Fig. 1e–h). Consistently, H3K4me3 exhibited a global decrease on average across all TSS regions genome-wide within 2 h of auxin/dTAG-13 treatment. These effects on H3K4 methylation were further validated by analysis using ChIP with quantitative PCR (ChIP–qPCR) (Extended Data Fig. 1i–l). The loss of DPY30 and RBBP5 prevented long-term subcloning of the degron cells, whereas their acute degradation led to slower proliferation (Extended Data Fig. 2a–c).

## Rapid turnover of H3K4me3 by KDM5

The rapid loss of H3K4me3 suggests that H3K4me3 levels are dynamically regulated by histone turnover and/or active demethylation. To investigate whether the rapid turnover of H3K4me3 was due to the activities of the KDM5 H3K4me3/me2 demethylases[22], we deleted *Kdm5a* and *Kdm5b* (*Kdm5a/b* double knockout (dKO)) in the DPY30–mAID mES cell line (DPY30–mAID *Kdm5a/b* dKO) (Fig. 1f). The deletion of the two genes did not lead to significant changes in cell proliferation and expression of pluripotent genes in the two independent isolated clones (Extended Data Fig. 2d,e). The dKO cells led to a global increase in H3K4me3 levels at steady state (Fig. 1g), consistent with previously reported data[22,23]. In contrast to in DPY30–mAID cells, in which global H3K4me3 was lost within 2 h, global H3K4me3 persisted for 8 h in dKO cells. By contrast, the turnover patterns of H3K4me1 and H3K4me2 were largely unaffected in the dKO cells (Fig. 1h). Furthermore, the *Kdm5a/b*-dKO cells showed a delayed reduction in H3K4me3 genome-wide relative to DPY30–mAID cells (Fig. 1e,i). Moreover, we found that the binding of the catalytic and non-catalytic components of the SET/COMPASS complexes was not significantly decreased after degradation of DPY30 (Extended Data Fig. 2f). These data demonstrate that the rapid turnover of H3K4me3 is dependent on KDM5A and KDM5B in mES cells.

## H3K4me3 is required for transcription

To determine the primary effects on transcription after rapid H3K4me3 removal, we measured the synthesis of newly transcribed RNAs using thiol(SH)-linked alkylation for the metabolic sequencing of RNA (SLAM-seq)[24] in the degron cell lines (Extended Data Fig. 3a–d). Short-term treatment with auxin and dTAG-13 (2 h and 8 h) led to a significant reduction in mRNA synthesis (Fig. 2a,b). At these time points, H3K4me3 was lost, whereas there was a minimal effect on H3K4me2 and H3K4me1 levels (Fig. 1b–e). The number of downregulated genes ($P < 0.05$ and $\log_2$-transformed fold change $< -1$), as well as their magnitude of decreased expression, increased over time in both degron systems (Fig. 2a,b). Notably, the earliest downregulated genes were more likely to be regulated by a CGI-rich promoter, to show higher levels and broader peaks of H3K4me3 than genes with unaltered expression, and to be involved in processes such as ribosome biogenesis, translation and cell cycle progression (Extended Data Fig. 3e–h). Taken together, as only H3K4me3 (and not H3K4me2/me1) was affected at early timepoints after acute loss of DPY30 or RBBP5, these data suggest that H3K4me3 is required for transcription.

If H3K4me3 were the determining factor regulating transcription, deletion of *Kdm5a* and *Kdm5b* should also lead to a significant delay in gene expression changes in response to DPY30 loss. To test this, we performed SLAM-seq analysis of auxin-treated dKO cells (Fig. 2c). Indeed, we observed a significant delay in the decrease in mRNA synthesis in the dKO cells compared with in the DPY30–mAID cells (Fig. 2d,e). Specifically, only 41 and 186 genes were downregulated in the dKO cells 2 h and 8 h after auxin treatment, respectively, whereas 379 and 1,115 downregulated genes were found at the same timepoints in DPY30–mAID cells after auxin treatment, respectively (Fig. 2f). Thus, these results suggest that the reduction in H3K4me3 levels contributes to the observed decrease in transcription.

## Intact PIC formation after loss of H3K4me3

As H3K4me3 has been reported to recruit proteins to enhance transcription initiation[1,25], we examined whether the loss of H3K4me3 could lead to changes in the expression of proteins in the RNAPII pre-initiation complex (PIC). As demonstrated in Fig. 3a, acute loss of H3K4me3 did not affect the global protein levels of multiple subunits of the PIC, such as CDK7 (TFIIH) and TBP (TFIID). Furthermore, reported H3K4me3-binding proteins also showed comparable protein levels and genome-wide binding patterns in control and auxin/dTAG-13-treated cells (Fig. 3a and Extended Data Fig. 4a). To address whether H3K4me3 loss would lead to changes in RNAPII PIC formation, we affinity-purified proteins that associate with RNAPII and determined their identities using mass spectrometry (MS) analysis of samples in which DPY30 was depleted or not (Extended Data Fig. 4b,c). However, loss of DPY30 did not lead to significant changes in the expression of RNAPII-associated proteins or in their interaction with RNAPII (Extended Data Fig. 4d and Supplementary Table 3). Thus, the loss of H3K4me3 does not lead to detectable changes in the formation of the RNAPII PIC or the association with PIC subunits to TSSs.

## H3K4me3 regulates RNAPII occupancy

To determine the relationship between H3K4me3 and RNAPII occupancy genome-wide, we performed a ChIP–seq analysis of total RNAPII in the degron cells. Consistent with the lack of detectable effects on RNAPII PIC formation, we did not observe decreased RNAPII enrichments at promoter regions after auxin or dTAG-13 treatment. By contrast, we found an increase in RNAPII occupancy at promoter regions (Extended Data Fig. 4e), which indicated that acute loss of H3K4me3 promoted RNAPII pausing. This was confirmed by calculating the RNAPII pausing index[26] in both DPY30–mAID and RBBP5–FKBP cells (Fig. 3b,c and Extended Data Fig. 4f,g). Consistent with the increased pausing, we observed an accumulation of the RNAPII CTD Ser5 phosphorylation (Ser 5p) and negative elongation factor A (NELFA) at promoter-proximal regions and a marked reduction in RNAPII CTD Ser2 phosphorylation (Ser 2p) throughout gene bodies (Extended Data Fig. 4h,i). These data suggest that H3K4me3 is involved in the regulation of RNAPII pausing.

To gain further support that the observed effect on RNAPII pause-release was caused by loss of H3K4me3, we determined the effect of degrading DPY30 in *Kdm5a/b*-dKO cells. We confirmed that KDM5A and KDM5B were lost from TSSs in the dKO cells using ChIP–seq, and that the loss of the KDM5s led to a corresponding increase in H3K4me3 levels under steady-state conditions (Fig. 3d). Moreover, we found that the increase in H3K4me3 corresponds to a concomitant decrease in steady-state RNAPII pausing in dKO cells (Fig. 3e and Extended Data Fig. 5a). We also observed a significant delay in the onset of RNAPII pausing in the dKO cells compared with in DPY30–mAID cells expressing KDM5A/B (Fig. 3e and Extended Data Fig. 5a–c). Moreover, we performed nascent transcription with mammalian native elongating transcript sequencing (mNET–seq)[27] at single-nucleotide resolution in the DPY30–mAID and dKO lines. Consistently, the mNET–seq data showed that loss of KDM5A/B led to a significant delay in RNAPII pausing compared with in KDM5A/B-expressing cells (Fig. 3f). Taken together, these results show a role for H3K4me3 in regulating RNAPII pause-release.

## H3K4me3 regulates RNAPII half-life

An increase in the occupancy of promoter-proximal RNAPII can be attributed to increased RNAPII initiation, blockage of RNAPII entering the gene body or both. To determine the stability and half-life of RNAPII at promoter-proximal regions, we inhibited transcription initiation with triptolide[28–30] and combined it with mNET–seq (Fig. 3g). The single-nucleotide resolution of the mNET–seq data enabled us to specifically measure the dynamics of RNAPII-engaged RNA at

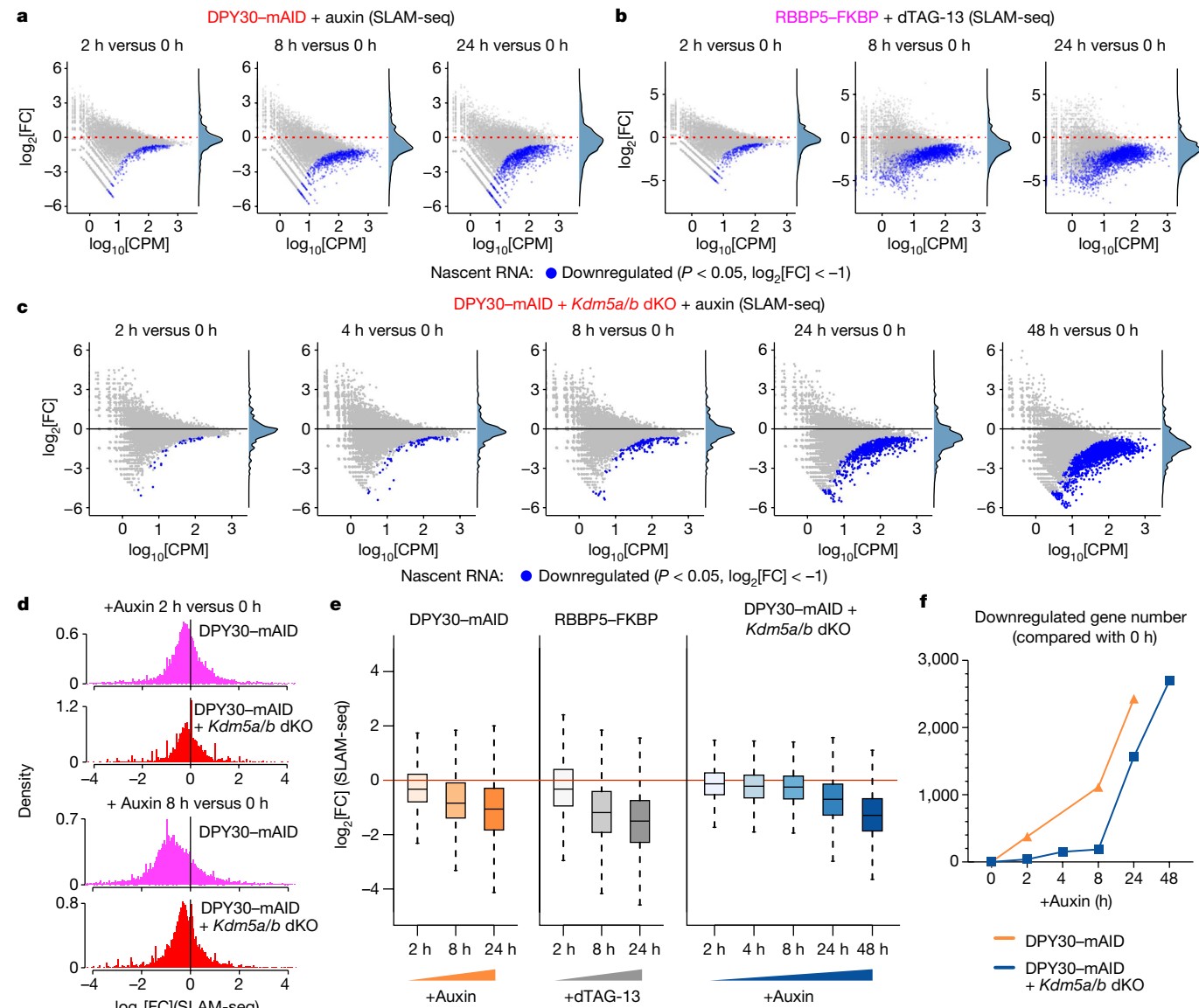

**Fig. 2 | H3K4me3 is required for nascent transcription. a**, *MA* plots depicting changes in nascent transcription (SLAM-seq) at the indicated times after auxin treatment in DPY30–mAID cells. $n = 3$ biological replicates. CPM, counts per million mapped reads; FC, fold change. Adjusted *P* values were calculated using Wald tests in DESeq2. **b**, *MA* plots depicting changes in nascent transcription (SLAM-seq) at the indicated times after dTAG-13 treatment in RBBP5–FKBP cells. $n = 3$ biological replicates. Adjusted *P* values were calculated using Wald tests in DESeq2. **c**, *MA* plots depicting changes in nascent transcription (SLAM-seq) at the indicated times after auxin treatment in DPY30–mAID *Kdm5a/b*-dKO cells. $n = 2$ biological replicates. Adjusted *P* values were calculated using Wald tests in DESeq2. **d**, The log$_2$-transformed fold change in nascent gene expression in the depicted cell lines on the basis of the data shown in **a** and **c**. **e**, The nascent transcriptional changes (log$_2$-transformed) for genes in indicated samples across timepoints with DPY30 or RBBP5 degradation kinetics. The box plots indicate the median (centre line), the third and first quartiles (box limits) and $1.5 \times$ interquartile range (IQR) above and below the box (whiskers). $n = 3$ (DPY30–mAID or RBBP5–FKBP degron cells) and $n = 2$ (dKO cells) biological replicates. **f**, Comparison of the number of downregulated genes after auxin treatment for the indicated cell lines, based on the data presented in **a** and **c**.

promoter-proximal regions. We fitted the RNAPII time-course measurements to an exponential decay model[28] and calculated the half-life of promoter-paused RNAPII on all protein-coding genes (Extended Data Fig. 6a). This analysis showed that the RNAPII half-life is 5.3 min at protein-coding genes in steady-state mES cells ($n = 4{,}007$ genes) and increased to 8.96 min ($P < 2.2 \times 10^{-16}$) in auxin-treated cells (Fig. 3h and Extended Data Fig. 6b,c). This change in RNAPII half-life is similar to what has been observed after CDK9 inhibition[31]. Indeed, we found that CDK9, BRD4 and HEXIM1 occupancies were increased at the promoter regions (Extended Data Fig. 4i) in both degron cell lines. Thus, our data suggest that H3K4me3 regulates transcription by facilitating the release of paused RNAPII into productive elongation in mES cells.

## H3K4me3 regulates elongation

To examine and monitor the effects of H3K4me3 on RNA synthesis rates genome-wide from actively transcribing RNAPII, we used a modified transient transcriptome sequencing (TT$_{chem}$-seq) method[32] with spike-in controls. Loss of H3K4me3 in auxin-treated or dTAG-13-treated cells led to a reduction of approximately 50% in the transcription of protein-coding genes at the 8 h timepoint (Fig. 4a,b) but resulted in increased RNAPII-engaged RNA at the promoter regions as measured by mNET–seq (Extended Data Fig. 6d). Furthermore, we found that there were no significant changes in RNAPII occupancy at either active or inactive enhancer regions after H3K4me3 loss (Extended Data Fig. 6e,f).

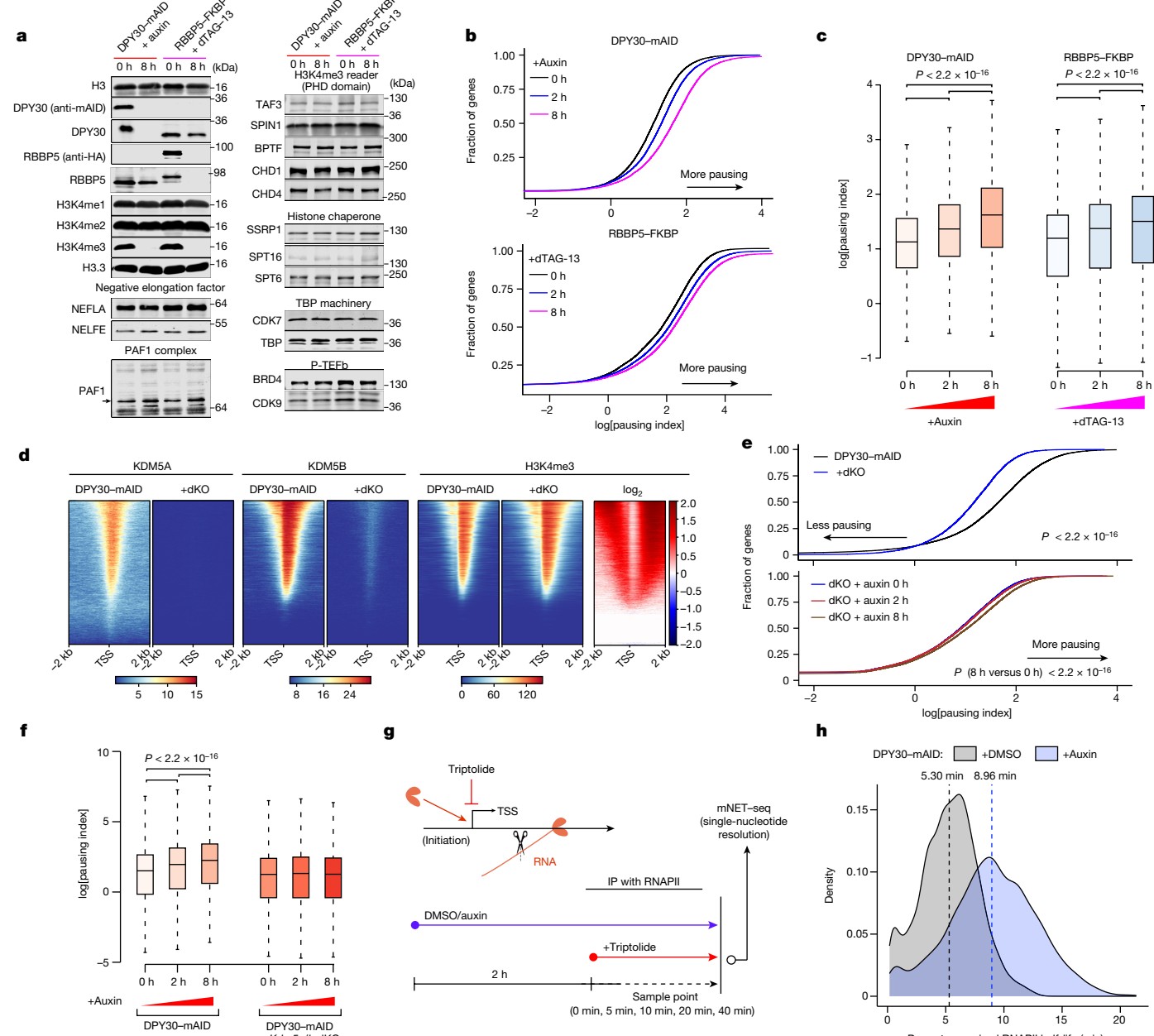

**Fig. 3 | Acute loss of H3K4me3 increases the residence time of paused RNAPII. a**, Immunoblot analysis of the indicated transcriptional core proteins and H3K4me3 readers in the indicated cell lines treated with or without auxin or dTAG-13 as shown. **b**, The RNAPII pausing index in control (0 h, black) and auxin-treated or dTAG-13-treated degron cells. Higher index values indicate a higher degree of RNAPII pausing. Cumulative index plots of the pausing index were calculated from total RNAPII ChIP–seq signals. **c**, The RNAPII pausing index was determined using ChIP–seq in the indicated samples with DPY30 or RBBP5 degradation kinetics. The box plots indicate the median (centre line), the third and first quartiles (box limits) and 1.5 × IQR above and below the box (whiskers). *P* values were calculated using two-sided Wilcoxon rank-sum tests. *n* = 12,621 genes. **d**, Comparison of the occupancy of KDM5A, KDM5B and H3K4me3 around the TSS region (TSS ± 2 kb) in DPY30–mAID and DPY30–mAID

*Kdm5a/b*-dKO cells. **e**, The RNAPII pausing index in DPY30–mAID (black), DPY30–mAID *Kdm5a/b*-dKO (blue) and auxin-treated cells. Higher index values indicate a higher degree of RNAPII pausing on promoter region of genes. *P* values were calculated using two-sided Wilcoxon tests. **f**, The RNAPII pausing index determined using mNET–seq in DPY30–mAID, DPY30–mAID *Kdm5a/b*-dKO and auxin-treated cells. The box plots indicate the median (centre line), the third and first quartiles (box limits) and 1.5 × IQR above and below the box (whiskers). *P* values were calculated using two-sided Wilcoxon tests. *n* = 10,332 genes. **g**, The experimental strategy of the mNET–seq approach to measure the promoter-proximal RNAPII half-life after treatment with triptolide. **h**, Density plot showing increased paused RNAPII half-life of *n* = 4,007 genes after acute loss of H3K4me3. The average of paused RNAPII half-life is shown as a dashed line. *n* = 2 biological replicates.

To estimate the change in elongation velocity, we used the ratio of nascent RNA synthesis measured using TT$_{chem}$-seq and RNAPII RNA occupancy measured using mNET–seq (TT$_{chem}$-seq/mNET–seq) as a proxy for elongation velocity, which is a measurement of the amount of ongoing RNA synthesis per RNAPII molecule[33,34]. This analysis showed that acute loss of H3K4me3 caused a general transcriptome-wide decrease

in elongation velocity (Fig. 4c). Consistent with this observation, the well-known epigenetic mark for transcription elongation H3K36me3 also showed a global decrease in the auxin-treated cells (Fig. 4d and Extended Data Fig. 6g).

As an orthogonal assay to investigate how H3K4me3 loss affects transcription elongation, we determined productive RNAPII elongation

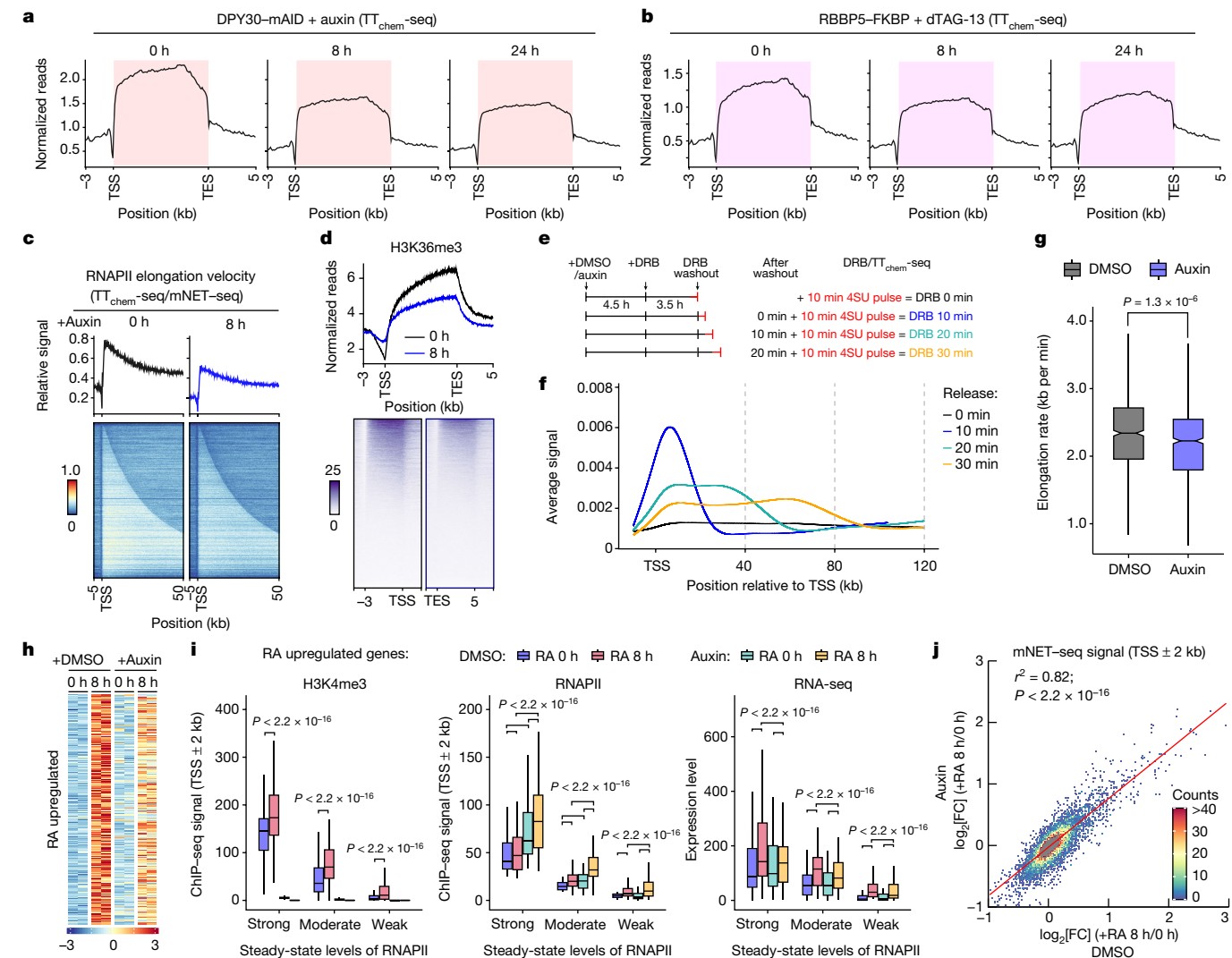

**Fig. 4 | H3K4me3 regulates transcriptional elongation. a,b**, Metagene profiles for transient transcriptome sequencing (TT$_{chem}$-seq) in control and auxin-treated (**a**) or dTAG-13-treated (**b**) cells in the indicated cell lines. TES, transcription end site. **c**, Heat maps and profiles showing changes in elongation velocities (TT$_{chem}$-seq/mNET-seq) after acute loss of H3K4me3. **d**, H3K36me3 ChIP-seq profiles and heat maps in control and auxin-treated DPY30-mAID cells. **e**, Outline of the DRB/TT$_{chem}$-seq experiment to measure RNAPII elongation rates. 4SU, 4-thiouridine. DRB 0 min, no release of DRB. **f**, DRB/TT$_{chem}$-seq metagene profiles of protein-coding genes (60–300 kb length) with non-overlapping transcriptional units (*n* = 3,566) in the depicted cells. Lines are computationally fitted splines. **g**, Box plot showing decreased RNAPII elongation rates after H3K4me3 loss. *P* values were calculated using two-sided Wilcoxon tests. *n* = 855 genes with RPM > 100. The box plots indicate the median (centre line), the third and first quartiles (box limits) and 1.5 × IQR above and below the box (whiskers). **h**, The upregulated genes response to RA treatment in auxin-treated and DMSO-treated cells (*n* = 2). Gene expression is shown as relative *Z*-scores across the samples. **i**, The changes in H3K4me3 and RNAPII ChIP-seq at TSSs (±2 kb), and RNA-sequencing analysis of RA-response genes (upregulated genes) in the indicated samples. The box plots indicate the median (centre line), the third and first quartiles (box limits) and 1.5 × IQR above and below the box (whiskers). *P* values were calculated using two-sided Wilcoxon tests. *n* = 77 genes for each group. **j**, The correlation of mNET-seq signal around the TSS region (TSS ± 2 kb) at 8 h after RA treatment with or without H3K4me3. Pearson correlation and *P* values are reported at the top. *P* values were calculated using two-sided Wilcoxon tests.

rates using TT$_{chem}$-seq in combination with the reversible CDK9 inhibitor 5,6-dichlorobenzimidazole 1-β-D-ribofuranoside (DRB) (DRB/TT$_{chem}$-seq)[32]. The progression of RNAPII was reflected by metagene coverage profiles after DRB release and clear transcription wave peaks on genes (Fig. 4e,f). The average elongation rates for the 3,655 non-overlapping protein-coding genes (60 kb to 300 kb in length) were 2.2 kb per min in control cells (Extended Data Fig. 6h), which is in good agreement with data obtained in HEK293[32] and HCT116[35] cells. We found that loss of H3K4me3 led to a significant decrease in RNAPII elongation rates (Fig. 4g and Extended Data Fig. 6i,j). Taken together, these observations suggest that H3K4me3 regulates both RNAPII pause-release and elongation in mES cells.

## Role of H3K4me3 in initiation

It has previously been suggested, mainly on the basis of in vitro experiments[1,25], that H3K4me3 functions to facilitate the formation of the PIC at TSSs. To test this hypothesis, we determined whether H3K4me3 is required for de novo activation of transcription in response to retinoic acid (RA)-induced differentiation (Extended Data Fig. 7a). A principal component analysis and correlation analysis of the entire transcriptome showed that loss of H3K4me3 did not impair the overall rewiring of gene expression induced by RA treatment (Fig. 4h and Extended Data Fig. 7b–e), suggesting that H3K4me3 is dispensable for the transcription initiation of these genes.

Considering that many RA-response genes are bivalent with both low levels of gene expression and the histone modifications H3K4me3 and H3K27me3[36] at steady state, they may already be primed for de novo transcription. To analyse this in more detail, we determined H3K4me3 and RNAPII location using ChIP–seq analysis of DPY30–mAID cells after RA treatment with or without auxin treatment. As expected, H3K4me3 was lost in auxin-treated cells and showed increased RNAPII at promoter regions (Extended Data Fig. 6f,g). By dividing the RA-induced genes into three categories on the basis of the levels of RNAPII enrichment (strong, moderate and weak) at promoter regions in steady-state mES cells (Fig. 4i), we found increased RNAPII enrichments and gene expression for the same three category genes in the auxin-treated cells (Fig. 4i), suggesting that the RA-response genes can be initiated de novo in the absence of H3K4me3. As these steady-state data cannot rule out an effect on transcriptional initiation, high-resolution mNET–seq was performed. Correlation analysis of the mNET–seq results also showed that the loss of H3K4me3 did not impair the loading of RNAPII at the promoter-proximal region of genes induced by RA treatment (Fig. 4j). Thus, we conclude that H3K4me3 is not required for RNAPII loading and for transcriptional initiation.

## The H3K4me3-dependent RNAPII interactome

To understand the mechanism leading to increased promoter-proximal pausing in response to H3K4me3 loss, we combined CRISPR-based genome editing, APEX2-based proximity labelling[37] and quantitative MS (SILAC/MS) to obtain a high-resolution view of the molecular and spatial organization of RNAPII with or without H3K4me3 (Fig. 5a). We knocked-in APEX2–Flag-tagged *Rbp1* in both degron cell lines, which did not lead to detectable effects on the cellular functions of the protein (Extended Data Fig. 8a–e). After activation with $H_2O_2$, APEX2 oxidizes phenol derivatives (biotin-phenol), which covalently react with the nearby endogenous proteins (Extended Data Fig. 8f,g). To capture the changes on chromatin after the loss of H3K4me3, we isolated the chromatin fraction from each sample before mixing the light and heavy conditions for MS (Extended Data Fig. 8h). We identified 1,901 proteins that were in the proximity of RPB1, and KEGG and Gene Ontology analyses showed that most of these proteins are associated with RNA biogenesis processes (Extended Data Fig. 8i–l and Supplementary Table 4), highlighting the quality of the RPB1–APEX2 data.

We focused on the changes in the two early timepoints (2 h and 8 h) for which only H3K4me3 (and not H3K4me2 and H3K4me1) was lost in response to auxin treatment (Extended Data Fig. 8m). Importantly, a large proportion of the proteins that showed differential interactions with RPB1 after DPY30 degradation were shared at the two times (Extended Data Fig. 8n). Consistent with the experimental design, DPY30 was the top downregulated protein in all of the auxin-treated samples (Extended Data Fig. 8o,p). Moreover, functional enrichment analysis of the common 228 downregulated proteins indicated a strong enrichment for mRNA-processing components of the core RNAPII machinery (Extended Data Fig. 8q). In addition to H3K4me3 loss in the cells, terms such as 'positive regulation of histone H3K4 methylation', 'chromatin binding' and 'histone binding' were more prominently decreased with chromatin RNAPII (Extended Data Fig. 8q).

## H3K4me3-dependent recruitment of INTS11

To identify proteins that could potentially explain the requirement for H3K4me3 in promoter-proximal pause-release, we overlapped the common downregulated proteins with proteins that have previously been shown to associate with H3K4me3 using cross-linked ChIP–MS[38]. Notably, three proteins were in common between the two protein groups: DPY30 itself, PAF1 and INTS11 (Extended Data Fig. 9a). These three proteins were also found to be preferentially enriched in the H3K4me3 ChIP–MS data, when compared with ChIP–MS data from

other heterochromatin or non-promoter specific histone modifications (Extended Data Fig. 9b). PAF1[39,40] and INTS11[41,42] have both been reported to have important roles in RNAPII pause-release and also in transcriptional elongation. We chose to focus on INTS11—an endonuclease subunit of the Integrator complex—as it showed a more substantial decrease in RPB1 association than PAF1 in response to H3K4me3 depletion (Extended Data Fig. 8p).

The reduction in RNAPII-bound INTS11 after acute loss of H3K4me3 was further confirmed by western blot analysis of both DPY30–mAID and RBBP5–FKBP degron cells (Fig. 5b and Extended Data Fig. 9c). Recent studies indicate that integrator primarily attenuates gene expression in *Drosophila*[43] and in human HeLa[44] cells, while other studies have shown a critical role of INTS11 in gene activation and elongation[45,46]. To improve our understanding of the role of INTS11 in transcription regulation, we generated INTS11–FKBP-knockin cells along with a double haemagglutinin (HA) epitope tag in the DPY30–mAID RPB1–APEX2 degron mouse ES cell line (Extended Data Fig. 9d). Western blot analysis detected similar levels of INTS11 in the constructed cell line and in non-tagged cells, and the tagging of INTS11 did not lead to detectable effects on the expression of pluripotency genes and cell proliferation (Extended Data Fig. 9e–g). The addition of dTAG-13 to these cells led to rapid degradation of the INTS11–FKBP–HA fusion protein within 2 h (Fig. 5c). Consistent with our findings for the role of H3K4me3 in regulating RNAPII pause-release, acute loss of INTS11 led to increased RNAPII pausing at promoter-proximal regions of genes (Fig. 5d–f) and to inhibition of cell proliferation (Extended Data Fig. 9g). Moreover, the enrichment of INTS11 at TSSs was significantly reduced in response to DPY30 degradation (Extended Data Fig. 9h). These data suggest H3K4me3-dependent recruitment of INTS11, which is required to enable transcriptional pause-release.

We further studied the effects of INTS11 loss by performing $TT_{chem}$-seq analysis of INTS11 degron cells (Extended Data Fig. 9i). Loss of INTS11 led to a significant increase in the RNA synthesis and RNAPII of non-coding short transcripts, such as upstream antisense RNAs (uaRNAs/PROMPTs) (Extended Data Fig. 9j,k). By contrast, a global decrease in the expression of protein-coding genes (Fig. 5g), especially at the pausing sites between TSS and the first (+1) nucleosome (Fig. 5h) was observed after INTS11 degradation. Analysis of elongation velocity at protein-coding genes showed broadly decreased productive elongation after INTS11 loss (Fig. 5i). Notably, the degradation of INTS11 led to the loss of only part of the integrator complex from chromatin (Fig. 5j). Taken together, these findings further support a functional link between H3K4me3 and INTS11.

To further investigate this link, we measured ongoing transcription using SLAM-seq analysis of the INTS11–FKBP degron cell line (Extended Data Fig. 10a). Notably, short-term treatment with dTAG-13 led to a significant decrease in nascent mRNA synthesis (Fig. 5k and Extended Data Fig. 10b). We also observed a strong correlation between the enrichments of H3K4me3 and INTS11 at TSS promoter-proximal regions (Fig. 5l) and that there was a significant enrichment of H3K4me3 at INTS11-bound genes, but not at INTS11-unbound genes (Fig. 5m and Extended Data Fig. 10c,d). By plotting the fold changes in nascent RNA expression determined in DPY30–mAID cells in response to auxin treatment and in INTS11–FKBP cells treated with dTAG-13, we also showed that most of the changes were correlated between the two degron systems (Fig. 5n).

Finally, we investigated whether similar correlations exist in other cell types and analysed published data from different human cell lines—HeLa[42], THP1[47] and HL60[48] (Supplementary Fig. 1a). ChIP–seq experiments in HeLa, THP1 and HL60 cells identified that ~60%, ~52% and ~61% of INTS11 peaks overlapped with promoters, respectively (Supplementary Fig. 1b). Notably, compared with the H3K4me1-occupied regions, most of the H3K4me3-occupied regions were marked with significant INTS11 and RNAPII levels (Supplementary Fig. 2a). In support of the results obtained in mES cells, INTS11 occupancy was heavily enriched towards the promoter-proximal region of genes in all of

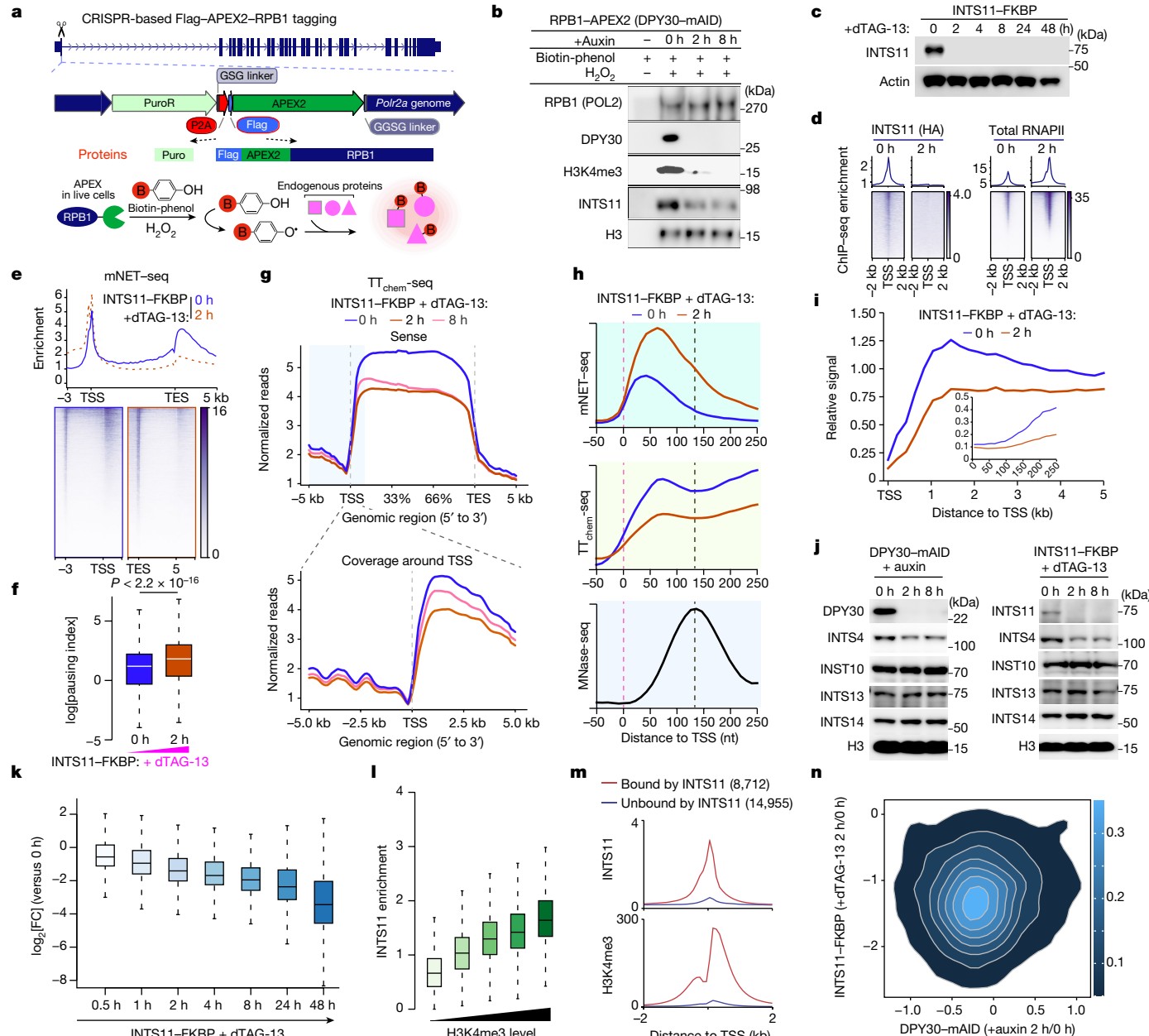

**Fig. 5 | INTS11 regulates pause-release and transcription dependent on H3K4me3. a**, The strategy for CRISPR-based Flag–APEX2–RPB1 (RNAPII–APEX2) tagging. **b**, Validation of H3K4me3-dependent INTS11 chromatin interaction in DPY30–mAID cells. **c**, Western blot analysis of HA-tagged INTS11 and actin in INTS11–FKBP cells. **d**, HA-tagged INTS11 and total RNAPII ChIP-seq profiles and heat maps in INTS11–FKBP cells. **e**, mNET–seq profiles and heat maps in INTS11–FKBP degron cells. **f**, The RNAPII pausing index was determined using mNET–seq in INTS11–FKBP cells. The box plots indicate the median (centre line), the third and first quartiles (box limits) and 1.5 × IQR above and below the box (whiskers). *P* values were calculated using two-sided Wilcoxon tests. *n* = 10,332 genes. **g**, Metagene transcriptional profiles were acquired using TT~chem~–seq in INTS11–FKBP degron cells. **h**, Metagene analyses of mNET–seq and TT~chem~–seq signals at single-nucleotide resolution acquired in INTS11–FKBP cells. MNase-seq, micrococcal nuclease sequencing. **i**, Analysis of

elongation velocities (TT~chem~-seq/mNET–seq) after acute loss of INTS11. **j**, Immunoblot analysis of integrator subunits in the indicated cell lines treated with or without auxin or dTAG-13 as shown. **k**, Box plot comparing the log₂-transformed fold change in SLAM-seq at the indicated timepoints. The box plots indicate the median (centre line), the third and first quartiles (box limits) and 1.5 × IQR above and below the box (whiskers). *n* = 13,776 genes. **l**, The INTS11 signal at TSSs (± 2 kb) in the indicated groups. The box plots indicate the median (centre line), the third and first quartiles (box limits) and 1.5 × IQR above and below the box (whiskers). *n* = 4,700 genes. **m**, The average distribution of INTS11 and H3K4me3 ChIP-seq signals at INTS11-bound genes (*n* = 8,712) versus INTS11-unbound genes (*n* = 14,955) in mES cells. **n**, 2D kernel density plot showing the relationship between SLAM-seq changes in INTS11–FKBP and DPY30–mAID degron cells. The colour bar reflects the intensity.

these human cell lines (Supplementary Fig. 2b). Moreover, heat maps also demonstrated a significant colocalization of the peak binding sites for INTS11 and RNAPII with the peak sites of H3K4me3 enrichment on active promoters (Supplementary Fig. 2b). Thus, these published ChIP–seq results from different cells validated a correlation

between INTS11-binding sites and sites of H3K4me3 enrichment across the genome. Taken together, these results show that H3K4me3 regulates promotor-proximal pausing through a mechanism involving the recruitment of the INTS11, which is essential for the eviction of paused RNAPII and transcriptional elongation.

## Discussion

H3K4me3, which is catalysed by the SET1/COMPASS complexes, is tightly associated with TSSs and is widely believed to be involved in regulating transcription initiation[1,2,7,25]. By generating and using model systems to study the direct role of H3K4me3 in regulating transcription, we have shown that loss of H3K4me3 leads to an increase in RNAPII at promoters. Moreover, we have shown, through multiple orthogonal assays, that H3K4me3 regulates RNAPII pausing and potentially also has a role in elongation. A connection between H3K4me3 and elongation has previously been suggested on the basis of results in cancer cells[12] and plants[49]. Notably, we did not detect a role for H3K4me3 in transcriptional initiation and activation of de novo transcribed genes in response to RA-induced differentiation. Furthermore, we observed a fast turnover of H3K4me3 that is dependent on KDM5 demethylase activity, and we propose that this rapid turnover of H3K4me3 is important for the dynamic regulation of transcriptional output. Our experimental approach does not enable us to delineate specific roles for H3K4me1 and H3K4me2 in transcriptional regulation because all H3K4 methylation states are influenced at later time points. Notably, recent results suggest that SET1A/B can also regulate CpG-island-associated gene expression independently of H3K4me3 and its methyltransferase activity[50], indicating potential distinct roles of components of the SET1/COMPASS complexes.

Paused RNAPII has been proposed to be coupled with transcription and mRNA-processing events by helping to maintain the accessibility of promoters for regulatory factors that activate transcription by recruiting P-TEFb and other factors, and promoting the release of paused RNAPII into gene bodies[30,33,39]. Whether histone modifications have a role in regulating the RNAPII pause-release step is not well understood. Our results show that H3K4me3 coupled with INTS11 is involved in regulating RNAPII promoter-proximal pausing. We propose a model in which H3K4me3 regulates transcriptional cycles by facilitating the recruitment of INTS11 to protein-coding genes (Extended Data Fig. 11). INTS11 can subsequently, through its RNA endonuclease activity, mediate productive transcriptional elongation by evicting paused RNAPII. This model is supported by recent studies showing that mammalian INTS11 facilitates RNAPII pause-release and gene expression[45,51]. Most of our conclusions are also supported by another recent study; however, in this study the authors observed an upregulation of protein-coding short transcripts after INTS11 downregulation[52]. Although we do not know the reason for the different observations, it may reflect the efficiency of the two different degron systems used in the studies.

Further work will be needed to determine the detailed molecular mechanism that leads to the recruitment of INTS11 and its function in elongation. In summary, our study demonstrates that H3K4me3 is indeed an important post-translational modification that regulates promoter-proximal pause-release and facilitates gene expression.

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

# Methods

## mES cell culture and differentiation

E14 ES cells (129/Ola background) were maintained on 0.2% gelatin-coated plates in Glasgow minimum essential medium (GMEM, Sigma-Aldrich, G5154) containing 15% fetal bovine serum (Gibco, 26140079), supplemented with 1× penicillin–streptomycin (Thermo Fisher Scientific, 15140122), 2 mM GlutaMax (Thermo Fisher Scientific, 35050061), 50 μM β-mercaptoethanol (Thermo Fisher Scientific, 21985023), 0.1 mM non-essential amino acids (Thermo Fisher Scientific, 11140050), 1 mM sodium pyruvate (Thermo Fisher Scientific, 11360070) and Leukaemia inhibitory factor (LIF, 1,000 U ml$^{-1}$, Millipore), referred to as serum mES cell medium. Cells were passaged every 2 days by aspirating the medium, dissociating the cells with trypsin/EDTA solution (TE) briefly at room temperature before rinsing and dissociation in mES cell medium by pipetting. Cells were pelleted by centrifugation at 300$g$ for 5 min. mES cell transfection was performed using Lipofectamine 3000 (Thermo Fisher Scientific, L3000-001) according to the manufacturer's instructions. Cell counts were performed using the Countess II automated cell counter (Thermo Fisher Scientific, AMQAX1000) using 10 μl of cell suspension and 10 μl of Gibco Trypan Blue Solution (Gibco, 15250061) according to manufacturer's instructions. For all-*trans*-retinoic acid (RA, Sigma-Aldrich, R2625-50MG) treatment, cells were induced with 1 μM RA and without LIF for the indicated time. During differentiation, RA medium was changed every 24 h. For SILAC experiments, cells were cultured in SILAC DMEM (Thermo Fisher Scientific, A33822) containing 15% dialysed FBS (Thermo Fisher Scientific, 26400044), to which either $^{13}C_6{}^{15}N_2$ L-lysine-2HCl (Thermo Fisher Scientific, 88209) and $^{13}C_6{}^{15}N_4$ L-arginine-HCl (Thermo Fisher Scientific, 89990) (heavy), or L-lysine (Sigma-Aldrich, L8662) and L-arginine (Sigma-Aldrich, L8094) containing only light isotopes (light) was added. All cell lines were subjected to STR authentification through ATCC and were tested for mycoplasma contamination.

## Generation of mES cell knockin cell lines

For the generation of the auxin-inducible degradation system for DPY30 sgRNA targeting the stop-codon region were cloned into eSpCas9(1.1)-T2A-eGFP. Left and right homology arms, as well as the mAID-T2A-BFP middle part were ligated into a modified pUC19 vector backbone (a gift from S. Pollard) using the In-Fusion cloning kit (Takara, 638910). mES cells were co-transfected with sgRNA- and donor-vector using Lipofectamine 3000 and sorted 48 h later for GFP/BFP-double-positive cells. Homozygous clones were then transfected with pPB-hygro-OsTIR1-P2A-mCherry and pBase plasmids and selected with 100 μg ml$^{-1}$ hygromycin B (Thermo Fisher Scientific, 10687010). For the generation of the endogenous dTAG-inducible degradation system for RBBP5, sgRNA targeting the stop codon region were co-transfected into the cells and contained the following elements: left and right homology arms, as well as FKBP12(F36V), 2× HA tags, P2A and a neomycin-resistance gene. The transfected cells were selected with 100 μg ml$^{-1}$ Geneticin selective antibiotic (G418 Sulfate) (Thermo Fisher Scientific, 10131027), single-cell sorted to obtain clonal cell lines and screened for correct biallelic integration. All homozygous insertions and knock-ins were confirmed by Sanger sequencing and western blotting. A list of the oligos and the sequences of the sgRNAs is provided in Supplementary Table 1.

## Generation of mES knockout cell lines

Cells were transfected with eSpCas9(1.1)-T2A-eGFP or eSpCas9(1.1)-T2A-mCherry vectors containing a sgRNA targeting the specific genomic locus, respectively, and cells were single-cell sorted 48 h after transfection. To generate the *Kdm5a/b*-dKO cells in the DPY30–mAID cell line, we first generated the *Kdm5a*-KO in the DPY30–mAID line (*Kdm5a*$^{-/-}$) by Cas9 (sgRNAs are listed in Supplementary Table 1), then we generated the *Kdm5b* knockout in the *Kdm5a*$^{-/-}$ line (at passage three

of the *Kdm5a*$^{-/-}$ line). Subsequently, two clones with both *Kdm5a* and *Kdm5b* knockout (referred to as DPY30–mAID *Kdm5a/b*-dKO in this study) were picked up for the downstream analysis. All homozygous insertions and knockouts were confirmed by Sanger sequencing and western blotting. A list of the sgRNAs is provided in Supplementary Table 1. Further characterization of the dKO cell lines showed that they did not have detectable changes in proliferation and expressed normal levels of pluripotent genes; however, as expected, they showed an increase in H3K4me3 levels as compared with the wild-type cells.

## Western blotting

Cells were lysed in RIPA buffer with Halt protease inhibitor (Thermo Fisher Scientific, 78429). Proteins that were separated by SDS–PAGE using acrylamide gels (BioRad gel system) were transferred onto nitrocellulose membranes (LI-COR, 926-31092). The membranes were blocked in 5% skimmed milk (Sigma-Aldrich) in PBS-T (0.1% Tween-20 in PBS) and incubated with the primary antibody of interest (Supplementary Table 2). As secondary antibodies, either IRDye 800CW goat anti-rabbit IgG (925-32211, LI-COR Bioscience, 1:15,000) or IRDye 800CW goat anti-mouse IgG (925-32210, LI-COR Bioscience, 1:15,000) was used. Proteins were imaged using Image Studio Lite (Odyssey CLx imager, Li-COR Biosciences). Immunoblotting source data are provided in Supplementary Fig. 4.

## Flow cytometry

mES cells were dissociated with trypsin/EDTA, resuspended in culture medium, centrifuged and resuspended in PBS. For intracellular flow cytometry, 0.5 ml of cold fixation buffer (BioLegend, 420801) was added and then incubated at room temperature for 10 min. Subsequently, the cells were labelled with the unconjugated rabbit DPY30 antibodies (Bethyl Laboratories, A304-296A) and subsequently with a FITC-conjugated goat anti-rabbit IgG antibody. Flow cytometry was performed using the Beckman Coulter CytoFlex system. The gating strategy is shown for the E14 sample in Supplementary Fig. 5.

## RNA extraction, cDNA synthesis and RT–qPCR analysis

Total RNA was extracted using the RNeasy Plus Mini Kit (Qiagen, 74134) according to the manufacturer's protocol. One microgram of total RNA was subjected to reverse transcription using Transcriptor Universal cDNA Master (Sigma-Aldrich, 5893151001). qPCR with reverse transcription (RT–qPCR) reactions were set up in triplicate using PowerUp SYBR Green Master Mix (Thermo Fisher Scientific, A25778) and primers (listed in Supplementary Table 1). Relative quantitation was performed to a housekeeping gene using a $\Delta\Delta C_t$ method, as indicated in the corresponding figure legends. Statistical analysis was performed using GraphPad Prism v.7 (GraphPad).

## 3′-RNA Quant-seq and SLAM-seq

Cells ($1 \times 10^6$ per treatment condition) were resuspended in 350 μl of buffer RLT plus, and total RNA was extracted from cell pellets using the RNeasy Plus Mini kit (Qiagen, 74134). For SLAM-seq experiments[24], cells were incubated with 100 μM 4-thiouridine (4SU; Biosynth, NT06186) for 60 min before RNA isolation. RNA (1 μg) was treated with 10 mM iodoacetamide in a 50 μl reaction volume at 50 °C for 15 min with 50 mM NaH$_2$PO$_4$ (pH 8.0), and 50% (v/v) DMSO followed by addition of 1 μl of 1 M dithiothreitol (DTT) to stop the reaction. RNA was precipitated at −80 °C for 60 min with 1 μl of GlycoBlue (Thermo Fisher Scientific, AM9515), 5 μl of 3 M sodium acetate (pH 5.2) and 180 μl of ethanol (≥99.0%). RNA was pelleted at 4 °C (12,000$g$ for 30 min), washed with 1 ml of 80% ethanol and centrifuged at 4 °C (12,000$g$ for 30 min). The RNA pellet was dried at room temperature for 10 min and resuspended in 20 μl of nuclease-free H$_2$O. RNA yield and quality were assessed using the 2200 TapeStation (Agilent). Sequencing libraries were prepared using the QuantSeq 3′-mRNA Seq Library Prep Kit FWD for Illumina (Lexogen, SKU: 015.96) from 500 ng or 100 ng of total RNA

spiked with ERCC RNA Spike-In Mix 1 (1:1,000, Thermo Fisher Scientific, 4456740). In brief, first-strand (oligo(dT)) cDNA synthesis was followed by RNA removal and second-strand synthesis by random priming. The double-stranded library was bead-purified to remove reaction components before PCR amplification with i7 single-index primers for 10 cycles. Amplified libraries were again bead-purified according to the manufacturer's protocol, and the concentration was measured by Qubit assay. All of the samples were checked for fragment size distribution on the TapeStation before pooling for 75 bp or 45 bp single-end read sequencing on the Illumina NextSeq 550 platform.

## ChIP–seq

ChIP experiments were performed according to a standard protocol. In brief, ES cells were cross-linked by the addition of 1% formaldehyde (Sigma-Aldrich, 252549-1L) in the dish for 10 min at room temperature before quenching with 0.125 M glycine. The fixed cells were washed with PBS and resuspended in SDS buffer (100 mM NaCl, 50 mM Tris-HCl pH 8.0, 5 mM EDTA, 0.5% SDS, 1× protease inhibitor cocktail from Roche). The resulting nuclei were precipitated, resuspended in the immunoprecipitation buffer at 1 ml per 16 million cells (SDS buffer and Triton dilution buffer (100 mM NaCl, 100 mM Tris-HCl pH 8.0, 5 mM EDTA, 5% Triton X-100) mixed at a 2:1 ratio with the addition of 1× protease inhibitor cocktail from Roche) and processed on the Bioruptor Plus Sonicator (Diagenode) to achieve an average fragment length of 200–300 bp. Chromatin concentrations were estimated using the Nanodrop according to manufacturer's protocols. The immunoprecipitation reactions were set up in 1 ml of the immunoprecipitation buffer as indicated below and incubated overnight at 4 °C. The next day, BSA-blocked Protein G Dynabeads (Thermo Fisher Scientific, 10004D) were added to the reactions and incubated for 2 h at 4 °C. The beads were then washed three times with low-salt washing buffer (150 mM NaCl, 1% Triton X-100, 0.1% SDS, 2 mM EDTA, 20 mM Tris-HCl pH 8.0) and twice with high-salt washing buffer (500 mM NaCl, 1% Triton X-100, 0.1% SDS, 2 mM EDTA, 20 mM Tris-HCl pH 8.0). The samples were then reverse cross-linked overnight at 65 °C in the elution buffer (1% SDS, 0.1 M NaHCO$_3$) and purified using the QIAQuick PCR purification kit (Qiagen, 28506). A list of the antibodies used in this study is provided in Supplementary Table 2. Libraries for ChIP–seq were prepared using the NEBNext Ultra II DNA Library prep kit (NEB, E7645L), and AmpureXP beads (Beckman, A63881) were used for size selection. Libraries were quantified using the Qubit High Sensitivity DNA kit (Agilent, Q32854) and assessed on the TapeStation. Libraries were pooled as required, denatured and loaded onto the Illumina NextSeq 550 system with high-output kits (75 cycles). A list of all of the primers used for ChIP–qPCR is provided in Supplementary Table 1. For spiked-in ChIP–seq, 5% of the cross-linked *Drosophila* chromatin (homemade) with Spike-in Antibody (Active Motif, 61686) was added before the immunoprecipitation step according to the manufacturer's instructions.

## RNAPII IP followed by MS

To measure the difference between RNAPII interactome in the presence and absence of DPY30, $1 \times 10^8$ DPY30–mAID cells were incubated with auxin for 8 h to induce DPY30 degradation (8 h samples), while $1 \times 10^8$ DPY30–mAID cells treated with DMSO served as the control (0 h samples). Cells were collected and frozen on dry ice and kept at −80 °C until immunoprecipitation (IP). The cells were thawed at 37 °C for 30 s lysed in 1.6 ml of ice cold 50 mM EPPS pH 7.5, 150 mM NaCl, 1% Triton X-100 with cOmplete, EDTA-free Protease Inhibitor Cocktail (1 tablet per 20 ml of lysis buffer), 1:100 of Sigma-Aldrich phosphatase inhibitor 2 and 3 cocktails and 250 U μl$^{-1}$ of benzonase. Lysates were incubated on ice for 5 min to allow DNA digestion, centrifuged at 20,000$g$ for 5 min to remove insoluble material and filtered through the AcroPrep 1.0 μm glass filter plate at 2,000$g$ for 1 min. The concentration of protein was then estimated using the bicinchoninic acid assay. For each of the samples (DMSO- and auxin-treated cells) six immunoprecipitation

reactions were performed: three with the anti-RNAPII antibody (Abcam, ab817, 8WG16) and three with IgG control (Invitrogen, 02-6102). Each reaction was performed with 1 mg of lysate at 3.3 g l$^{-1}$ lysate concentration and 5 μg of antibody bound to protein G Sepharose (Sigma-Aldrich, 17-0618-02). The incubation was performed at 4 °C with shaking at 1,100 rpm for 1 h. The beads were then transferred to the OF 1100 filter plate (Orochem Technologies) and washed five times with ice-cold 50 mM EPPS pH 7.5, 150 mM NaCl using vacuum manifold. Then, 18 μl of 10 mM EPPS pH 8.5 with 20 ng μl$^{-1}$ trypsin and 1 ng μl$^{-1}$ LysC was added to the beads in each well and digestion was performed for 2 h at 37 °C at 2,000 rpm. The partial digest was then collected into a 96-well PCR plate and left overnight at room temperature to complete digestion. Then, 4 μl of 22 g l$^{-1}$ 11 plex TMTPro tags were added to each sample. The samples were then pulled and 20 μl of the combined sample was set aside, and the rest was fractionated into six fractions using the High pH Reversed-Phase Peptide Fractionation Kit, as suggested by the manufacturer. The fractions were concatenated into four fractions (the first and fifth fractions, the second and sixth and so on were mixed) and evaporated in speed vac (0.5 μl of DMSO was added to each sample to prevent complete evaporation) and resuspended in 20 μl 0.1% TFA. For data acquisition, 4.5 μl of unfractionated sample and every fraction was analysed by using the nanoAcquity 2 μm particle size, 75 mm × 500 mm easyspray column in direct injection mode. The samples were separated using the following gradient at 300 nl min$^{-1}$ of buffer A (0.1% formic acid in water) and buffer B (0.1% formic acid in acetonitrile): 0–7% in 10 min, 7–30% in 92 min, 30–60% in 18 min, the column was then washed with 95% B for 10 min at 400 nl min$^{-1}$. The column was kept at 60 °C. Eluting peptides were analysed on the Orbitrap Fusion Lumos instrument using MS3 SPS with the settings recommended by the instrument manufacturer for TMT11 plex analysis with the following modifications: (1) CID NCE for MS2 was set at 32; (2) HCD NCE for MS3 was set at 45; (3) C series exclusion was disabled as TMTPro reagent was not enabled in C-series exclusion node. The cycle time was set at 3 s and the dynamic exclusion time was set at 15 s.

## APEX2-based RNAPII proximity labelling and affinity enrichment of biotinylated proteins

CRISPR–Cas9 technology was used to target endogenous *Rpb1* at the 5′ end with a cassette encoding a Flag affinity-tag and APEX2 (Flag–APEX2), resulting in RPB1 fused at its N terminus to Flag–APEX2. DPY30–mAID OsTIR1 E14 cells were co-transfected with espCas9 plasmid and a donor plasmid containing the puromycin-resistance selection gene, P2A self-cleavage site and Flag–APEX2 flanked by homology arms corresponding to the respective target genes. The sequences of the guide RNA and homology arms for targeting *Rpb1* are provided in Supplementary Table 1. The APEX2-expressing cells were incubated with 4 mM biotin-phenol reagent (Iris Biotech, LS-3500.1000) for 2 h before the start of the labelling reaction. The cells were washed with PBS (with Ca$^{++}$ and Mg$^{++}$) and the labelling reaction was initiated by adding 1 mM H$_2$O$_2$ in PBS for 2 min at room temperature. The reaction was terminated by washing the cells three times with a quencher solution containing 10 mM sodium azide, 10 mM sodium ascorbate and 5 mM Trolox in PBS.

## Cell fractionation for SILAC chromatin MS

Chromatin fractions were prepared as described previously[53] with some modifications. In brief, cells were lysed by swelling and mechanical force in buffer A (10 mM ammonium bicarbonate pH 8.0, 1.5 mM MgCl$_2$, 10 mM KCl, 10 mM sodium ascorbate, 5 mM Trolox, 10 mM sodium azide, 1× protease inhibitor cocktail and 0.2% NP40). Nuclei were then collected by centrifugation and chemically lysed in buffer C (20 mM ammonium bicarbonate pH 8.0, 420 mM NaCl$_2$, 20% (v/v) glycerol, 2 mM MgCl$_2$, 0.2 mM EDTA, 0.1% NP40, 10 mM sodium ascorbate, 5 mM Trolox, 10 mM sodium azide, 1× protease inhibitor cocktail and 0.5 mM DTT). Lysates were centrifuged at 20,800$g$ for 45 min at 4 °C. The pellet contains the insoluble chromatin fraction and consists of DNA and

proteins tightly bound to chromatin. To solubilize the chromatin pellet, 750 U Benzonase (Sigma-Aldrich) was added, followed by 10 min incubation on ice and 5 min of agitation at room temperature. Clarified lysate was collected and the protein concentration was quantified using Bio-Rad Bradford's reagent. Approximately 4 mg lysates from SILAC heavy or light cells were mixed 1:1 and incubated with 50 µl Streptavidin magnetic beads (Pierce, 88817) at 4 °C on a rotating wheel overnight. The beads were washed four times with RIPA buffer.

### Sample preparation for SILAC/MS

Eluates from biotin pull-down were transferred to fresh microfuge tubes. NuPAGE sample loading buffer was added to the beads and heated at 90 °C for 5 min. A magnetic rack was used to separate the beads from the proteins. The supernatant was then run on an SDS–PAGE gel (Bis-Tris, 4–12%) enough to get the sample into the gel. Gel sections were excised, washed, reduced with DTT, alkylated with iodoacetamide and digested overnight with trypsin at 37 °C (ref. [54]). Homemade C18 StageTips were prepared as described previously[55] and preconditioned with a 50 µl wash of methanol, 50 µl wash of 70% acetonitrile/0.1% trifluoroacetic acid and two 50 µl washes of 0.1% trifluoroacetic acid at 1,000$g$. Peptides were then loaded onto StageTips and washed with 50 µl of 0.1% formic acid and were eluted with 60 µl of 70% acetonitrile/0.1% formic acid. The samples were then vacuum centrifuged using the SpeedVac and reconstituted in 0.1% formic acid for LC–MS/MS and were analysed by microcapillary LC–MS/MS using the nanoAcquity system (Waters) with a 100 µm inner-diameter × 10 cm length C18 column (1.7 µm BEH130, Waters) configured with a 180 µm × 2 cm trap column coupled to a Q-Exactive Plus mass spectrometer (Thermo Fisher Scientific). Peptides were eluted at 300 nl min$^{-1}$ using a 4 h acetonitrile gradient (0.1% formic acid). The Q-Exactive Plus mass spectrometer was operated in automatic, data-dependent MS/MS acquisition mode with one MS full scan (380–1,600 $m/z$) at 70,000 mass resolution and up to ten concurrent MS/MS scans for the ten most intense peaks selected from each survey scan. Survey scans were acquired in profile mode and MS/MS scans were acquired in centroid mode at 17,500 resolutions with an isolation window of 1.5 amu and normalized collision energy of 27; AGC was set to $1 \times 10^6$ for MS1 and $5 \times 10^4$ and 50 ms max IT for MS2; charge exclusion of unassigned, +1 and greater than 6 was enabled with dynamic exclusion of 15 s.

### TT$_{chem}$-seq and DRB/TT$_{chem}$-seq analysis

We performed TT$_{chem}$-seq as previously described[32]. In brief, cells in a 10 cm dish at 80% confluency were treated in biological duplicates at the specified time points. After the specified treatment, we supplemented the treatment medium with 1 mM 4SU and metabolically labelled the cells for 10 min. The cells were lysed in QIAzol (Qiagen, 79306) and total RNA was isolated according to the manufacturer's instructions before the addition of 100 ng of RNA spike-in mix together with QIAzol. The RNA spike-in was extracted from *Drosophila* S2 cells using 4SU, metabolically labelling the cells for 20 min. The 100 µg RNA (in a total volume of 100 µl) was fragmented by addition of 20 µl of 1 M NaOH and left on ice for 20 min. Fragmentation was stopped by addition of 160 µl of 0.5 M Tris pH 6.8 and cleaned up twice using the Micro Bio-Spin P-30 Gel Columns (BioRad, 7326223) according to the manufacturer's instructions. Biotinylation of 4SU-residues was performed in a total volume of 250 µl, containing 10 mM Tris-HCl pH 7.4, 1 mM EDTA and 5 mg of MTSEA biotin-XX linker (Biotium, BT90066) for 30 min at room temperature in the dark. RNA was then purified by phenol–chloroform extraction, denatured by 10 min incubation at 65 °C and added to 200 µl µMACS Streptavidin MicroBeads (Milentyl, 130-074-101). RNA was incubated with beads for 15 min at room temperature and beads were applied to a µColumn in the magnetic field of a µMACS magnetic separator. The beads were washed twice with pull-out wash buffer (100 mM Tris-HCl, pH 7.4, 10 mM EDTA, 1 M NaCl and 0.1% Tween-20). Biotinylated RNA was eluted twice by addition of 100 mM DTT and cleaned up using the

RNeasy MinElute kit (Qiagen, 74204) using 1,050 µl ethanol (≥99%) per 200 µl reaction after addition of 700 µl of RLT buffer to precipitate RNA of less than 200 nucleotides. A total of 200 ng of the purified 4SU-labelled RNA was then used as input for the TruSeq Stranded Total RNA kit (Illumina, 20020596) for library preparation. The libraries were amplified according to the manufacturer's instructions with modifications as previously described[32]. The library was amplified with 10 PCR cycles and quality-control checked on the TapeStation (Agilent) using the High Sensitivity DNA kit before pooling and paired-end sequencing on the NextSeq 550 (Illumina) system.

For DRB/TT$_{chem}$-seq, cells were incubated in 100 µM DRB (Sigma-Aldrich, D1916) for 3.5 h. The cells were then washed twice in PBS, and the prewarmed fresh DRB-free medium was added to restart transcription. The RNA was labelled in vivo with 1 mM 4SU for 10 min before the addition of QIAzol, which was used to stop the reaction at the desired time point.

### mNET–seq

We performed mNET–seq with minor modifications to the original protocol[27]. The Flag epitope tag was added to the N terminus of the first RNAPII subunit (RPB1) in DPY30–mAID degron cells (RPB1–APEX2 cells). Cells were seeded the day before the experiment to get 100 million cells per sample the next day. We randomly assigned flasks for each treatment and treated cells with DMSO only or auxin ligand, before extracting the chromatin-bound RNAPII. The cells were first washed with ice-cold DPBS, resuspended in 4 ml of ice-cold HLB + N buffer (10 mM Tris-HCl (pH 7.5), 10 mM NaCl, 2.5 mM MgCl$_2$, 0.5% (v/v) NP-40 and 1× proteinase inhibitor) and incubated on ice for 5 min, then cells were scraped to a 15 ml centrifugate tube. The cell suspension was then underlaid with 1 ml of HLB + NS buffer (10 mM Tris-HCl pH 7.5, 10 mM NaCl, 2.5 mM MgCl$_2$, 0.5% (v/v) NP-40, 10% (w/v) sucrose and 1× proteinase inhibitor) and centrifuged to pellet the nuclei at 400$g$ at 4 °C. The supernatant and membrane debris were then removed, and the nuclei were resuspended in 125 µl of NUN1 lysis buffer (20 mM Tris-HCl pH 8.0, 75 mM NaCl, 0.5 mM EDTA, 50% (v/v) glycerol and 1× proteinase inhibitor) to which we added 1.2 ml NUN2 buffer (20 mM HEPES-KOH pH 7.6, 300 mM NaCl, 0.2 mM EDTA, 7.5 mM MgCl$_2$, 1% (v/v) NP-40, 1 M urea and 1× proteinase inhibitor) to precipitate the chromatin, and the sample was incubated on ice for 15 min with occasional vortexing. The lysates were then centrifuged at 16,000$g$ for 10 min at 4 °C to pellet the chromatin. The chromatin pellets were then washed with 1× MNase buffer and then digested with 50 U MNase (NEB, M0247S) for 2 min at 37 °C with 1,400 rpm on a thermomixer. The digestion was stopped by adding 5 µl of 500 mM EGTA (to a final concentration of 25 mM; Thermo Fisher Scientific, 50-255-956) and transferred onto ice. The reactions were then centrifuged at 16,000$g$ for 5 min at 4 °C and the supernatant was subsequently diluted with 1 ml of NET-2 buffer (50 mM Tris-HCl pH 7.4, 150 mM NaCl, 0.05% (v/v) NP-40) per fraction and pooled per sample for the N-terminal Flag-RNAPII IP. We added 50 µl of anti-Flag M2 Affinity gel (Sigma-Aldrich, A2220) and incubated in the cold room for 1 h. This was followed by eight washes with NET-2 buffer and one wash with 500 µl of PNKT buffer containing 1× T4 polynucleotide kinase (PNK) buffer (NEB, M0201L) and 0.1% (v/v) Tween-20 (Thermo Fisher Scientific, BP337-100). The beads were incubated in 100 µl of PNK reaction mix containing 1× PNK buffer, 0.1% (v/v) Tween-20, 1 mM ATP and T4 PNK 3′ phosphatase minus (NEB, M0236L) at 37 °C for 10 min. We eluted the RNA by adding 350 µl of buffer RLT plus and 1 ng of fragmented *Drosophila* S2 mRNA spike-in to the extraction buffer. Next, short immunoprecipitated RNA fragments were size-selected (under 200 nucleotides) and purified from the eluates according to the manufacturer's protocol after the purification of miRNA from animal cells using the RNeasy Plus Mini Kit and the RNeasy MinElute Cleanup Kit (Qiagen, 74204), and eluted in 14 µl of nuclease-free water. We checked 1 µl of the eluted RNA on the TapeStation for the RNA

quality and proceeded to NGS library preparation using the NEBNext Multiplex Small RNA Library Prep kit (NEB, NC0477293). Both were performed according to the manufacturer's protocol with low input material, with 14 PCR cycles for the library amplification step. We cleaned up and concentrated the DNA using the Monarch kit (NEB) and separated the library on a 6% TBE gel and performed size-selection by excising the smear between 147 and 307 nucleotides (according to the Quick-Load pBR322 DNA-MspI Digest ladder (NEB)). These libraries were quality-checked and quantified using the TapeStation. The samples were pooled and sequenced using paired-end sequencing on the NextSeq 550 (Illumina) system.

For promoter-proximal RNAPII half-life experiments determined by mNET–seq, chromatin was isolated from cells that were pretreated with DMSO or auxin and then incubated with 1 µM triptolide for 0, 5, 10, 20 or 40 min in the presence of DMSO/auxin and analysed using mNET–seq. Chromatin was digested with micrococcal nuclease (MNase, scissor) to release RNAPII engaged RNA from insoluble chromatin, and then immunoprecipitated using anti-Flag–RNAPII antibodies.

### IP–MS data analysis

Data were analysed in Proteome Discoverer v.3.1. A database search was performed with the Sequest HT search engine using the Mouse UniProt database containing only reviewed entries and canonical isoforms (retrieved on 10 October 2019). Oxidation (M) was set as a variable modification, while TMT was set as fixed modification. A maximum of two missed cleavages were permitted. The precursor and fragment mass tolerances were 10 ppm and 0.6 Da, respectively. Peptide-spectrum matches (PSMs) were validated by percolator with a 0.01 posterior error probability threshold. Only PSMs with an isolation interference of <25% and at least 5 MS2 fragments matched to the peptide sequence selected for MS3 were considered. The quantification results of PSMs were combined into protein-level quantification using the MSstatsTMT R package[56]. Only proteins with at least three peptides were reported. To identify interactors, we performed differential abundance analysis between the IP samples and their corresponding controls (that is, 0 h IP was compared to 0 h IgG control and 8 h IP was compared to 8 h IgG control). A protein was considered to be an interactor if in one or both comparisons its levels were statistically significantly different ($Q \leq 0.05$, limma test, with $P$ values adjusted by the Storey method) and at least twice higher in IP reactions than in the corresponding IG control (Supplementary Table 3).

### SILAC/MS analyses

All SILAC/MS data were processed using the MaxQuant software (Max Planck Institute of Biochemistry; v.1.5.3.30). The default values were used for the first search tolerance and main search tolerance—20 ppm and 6 ppm, respectively. Labels were set to Arg10 and Lys8. MaxQuant was set up to search the reference mouse proteome database downloaded from UniProt on 9 January 2020. MaxQuant performed the search assuming trypsin digestion with up to two missed cleavages. Peptide, site and protein false-discovery rate (FDR) were all set to 1% with a minimum of one peptide needed for identification but two peptides needed to calculate a protein level ratio. The following modifications were used as variable modifications for identifications and included for protein quantification: oxidation of methionine, acetylation of the protein N terminus, ubiquitination of lysine, phosphorylation of serine, threonine and tyrosine residues, and carbamidomethyl on cystine. Intensity values measured in all replicates were $\log_2$-transformed (Supplementary Table 4), $P$ values were computed using Fisher's tests and corrected using Benjamini–Hochberg FDR correction. All raw MS data files have been deposited to the ProteomeXchange Consortium (and PXD039176). The Gene Ontology term enrichment analysis was performed using Enrichr online tool (https://maayanlab.cloud/Enrichr/), STRING (https://string-db.org) and clusterProfiler[57].

### ChIP–seq data analysis

The sequenced reads were demultiplexed using bcl2fastq (v.2.19.0.316), and basic quality control was performed on the resulting FASTQ files using FastQC (v.0.11.8). FASTQ reads were mapped to the GRCm38 (mm10) genome using Bowtie2 (v.2.4.1) using the standard settings. The resulting SAM files were converted to BAM files using the SAMtools (v.1.10) view command, after which the BAM files were sorted and indexed, and potential PCR duplicates were removed using the rmdup function. DeepTools (v.3.3.0) was used to generate occupancy heat maps, and the resulting normalized occupancy matrix was used as input for public R scripts to generate average profile plots and to calculate processivity indices. In brief, the BAM files were converted into BigWig files using the bamCoverage function (bamCoverage -p 8 --normalizeUsing RPGC --effectiveGenomeSize mm10 --centerReads -e --scaleFactor X --blackListFileName mm10.blacklist.bed). For comparison, quantitative ChIP–seq data using spike-in normalization were used, normalization to the *Drosophila* S2 spike-in was performed at this stage according to the manufacturer's instructions (Active Motif, 61686; https://www.activemotif.com/catalog/1091/chip-normalization). The computeMatrix function was used to quantify the occupancy of reads across the specified intervals, and the plotProfile and plotHeatmap functions were used to plot the data. Reproducibility of replicates is shown in Supplementary Fig. 3.

### Quant-seq/SLAM-seq analysis

Gene and 3' untranslated region (UTR) annotations were obtained from the UCSC table browser (https://genome.ucsc.edu/cgi-bin/hgTables, mm10 vM14 3' UTR). Adapters were trimmed from raw reads using cutadapt through the trim_galore wrapper tool with adapter overlaps set to 3 bp for trimming. For Quant-seq, concatenated fastq files were trimmed for adapter sequences, and masked for low-complexity or low-quality sequences using trim_galore, then mapped to the mm10 whole genome using HISAT v.2.2.1 with the default parameters. The number of reads mapped to the 3' UTR of genes was determined using featureCounts. Raw reads were normalized to CPM. SLAM-seq analysis was performed as previously described[58] using the SlamDunk package[59]. Trimmed reads were further processed with SlamDunk (v.0.3.4 16). The 'Slamdunk all' command was executed with the default parameters except '-rl 74 -t 8 fastq.gz -n 100 -m -mv 0.2 -o Slamdunk2', running the full analysis procedure (slamdunk all) and aligning against the mouse genome (GRCm38), filtering for variants with a variant fraction of 0.2. Unless indicated otherwise, reads were filtered for having ≥2 T>C conversions. The remaining parameters were left as defaults.

Analysis of differential gene expression was restricted to genes with ≥10 reads in at least one condition. Differential gene expression calling was performed on raw read counts with ≥2 T>C conversions using DESeq2 with the default settings, and with size factors estimated on corresponding total mRNA reads for global normalization. Downstream analysis was restricted to genes that passed all internal filters for FDR estimation by DESeq2. Plots of differential gene expression were visualized using the ggplot2 package in R with significant genes ($P$ value < 0.05, |log2FC| ≥ 1). Reproducibility of replicates is shown in Supplementary Fig. 3.

### mNET–seq data analysis

Reads were demultiplexed using bcl2fastq, then trimmed for adapter content with cutadapt (-m 10 -e 0.05 --match-read-wildcards -n 1), and mapped with STAR to the GRCm38 (mm10) genome assembly. Further data processing was performed using the R/Bioconductor environment. Coverage tracks for further analysis were restricted to the last nucleotide incorporated by the RNAPII in the aligned mNET–seq reads as described[60]. To calculate the half-life of paused RNAPII for each gene by mNET–seq, RNAPII density was calculated in a 300 bp window downstream of the TSS. RNAPII time-course measurements were fitted into

an exponential decay model using the RNAdecay R package (https://bioconductor.org/packages/release/bioc/html/RNAdecay.html). We selected genes fulfilling the current criteria: (1) detectable RNAPII levels (reads per kilobase of transcript, per million mapped reads > 1), (2) highest RNAPII density under the no triptolide (0 min) condition and (3) low variance between replicates ($\sigma < 0.05$). Genes fitting the above criteria ($n = 6,338$) were used to calculate the RNAPII half-life. Reproducibility of replicates is shown in Supplementary Fig. 3.

## $TT_{chem}$-seq data analysis

Paired-end reads were demultiplexed using bcl2fastq. $TT_{chem}$-seq raw data were processed essentially as described previously[32]. Raw reads were aligned to the mouse mm10 genome assembly using STAR. Mapped reads with a mapping quality score <10 were discarded with SAMtools. All further processing was performed using the R/Bioconductor framework. Antisense bias, sequencing depth and cross-contamination rates were calculated as described previously[32]. Reads were mapped to transcription units, which represent the union of all annotated UCSC RefSeq isoforms per gene. The number of transcribed bases per transcription unit was calculated as the sum of the coverage of evident (sequenced) fragment parts (read pairs only) for all fragments in addition to the sum of the coverage of the inner mate interval if not entirely overlapping a RefSeq annotated intron (UCSC RefSeq GRCm38). Computational analysis DRB/$TT_{chem}$-seq data were processed using a previously published protocol[32]. In brief, reads were aligned to human GRCm38 (mm10). Read depth coverage was normalized to account for differences between samples using a scale factor derived from a spike-in aligned and counted against *Drosophila melanogaster*. Biological replicate alignments were combined for the purpose of visualization and wave-peak analysis to increase read-depth coverage.

A set of non-overlapping protein-coding genes of >60 kb and <300 kb was selected for wave-peak analysis. A meta-gene profile was calculated by taking a trimmed mean of each base-pair coverage in the region around the TSS. This was further smoothened using a spline. Wave peaks were called at the maximum points on the spline, with the stipulation that the peak must advance with time before being subjected to manual review. Elongation rates (kb per min) were calculated by fitting a linear model to the wave-peak positions as a function of time. For elongation-rate analysis, the following criteria were used to filter genes: The 0 min timepoint DMSO control sample was required to show expression of the gene (mean expression of >100 rpm by TT-seq) and was required to have a wave peak called within 10 kb of the pausing peak region to remove artifacts. Genes showing an increase in transcription in the DMSO control sample for the time course were identified by requiring the wave peak in the 0 min sample to be less than the wave peak in the 10 min timepoint wave peak, and the wave peak in the 10 min sample to be less than the wave peak in the 20 min timepoint, and the wave peak in the 20 min sample to be less than the wave peak in the 30 min timepoint. This resulted in the identification of 855 genes, for which elongation rates were calculated for the samples by dividing the wave peak position by the timepoint. Reproducibility of replicates is shown in Supplementary Fig. 3.

## Statistics and reproducibility

The statistical details of the experiments can be found in the figure legends and in the Methods. Western blotting in Figs. 1b,c,f–h, 3a and 5b,c,j was independently performed three times with similar results, and western blotting in Extended Data Figs. 8c,f,g and 9c,e was performed twice as biologically independent experiments.

## Reporting summary

Further information on research design is available in the Nature Portfolio Reporting Summary linked to this article.

## Data availability

All related raw sequencing and processed data have been deposited at the Gene Expression Omnibus under accession number GSE181714.

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

**Acknowledgements** We thank H. Damhofer, K. Nishimura, D. Shlyueva, S. Sidoli, Z. Li, Z. Sun, Y. Lin and Y. Fang for technical advice and reagents; C. Huang, A. Radzisheuskaya, D. Shlyueva and R. Armstrong for comments on the manuscript; and members of the Helin laboratory for discussions and support. This work was supported by the Memorial Sloan Kettering Cancer Center Support Grant (no. NIH P30 CA008748), a Tri-Institutional Stem Cell grant (no. 2019-035), startup funds from The Institute of Cancer Research and the Novo Nordisk Foundation to the NNF Center for Stem Cell Biology (no. NNF17CC0027852).

**Author contributions** H.W. and K.H. conceived the project. H.W. designed and performed the experiments and bioinformatics analysis. P.V.S. and M.M. performed the MS studies, supervised by R.C.H. Z.F. and X.J. provided input to the project. H.W. and K.H. wrote the manuscript. K.H. supervised the study and acquired funding.

**Competing interests** K.H. is a co-founder of Dania Therapeutics, consultant for Inthera Bioscience and a scientific advisor for MetaboMed and Hannibal Innovation. The other authors declare no competing interests.

**Additional information**
**Correspondence and requests for materials** should be addressed to Kristian Helin.

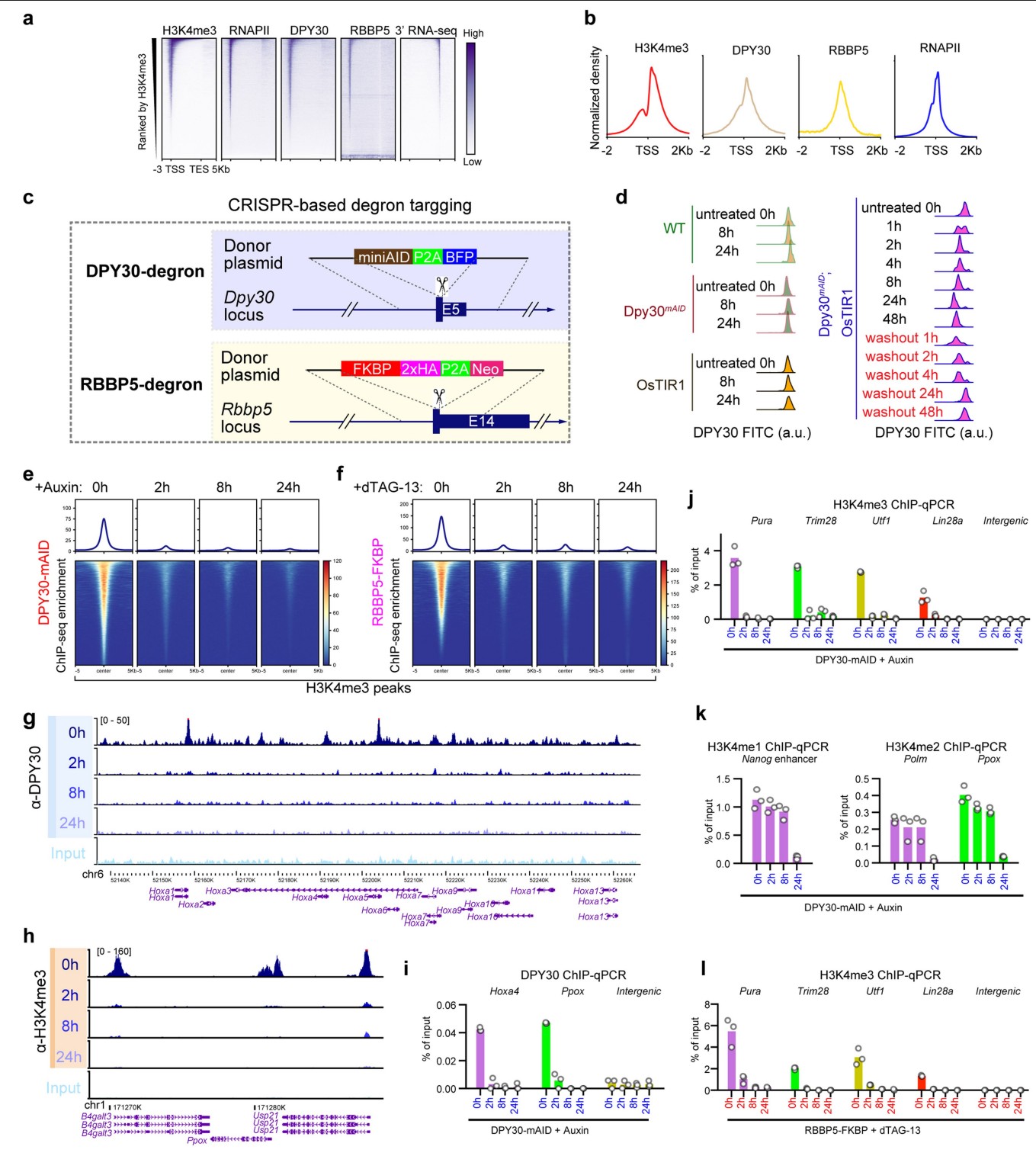

**Extended Data Fig. 1** | See next page for caption.

**Extended Data Fig. 1 | Generation of DPY30–mAID and RBBP5–FKBP mES cell lines.** (**a,b**) DPY30, RBBP5, H3K4me3 and RNAPII occupancy in mES cells plotted around the TSS of all protein-coding genes. Genes were sorted based on H3K4me3 binding levels in the heat maps (**a**). 3′ RNA-seq data is from Quant-seq in mES cells. The average signal in the profiles (**b**) is plotted over the transcription start sites (TSS ± 2 kb) of all protein-coding genes. (**c**) Outline of CRISPR-HDR based knock-in targeting approach generating DPY30-AID and RBBP5–FKBP, by knocking-in mAID and FKBP12 degron tags into the 3′ end of both alleles of endogenous *Dpy30* and *Rbbp5* loci of the E14 mES cell cell line, respectively. The endogenous *Dpy30* gene was edited to encode Auxin inducible degradation (mAID)-BFP at the C terminus of DPY30 in OsTir1 E14 mES cells. The endogenous *Rbbp5* gene was edited to encode FKBP12 tag-Neomycin that enables targeted degradation upon dTAG-13 treatment. (**d**) DPY30 expression in the indicated cell lines as measured by flow cytometry analysis using an anti-DPY30 antibody. (**e,f**) H3K4me3 ChIP–seq heat maps and profiles at H3K4me3 peak centre in control and Auxin-treated cells in DPY30–mAID (E) or dTAG-13-treated in RBBP5–FKBP (F) cells. The signal was plotted over H3K4me3 peaks centre (peak centre ± 5 kb) which are called from WT mES cells. Sites are sorted by the ChIP–seq signals at 0 h. (**g,h**) Integrative genomics viewer (IGV) browser snapshots comparing DPY30 (g) or H3K4me3 (h) enrichments determined by ChIP–seq in control and Auxin-treated in DPY30–mAID degron mES cells. (**i–k**) ChIP-qPCR enrichments of DPY30 (I) and H3K4 methylations (j and k) in DPY30–mAID cells after treatment with Auxin for the indicated times. (**l**) ChIP-qPCR for H3K4me3 in RBBP5–FKBP cells. In panels (i-l) target sites around the promoter of the genes and control region (intragenic chr8: 72,806,101- 72,806,240) were used. For H3K4me1, the enhancer region of *Nanog* was used for DPY30, H3K4me2 and H3K4me3 ChIP-qPCR. Graph shows mean values from technical triplicates (n = 3), from one representative out of two independent experiments.

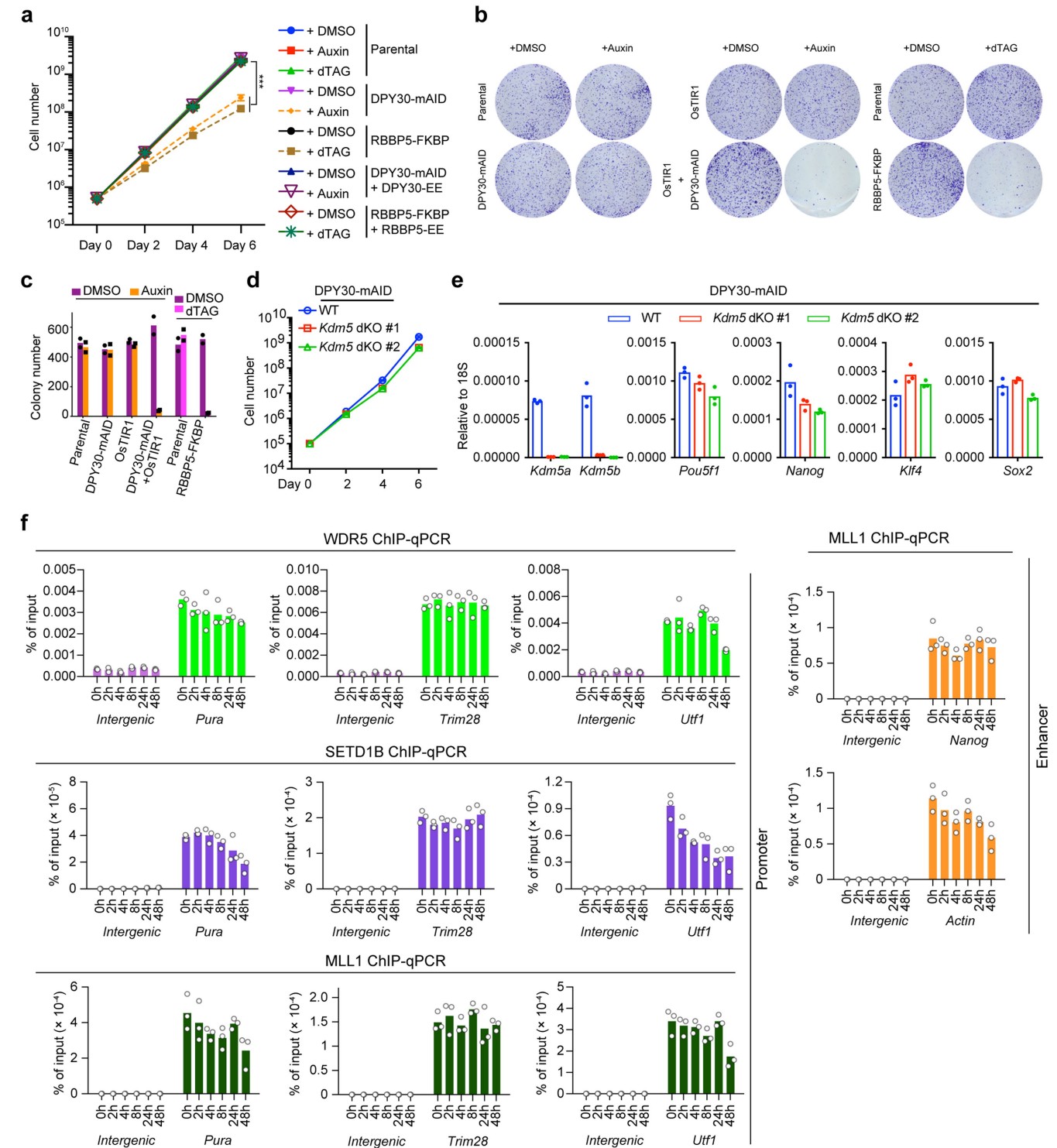

**Extended Data Fig. 2 | DPY30 and RBBP5 are required for cell proliferation.** (**a**) Cell proliferation assays of DPY30–mAID and RBBP5–FKBP cells grown either with or without Auxin/dTAG-13. The line graph represents the mean ± SD of the numbers of cells counted at each time point. Parental represents the parental E14 ES cells without CRISPR editing. Ectopic expression (EE) of DPY30 or RBBP5 rescues the proliferation of defect observed by degrading endogenous DPY30 and RBBP5, respectively. Data are from three biological replicates (n = 3) and were analysed using Two-way ANOVA (n = 3) and represented as mean ± s.d., ***P < 0.001. (**b**) Colony formation assay for DPY30–mAID and RBBP5–FKBP cells grown either with or without Auxin/dTAG-13 for two weeks. (**c**) Quantification of colony formation assays from two independent experiments. Data in bar plots are represented as mean and n = 2 replicates. (**d**) Cell proliferation assays of DPY30–mAID cells after treatment with Auxin at the indicated times with or without *Kdm5ab* dKO. (**e**) RT-qPCR analyses of mRNA expression of DPY30–mAID cells with or without *Kdm5ab* dKO. Two independent dKO clones were chosen for downstream analysis. The values are normalized to 18S rRNA. Graph shows mean values from technical triplicates (n = 3), from one representative out of two independent experiments. (**f**) ChIP-qPCR enrichment for WDR5, SETD1B and MLL1 in DPY30–mAID cells system after treatment with Auxin at the indicated points. Graph shows mean values from technical triplicates (n = 3), from one representative out of two independent experiments.

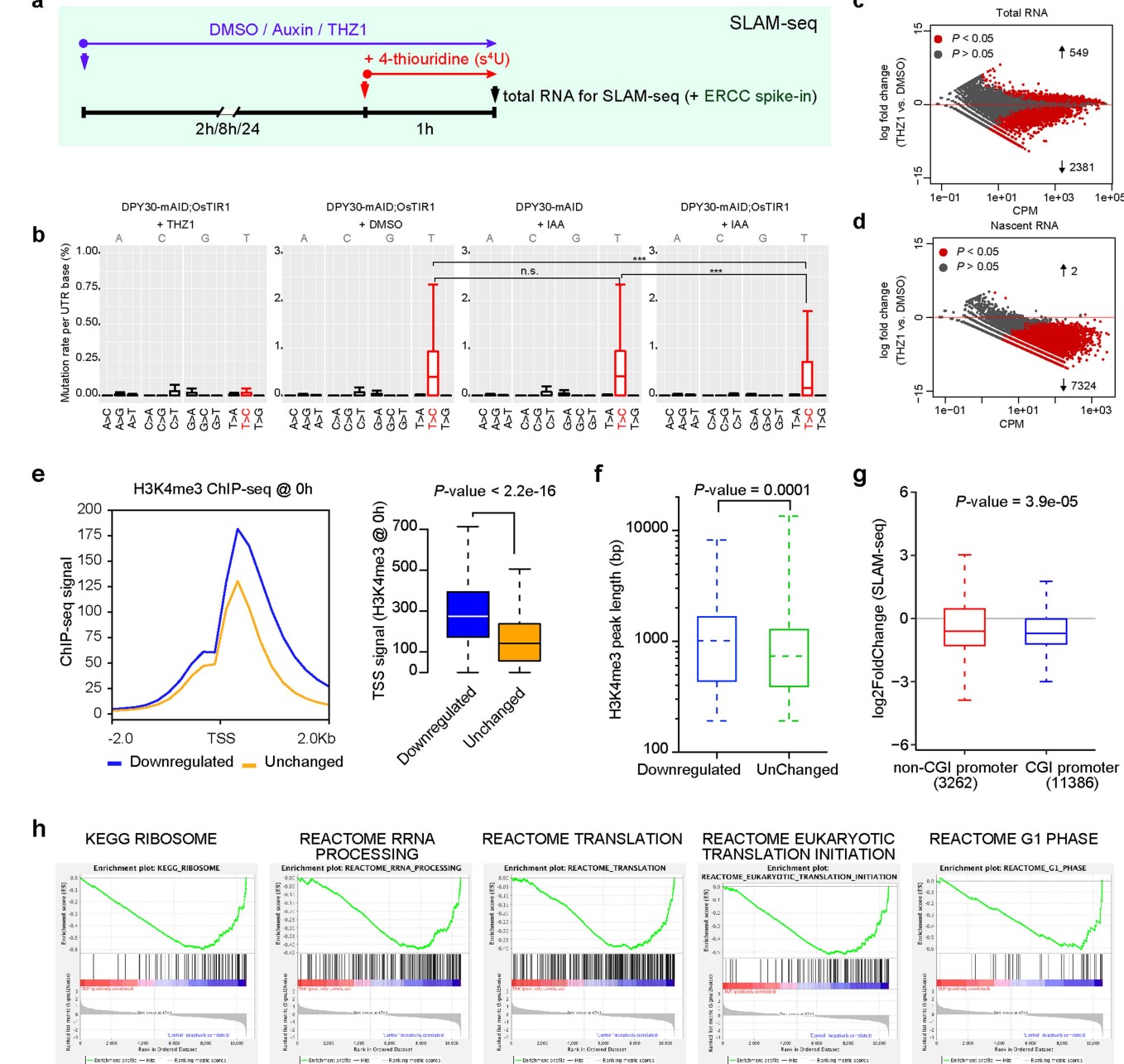

**Extended Data Fig. 3** | See next page for caption.

**Extended Data Fig. 3 | Outline and controls for SLAM-seq experiments.**
(**a**) Experimental design. To validate the SLAM-seq protocol, we performed a pilot experiment for mapping responses to short-term THZ1 (2h) treatment by SLAM-seq in mES cells. SLAM-seq utilizes thymine-to-cytosine (T>C) conversion from 4-thiouridine (4sU)-labelled mRNAs to quantify the abundance of nascent RNA transcripts using 3′ end mRNA-sequencing (Quant-seq). To monitor the consistency and reproducibility of different SLAM-seq data, we inhibited transcription with THZ1 (reduces RNAPII-mediated gene transcription by inhibiting cyclin-dependent kinase 7 (CDK7)). (**b**) Conversion rates for each position of 4-thioU-containing SLAM-seq reads (≥ 2 T>C conversions) before or after Auxin or THZ1 treatment for 2 h. Changes in the abundance of newly synthesized mRNAs (detected in SLAM-seq based on T>C conversions). Average conversion rates (centre line) ± s.d. (whiskers) of two independent experiments (points) are shown. $P$ value (Two-sided Mann-Whitney test) is indicated (***$P < 0.001$, n.s., not significant.). $n = 20,428$ transcripts. The boxplot indicates the median (middle line) and the third and first quartiles (box); the whiskers show the 1.5× IQR above and below the box. (**c,d**) Transcriptional response of the cells treated with THZ1/DMSO for 2h followed by 4sU labelling over 60 min. (c) MA plots comparing total gene expression level with log change in transcription per gene measured by 4-thioU RNA-seq (SLAM-seq). (d) MA plots comparing nascent gene expression levels with log change in transcription per gene measured by SLAM-seq. THZ1 treatment confirmed that transcripts containing T>C conversions of protein-coding genes were broadly repressed, which captured the prominent immediate responses, while the total mRNA level showed fewer changes. $P$-adjusted value by Wald test in DESeq2. (**e**) H3K4me3 levels on TSS before Auxin treatment of downregulated genes ($n = 1,111$) and unchanged genes ($n = 10,107$) measured by SLAM-seq (in response to Auxin treatment for 2h in DPY30−mAID cells). The $p$ value was calculated with a two-sided Wilcoxon test. The boxplot indicates the median (middle line) and the third and first quartiles (box); the whiskers show the 1.5× IQR above and below the box. (**f**) H3K4me3 peak width of downregulated genes and unchanged genes measured by ChIP−seq at steady-state. n= for downregulated and n= for unchanged genes. The $p$ value was calculated with a two-sided Wilcoxon test. The boxplot indicates the median (middle line) and the third and first quartiles (box); the whiskers show the 1.5× IQR above and below the box. (**g**) Box plot showing log2-transformed fold change of nascent transcription (SLAM-seq, 2 h vs. 0 h) of genes containing CGI ($n = 11,386$) and non-CGI ($n = 3,262$) promoters. The $p$ value was calculated with a two-sided Wilcoxon test. The boxplot indicates the median (middle line) and the third and first quartiles (box); the whiskers show the 1.5× IQR above and below the box. (**h**) Gene set enrichment analysis (GSEA) of downregulated genes in DPY30−mAID cells in response to 2 h Auxin treatment.

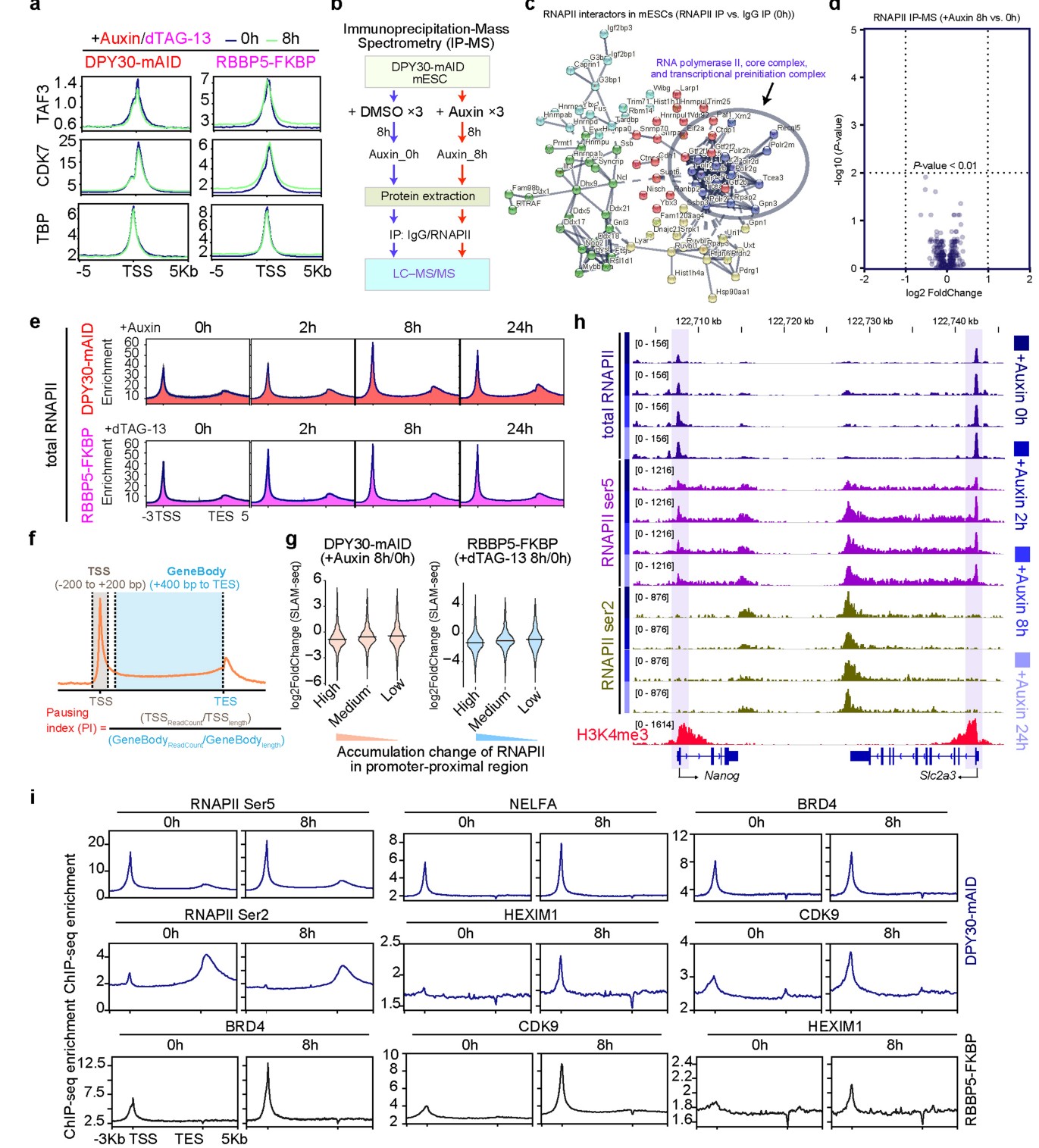

**Extended Data Fig. 4** | See next page for caption.

**Extended Data Fig. 4 | H3K4me3 loss does not have detectable effects on binding of TAF3, CDK7 and TBP to transcription start sites and to PIC formation.** (**a**) ChIP–seq profiles of the indicated proteins using DPY30–mAID and RBBP5–FKBP cells treated with or without Auxin/dTAG-13, respectively for 8 h. The enrichments were plotted over the transcription start sites (TSS ± 5 kb) of protein-coding genes. TSS, transcription start site. (**b**) Outline of the mass spectroscopy proteome profiling strategy for mapping the interaction networks of RNAPII in DPY30–mAID cells treated with or without Auxin. (**c**) String network of protein complexes (*k*-means clustering) showing RNAPII interactors in control cells compared with IgG mock IP. (**d**) Volcano plot showing proteins changing their association with RNAPII in response to acute loss of H3K4me3. The *x* axis displays the enrichment (log2 fold change) of proteins in Auxin-treated cells (Auxin 8 h) compared to DMSO-treated control cells (Auxin 0 h). The *y* axis shows the significance (-log10 *P* value) of enrichment calculated from three biological replicate experiments. A protein was considered an interactor if in one or both comparisons its levels were statistically significantly different (Q value ≤ 0.05, limma test, with *P* values adjusted by the Storey method). (**e**) RNAPII occupancy based on ChIP–seq in the indicated cell lines. Metagene analysis showing the genome-wide enrichment averages on protein-coding genes, data are shown along with 3 kb upstream of the transcriptional start site to 5 kb downstream of the end of each annotated gene. TSS, transcription start site, TES, transcription end site. (**f**) Estimation of a gene's "pausing index" (PI) from RNAPII ChIP–seq data. The promoter is defined as the region covering 200 bp upstream to 200 bp downstream of the TSS; the gene body is defined as the region from 400 bp downstream of the TSS to TES, genes with gene length less than 400 bp are removed for pausing index analysis. (**g**) Violin plots showing changes of gene expression in the indicated samples. Genes were separated into three equal parts based on their accumulation change of RNAPII in promoter-proximal region. (**h**) An IGV snapshot comparing RNAPII, RNAPII Ser 2p and Ser 5p ChIP–seq signals in control and Auxin-treated DPY30–mAID cells at the indicated times. (**i**) Average metagene ChIP–seq profiles for the indicated factors in control and dTAG-13-treated RBBP5–FKBP cells.

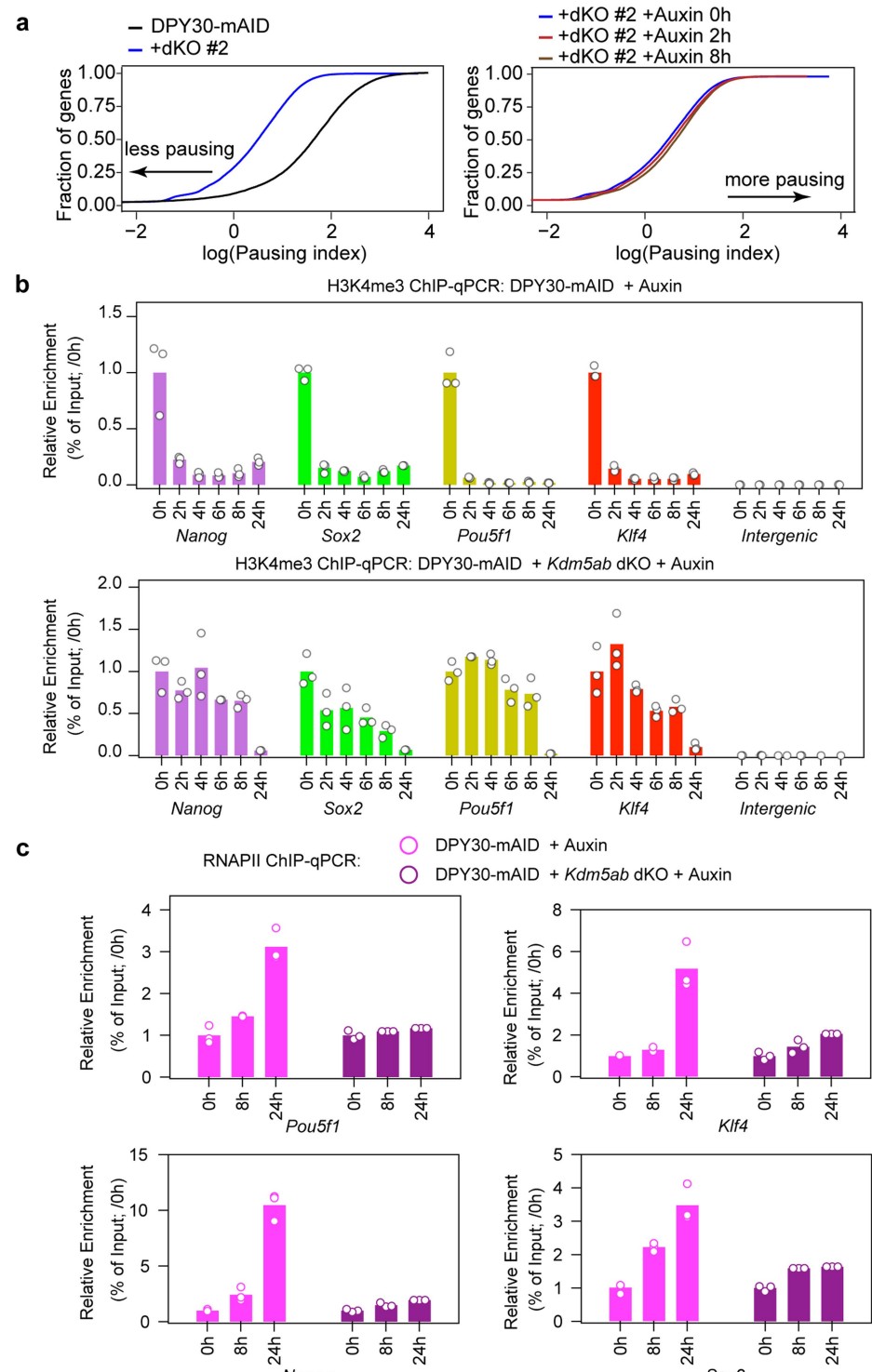

**Extended Data Fig. 5 | The fast turnover of H3K4me3 is dependent on KDM5 demethylases.** (**a**) RNAPII pausing index in DPY30–mAID (black), DPY30–mAID: *Kdm5a/b* dKO (blue, clone #2) and Auxin-treated cells. Higher index values indicate a higher degree of RNAPII pausing. (**b**) ChIP-qPCR enrichment of H3K4me3 in DPY30–mAID cells after treatment with Auxin at the indicated times with or without *Kdm5a/b* dKO. (**c**) ChIP-qPCR signals for RNAPII in DPY30–mAID cells following treatment with Auxin at the indicated times with or without *Kdm5a/b* dKO. For ChIP-qPCR, the target sites around promoter of the indicated genes and control region (intragenic chr8: 72,806,101- 72,806,240) were used. Graph shows mean values from technical triplicates (n = 3), from one representative out of two independent experiments.

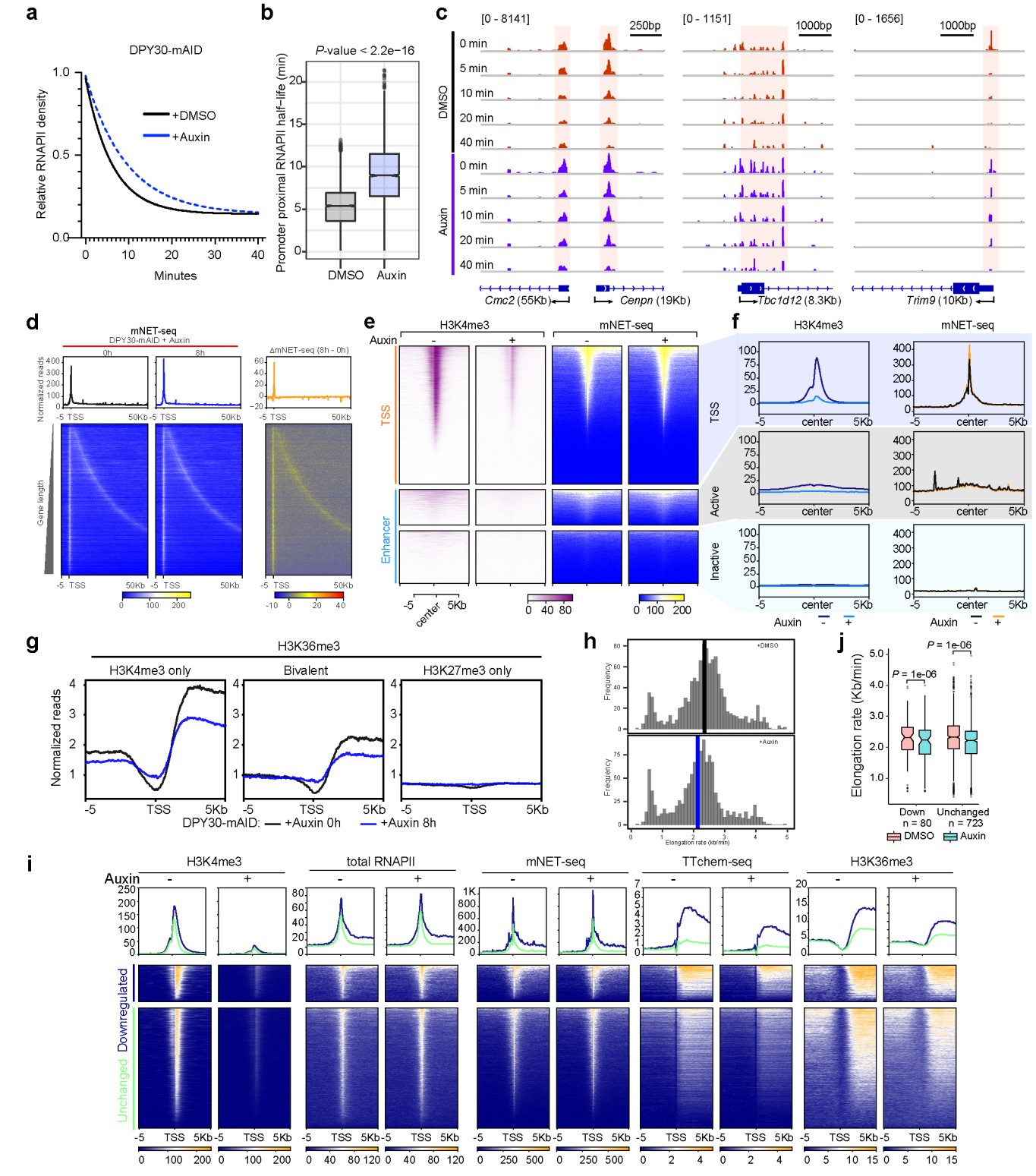

**Extended Data Fig. 6** | See next page for caption.

**Extended Data Fig. 6 | H3K4me3 regulates the paused RNAPII half-life.**
(**a**) Half-lives of paused RNAPII in control and Auxin-treated DPY30-mAID degron cells. The half-life was calculated based on an exponential decay model. Normalized promoter-proximal RNAPII density for each gene is shown over the course of triptolide treatment both for control (DMSO, black) and Auxin (blue) conditions. n = 2 biological replicates. (**b**) Boxplot showing increased paused RNAPII duration following H3K4me3 loss. *P* value was calculated with a two-sided Wilcoxon test, n = 4,007 genes. The boxplot indicates the median (middle line) and the third and first quartiles (box); the whiskers show the 1.5× IQR above and below the box. (**c**) An IGV snapshot comparing mNET–seq signals after triptolide-induced block of transcription in control and Auxin-treated in DPY30-mAID cells at the indicated time points. The decreasing levels of paused RNAPII were observed in the pausing window (shaded areas) over time. (**d**) Heat maps and profiles showing changes in mNET–seq signals upon acute loss of H3K4me3. Heat maps were plotted as increasing gene length over regions 5 kb upstream to 50 kb downstream of the TSS. The right panel display the difference (Δ) in mNET–seq between control (+ Auxin 0 h) and Auxin-treated (+ Auxin 8 h) cells. Genes were sorted by gene length. (**e**–**f**) Heat maps and profiles showing changes of H3K4me3 ChIP–seq and mNET–seq at promoters, active and inactive enhancers upon Auxin treatment (8 h) in DPY30-mAID degron cells. The signals were plotted over the transcription start sites (TSS ± 5 kb) or the centre of enhancers (centre ± 5 kb). (**g**) H3K36me3 ChIP–seq profiles in control (+ Auxin 0 h) and Auxin-treated (+ Auxin 8 h) cells in DPY30–mAID cells. Genes were split by their H3K4me3 and H3K27me3 levels around TSS regions. (**h**) Histogram of RNAPII elongation rates for individual genes between 60 and 300 kb with RPM value ≥100 across all time points (n = 855) with a 10-min wave peak called beyond 2 kb and sequential increase from the TSS over the 10-, 20- and 30-min time points. (**i**) Heat maps and profiles showing changes of ChIP–seq, mNET–seq and TT$_{chem}$-seq at promoters upon Auxin treatment (8 h) in DPY30-mAID degron cells. The downregulated genes and unchanged genes measured by SLAM-seq. The signals were plotted over the transcription start sites (TSS ± 5 kb). (**j**) Boxplot showing RNAPII elongation rates following H3K4me3 loss. *P* value was calculated with a two-sided Wilcoxon test. The number of genes (n) from each group are shown at the bottom. The boxplot indicates the median (middle line) and the third and first quartiles (box); the whiskers show the 1.5× IQR above and below the box.

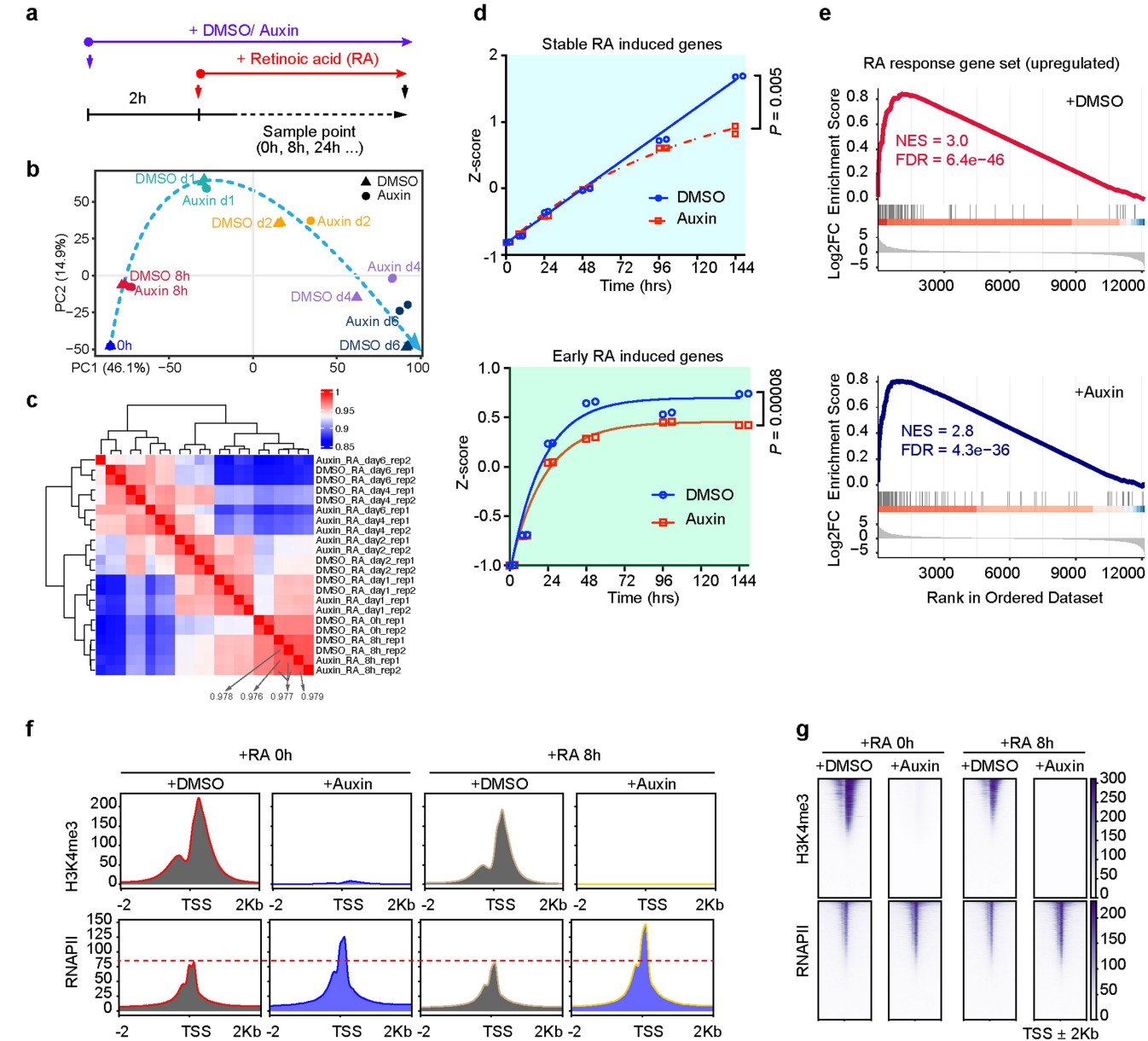

**Extended Data Fig. 7 | Loss of H3K4me3 does not have detectable effects on transcriptional initiation. (a)** Experimental strategy for retinoic acid (RA) differentiation in control and Auxin-treated degron cells. **(b)** Principal component (PC) analysis of RA-induced mRNA expression changes in DPY30–mAID cells prior to treatment with Auxin (circles) or DMSO (triangles) for 2 h at the indicated differentiation points: 0h, 8h, d1, d2, d4, d6. Developmental trajectory is shown by the dashed arrow. **(c)** Spearman correlation heat map of retinoic acid (RA) time course RNA-seq replicates of Auxin-treated and DMSO-treated cells. **(d)** Short- and long-term effects of DPY30 degradation on the expression of RA-induced genes. DPY30–mAID cells were treated with DMSO, Auxin and RA as indicated. The stable RA induced genes were identified from upregulated genes at day 6 in control cells (DMSO). The early RA induced genes were identified from upregulated genes at 8 h in control cells (DMSO). Data

were analysed using Two-way ANOVA (n = 2) and represented as mean ± s.d., **P < 0.01, ***P < 0.001. **(e)** Gene set enrichment analysis (GSEA) analysis of RA induced genes (RA upregulated gene list from previously published data, n = 227). The curve represents the evolution of the density of the genes identified in the RNA-seq. The False Discovery Rate (FDR) is calculated by comparing the actual data with 1000 Monte-Carlo simulations. The NES (Normalized Enrichment Score) computes the density of modified genes in the dataset with the random expectancies, normalized by the number of genes found in the given gene cluster, considering the size of the cluster. **(f,g)** H3K4me3 and RNAPII occupancy before or after RA treatment in control and Auxin-treated degron cells. The enrichments were plotted over the transcription start sites (TSS ± 2 kb) of protein-coding genes. Rows are sorted by decreasing H3K4me3 ChIP–seq occupancy in the control (DMSO RA 0h) cells.

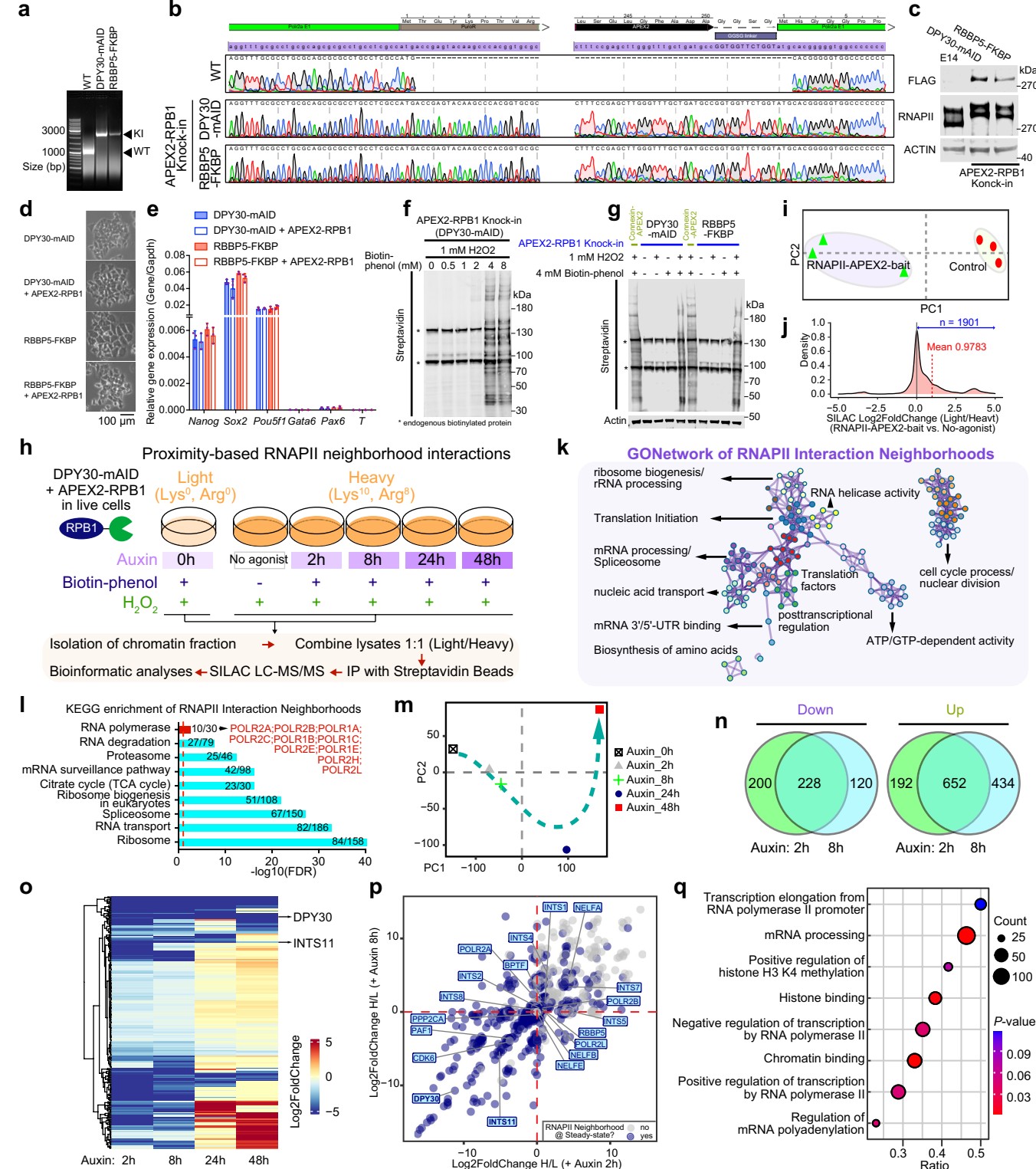

**Extended Data Fig. 8** | See next page for caption.

**Extended Data Fig. 8 | APEX2-based proteomic mapping scheme and characterization of endogenous RNAPII *in vivo* interactions.** (**a**) Genomic confirmation of the APEX-modified *Rpb1* loci in DPY30–mAID and RBBP5–FKBP degron cells. (**b**) Sanger sequencing of the wild type and Flag-APEX2-RPB1 knock-ins. (**c**) Western blot showing the expression of APEX-modified RPB1 in DPY30–mAID and RBBP5–FKBP degron cells. (**d**) Representative brightfield images of mES cell colonies. (**e**) RT-qPCR analysis of the expression of pluripotency and differentiation genes in the knock-in cells. Data are from three biological replicates (n = 3) and are analysed using Two-way ANOVA and represented as mean ± s.d. (**f**) Titration of biotin phenol (BP). Cells stably expressing Flag-APEX2-RPB1 were pre-incubated for 30 min with the indicated concentrations of BP, followed by the addition of 1 mM $H_2O_2$ for 1 min. Cell lysates were probed with Streptavidin-HRP. Proximity biotinylation is optimal at a BP concentration of 4 mM. (**g**) Confirmation of the APEX2 functionality by protein biotinylation in the APEX2-engineered DPY30–mAID and RBBP5–FKBP degron cells. The discrete bands, denoted with asterisks, show APEX2-independent biotinylation by native enzymes. The Connexin-APEX2 overexpressed cell was severed as positive control for the APEX2 system. (**h**) SILAC-based chromatin proteomic strategy for mapping the neighbourhood interaction networks of APEX2-tagged RNAPII. (**i**) Principal component analysis (PCA) of SILAC signal in the RNAPII-APEX2 cells with or without agonist. (**j**) Distribution of SILAC ratio of RNAPII interactions quantified in the chromatin proteomic analyses. Mean log2 SILAC ratio is shown. In total, 1,901 proteins were identified in this experiment. The RNAPII-APEX2-bait (BP+H2O2) population has a right-shifted distribution compared with the no agonist negative control population, which indicates that the log2(SILAC) ratio allows us to distinguish bona fide RNAPII interactions from non-RNAPII interactions. (**k**) GO network showing significantly (q value < 0.001) enriched terms for positive RNAPII interaction neighbourhoods from RNAPII-APEX2 experiment. The most prominent pathways are indicated. Connecting lines show interaction of protein nodes. (**l**) KEGG enrichment and GO network showing significantly (*q* value < 0.001) enriched terms for positive RNAPII interaction neighbourhoods from RNAPII-APEX2 experiment. (**m**) Principal component analysis (PCA) of SILAC signal in the RNAPII-APEX2 DPY30–mAID cells with or without Auxin treatment. Time trajectory is shown by the dashed arrow. (**n**) Venn diagram indicating overlap between up or downregulated targets in the indicated samples. (**o**) Heatmap representing relative protein abundance of DPY30 and selected targets in Auxin treated DPY30–mAID cells. n = 3 independently samples. (**p**) Scatter plot analysis of proteins identified by SILAC in RNAPII-APEX2 DPY30–mAID cells following Auxin treatment for 2 and 8 h. (**q**) Gene ontology-based functional classification of 228 downregulated proteins in RNAPII-APEX2 DPY30–mAID cells following Auxin treatment for 2 and 8 h. The dot size is proportional to the number of members in an enrichment set, and colour intensity reflects the *p* value. Significance based on clusterProfiler analysis with Benjamini-Hochberg-adjusted *P* values.

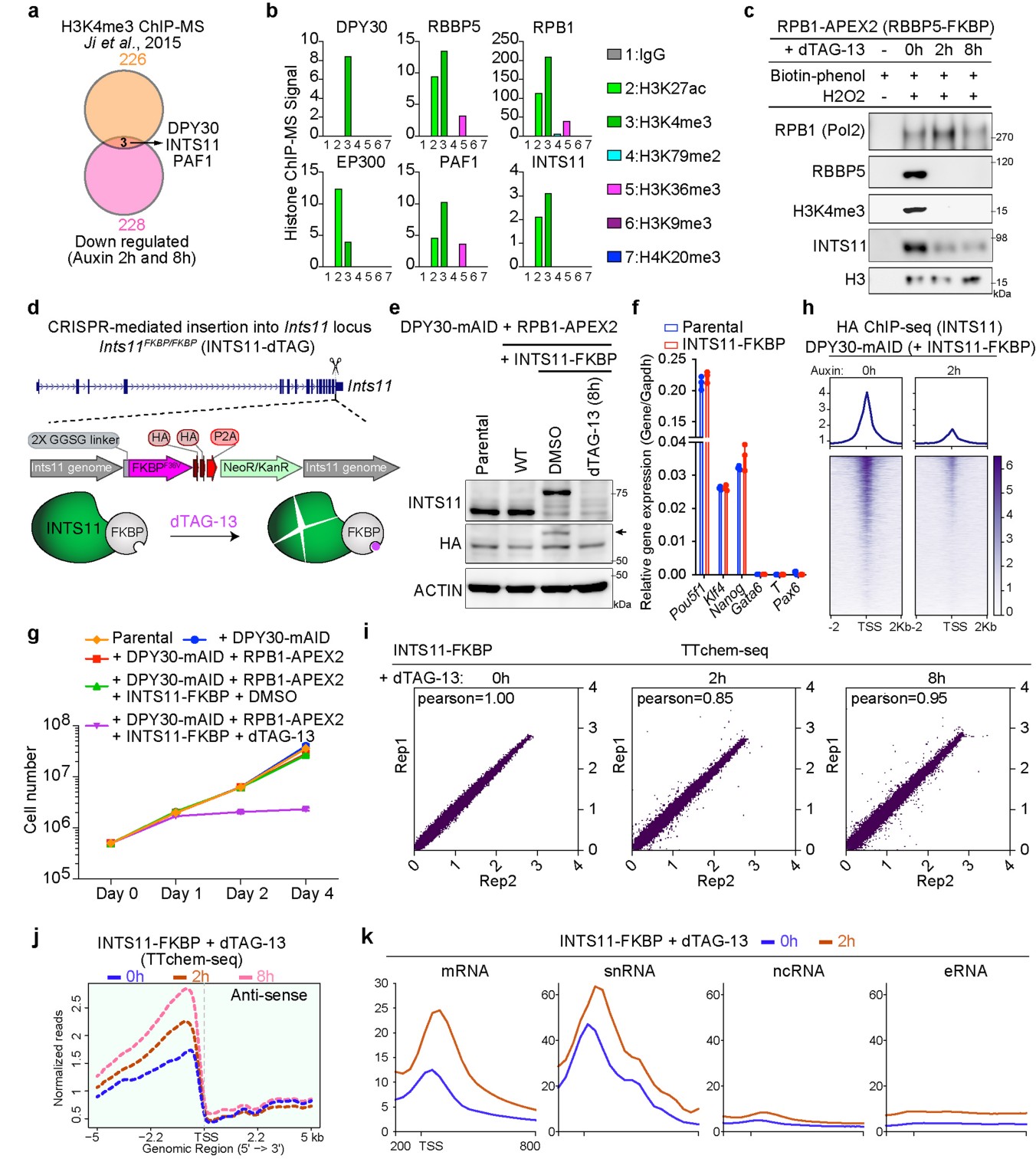

**Extended Data Fig. 9** | See next page for caption.

**Extended Data Fig. 9 | INTS11 is required for nascent transcription.** (**a**) Venn diagram indicating overlap of H3K4me3 interactors and RNAPII-APEX2 dependent interactors from ChIP-MS (chromatin proteomic profiling) data. (**b**) Relative enrichments of selected targets in various ChIP preparations based on ChIP-MS. (**c**) Validation of INTS11 interaction with H3K4me3 in RBBP5–FKBP degron cells at different times after dTAG-13 addition. Biotinylated proteins within lysates were enriched using Streptavidin-coated magnetic beads and analysed by Western blot. In parallel, sample in which $H_2O_2$ was omitted was prepared as negative control. (**d**) Schematic representation of the dTAG INTS11 targeting strategy for the INTS11–FKBP degron mES cells. (**e**) Western blot showing the expression of INTS11 and INTS11–FKBP–HA, using antibodies recognizing INTS11 or the HA in parental and knock-in degron cells. The arrow indicates the specific HA-tagged INTS11–FKBP–HA protein. (**f**) RT-qPCR analysis showing the expression of selected pluripotency and differentiation genes in the parental and INTS11–FKBP knock-in cells. Data are from three biological replicates (n = 3) and are analysed using Two-way ANOVA and represented as mean ± s.d. (**g**) Growth curve analysis of parental and INTS11–FKBP E14 cells treated with or without dTAG-13. (**h**) INTS11 enrichment profiles and heat maps as determined by using the HA-tag in control (0 h) and Auxin-treated (2 h) DPY30–mAID; INTS11–FKBP degron cells. Genome-wide binding averages showed enrichments at the TSS regions (TSS ± 2 kb) of protein coding genes. TSS, transcription start site. Rows were sorted by decreasing ChIP–seq occupancy in the control (0 h) cells. (**i**) Correlations between $TT_{chem}$-seq replicate experiments in INTS11–FKBP degron cells treated with or without dTAG-13 for the indicated times. (**j**) Average profiles for $TT_{chem}$-seq for the upstream anti-sense RNAs of each annotated protein-coding gene in INTS11 degron cells. TSS, transcription start site. (**k**) RNAPII profiles of various subclasses of annotations in INTS11–FKBP degron cells with or without dTAG-13 treatment. TSS, transcription start site. mRNA, messenger RNA. snRNA, small nuclear RNA. ncRNA, non-coding RNA. eRNA, enhancer RNA.

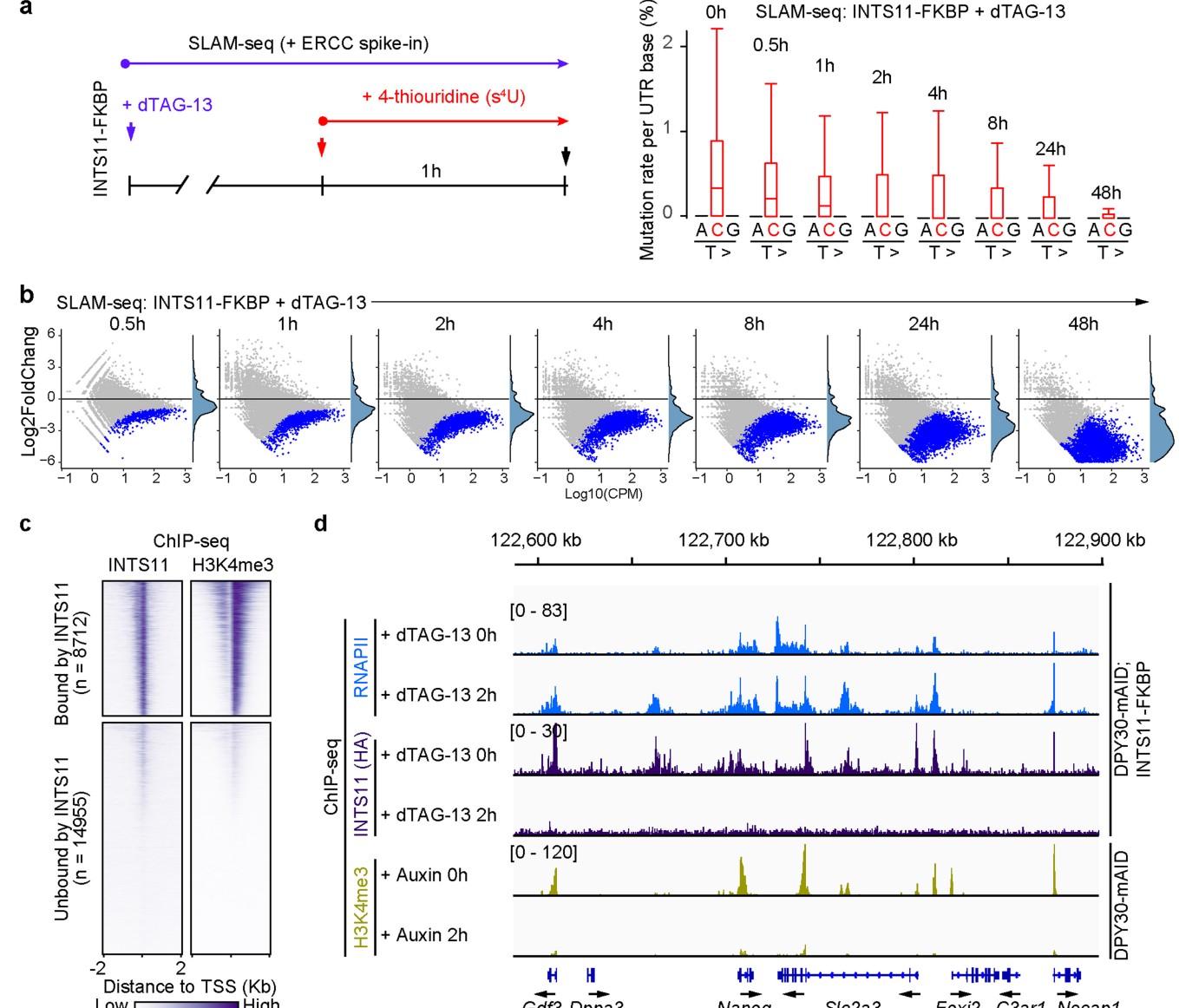

**Extended Data Fig. 10 | Loss of INTS11 causes reduced transcriptional output of protein-coding genes.** (**a**) Experimental design of SLAM-seq for INTS11–FKBP degron cells. Conversion rates for each position of a 4-thioU-containing SLAM-seq reads (≥ 2 T>C conversions) before or after dTAG-13 treatment. Average conversion rates (centre line) ± s.d. (whiskers) are shown. n = 3 biological replicates. The boxplot indicates the median (middle line) and the third and first quartiles (box); the whiskers show the 1.5× IQR above and below the box. (**b**) MA plots depicting changes in nascent transcription (SLAM-seq) at the indicated times after dTAG-13 treatment in INTS11–FKBP cells. n = 3 biological replicates. CPM, counts per million mapped reads. (**c**) H3K4me3 and INTS11 occupancy in mES cells. The enrichments were plotted over the transcription start sites (TSS ± 2 kb) of protein-coding genes. Rows are sorted by decreasing H3K4me3 ChIP–seq occupancy in the WT mES cells. (**d**) An IGV snapshot comparing ChIP–seq signals in control and dTAG-13-treated INTS11–FKBP cells.

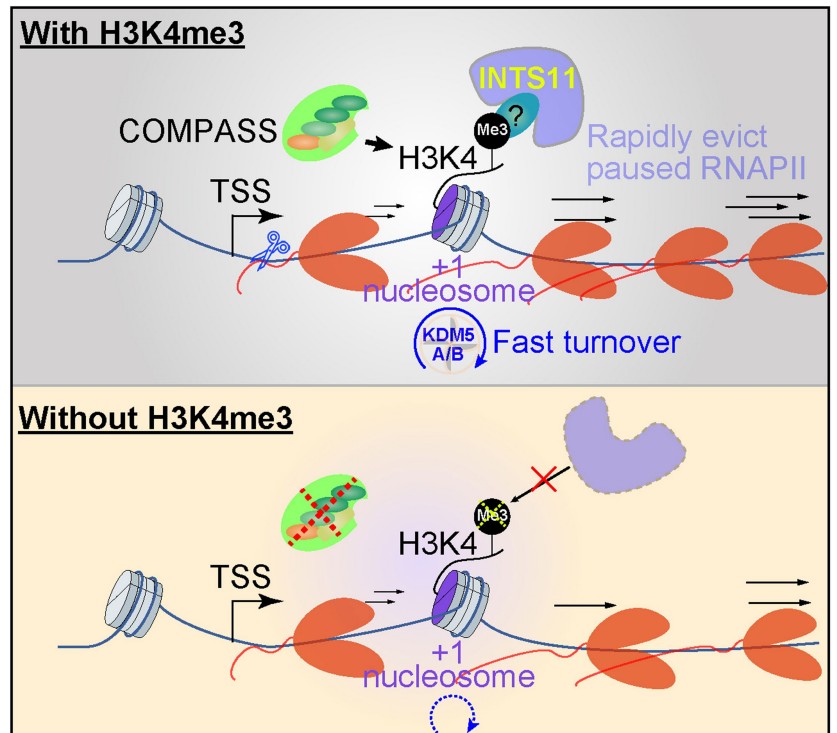

**Extended Data Fig. 11 | A Model for the roles of H3K4me3 in transcription regulation.** H3K4me3 facilitates the recruitment of factors regulating the release of paused RNAPII at the +1 nucleosome. The H3K4me3 at promoter regions is highly dynamic, and it is maintained by an equilibrium between SET1/COMPASS complexes and KDM5 demethylases at highly transcribed genes. The rapid turnover of H3K4me3 ensures that the pausing step is a highly regulated process by Integrator Complex Subunit 11 (INTS11), where an increase in H3K4me3 leads to a decrease in RNAPII pausing and acute depletion leads to an increase RNAPII pausing.

# Reporting Summary

## Statistics

For all statistical analyses, confirm that the following items are present in the figure legend, table legend, main text, or Methods section.

| n/a | Confirmed | |
|---|---|---|
| ☐ | ☒ | The exact sample size (*n*) for each experimental group/condition, given as a discrete number and unit of measurement |
| ☐ | ☒ | A statement on whether measurements were taken from distinct samples or whether the same sample was measured repeatedly |
| ☐ | ☒ | The statistical test(s) used AND whether they are one- or two-sided<br>*Only common tests should be described solely by name; describe more complex techniques in the Methods section.* |
| ☐ | ☒ | A description of all covariates tested |
| ☐ | ☒ | A description of any assumptions or corrections, such as tests of normality and adjustment for multiple comparisons |
| ☐ | ☒ | A full description of the statistical parameters including central tendency (e.g. means) or other basic estimates (e.g. regression coefficient) AND variation (e.g. standard deviation) or associated estimates of uncertainty (e.g. confidence intervals) |
| ☐ | ☒ | For null hypothesis testing, the test statistic (e.g. *F*, *t*, *r*) with confidence intervals, effect sizes, degrees of freedom and *P* value noted<br>*Give P values as exact values whenever suitable.* |
| ☒ | ☐ | For Bayesian analysis, information on the choice of priors and Markov chain Monte Carlo settings |
| ☒ | ☐ | For hierarchical and complex designs, identification of the appropriate level for tests and full reporting of outcomes |
| ☐ | ☒ | Estimates of effect sizes (e.g. Cohen's *d*, Pearson's *r*), indicating how they were calculated |

*Our web collection on statistics for biologists contains articles on many of the points above.*

## Software and code

Policy information about availability of computer code

| | |
|---|---|
| Data collection | Real Time quantitative PR data was collected on QuantStudio 6 Flex Real-time PR System v1.3 (Applied Biosystems). Next-generation sequencing data were collected via Illumina sequencing platforms (NextSeq 550). NextSeq550 control software (2.2.0) were used for high-throughput sequencing data collection. |
| Data analysis | The following software was used for data analysis: GraphPad Prism (v8.0.1) for general statistical analysis and graphing; FlowJo (v10.7.1) for flow cytometry analysis; Enrichr (v.2.1) for gene ontology (GO) enrichment; bcl2fastq (v2.19.0.316), FastQC (v0.11.8), Bowtie2 (v2.4.1), SAMtools (v1.10) , DeepTools (v3.3.0), HISAT (v2.2.1), SlamDunk (v0.3.4 16), DESeq2 (v1.34.0), cutadapt (v3.5) and STAR (v2.7.9a) for genomics data analysis; Proteome Discoverer (v3.1), MSstatsTMT (v2.4.0) and MaxQuant (v1.5.3.30) for proteomics data analysis. |

For manuscripts utilizing custom algorithms or software that are central to the research but not yet described in published literature, software must be made available to editors and reviewers. We strongly encourage code deposition in a community repository (e.g. GitHub). See the Nature Portfolio guidelines for submitting code & software for further information.

## Data

Policy information about availability of data

All manuscripts must include a data availability statement. This statement should provide the following information, where applicable:
- Accession codes, unique identifiers, or web links for publicly available datasets
- A description of any restrictions on data availability
- For clinical datasets or third party data, please ensure that the statement adheres to our policy

All related raw sequencing and related processed data are deposited and available from the Gene Expression Omnibus (GEO) under the accession numbers GSE181714.

# Field-specific reporting

Please select the one below that is the best fit for your research. If you are not sure, read the appropriate sections before making your selection.

☒ Life sciences ☐ Behavioural & social sciences ☐ Ecological, evolutionary & environmental sciences

For a reference copy of the document with all sections, see nature.com/documents/nr-reporting-summary-flat.pdf

# Life sciences study design

All studies must disclose on these points even when the disclosure is negative.

| | |
|---|---|
| Sample size | No specific statistical measure was taken to decide sample size. Minimum sample sizes were predetermined from power estimates based on pilot experiments. |
| Data exclusions | No data has been excluded from analysis. |
| Replication | All attempts at replication were successful. Figure legends state how many times each experiment was performed. |
| Randomization | We did not carry out any randomization because this is either irrelevant or not applicable to this study. |
| Blinding | Blinding was not done since this study relies on the investigator studying differences in cell lines. |

# Reporting for specific materials, systems and methods

We require information from authors about some types of materials, experimental systems and methods used in many studies. Here, indicate whether each material, system or method listed is relevant to your study. If you are not sure if a list item applies to your research, read the appropriate section before selecting a response.

## Materials & experimental systems

| n/a | Involved in the study |
|---|---|
| ☐ | ☒ Antibodies |
| ☐ | ☒ Eukaryotic cell lines |
| ☒ | ☐ Palaeontology and archaeology |
| ☒ | ☐ Animals and other organisms |
| ☒ | ☐ Human research participants |
| ☒ | ☐ Clinical data |
| ☒ | ☐ Dual use research of concern |

## Methods

| n/a | Involved in the study |
|---|---|
| ☐ | ☒ ChIP-seq |
| ☐ | ☒ Flow cytometry |
| ☒ | ☐ MRI-based neuroimaging |

## Antibodies

| | |
|---|---|
| Antibodies used | anti-FLAG M2 Affinity gel SIGMA #A2220 50µl for each IP<br>anti-mini-AID-tag MBL #M2140-3 1:500 for WB<br>beta-ACTIN Abcam #ab6276 1:10000 for WB<br>BPTF Abcam #ab72036 1:1000 for WB<br>BRD4 Abcam #ab128874 1:1000 for WB; 2µg for ChIP<br>CDK7 Santa Cruz #sc-7344 1:500 for WB<br>CDK9 Santa Cruz #sc-13130 1:500 for WB; 2µg for ChIP<br>CHD1 Santa Cruz #sc-271626 1:500 for WB<br>CHD4 Abcam #ab240640 1:1000 for WB<br>DPY30 Bethyl Laboratories #A304-296A 1:1000 for WB; 2µg for ChIP<br>H3 Abcam #ab1791 1:10000 for WB |

H3K36me3 Active Motif #61022 1:2000 for WB; 1µg for ChIP
H3K4me1 Cell Signalling #5326S 1:2000 for WB; 1µg for ChIP
H3K4me2 Cell Signalling #9725S 1:2000 for WB; 1µg for ChIP
H3K4me3 Cell Signalling #9751S 1:2000 for WB; 1µg for ChIP
HA-tag  (C29F4) Cell Signaling #3724S 1:1000 for WB; 2µg for ChIP
HEXIM1 Cell Signaling #12604S 1:1000 for WB; 2µg for ChIP
INTS11 Atlas Antibodies #HPA029025 1:2000 for WB
KDM5A Cell Signaling #3876T 1:1000 for WB; 2µg for ChIP
KDM5B Helin Lab. Home made #Dain78 1:5000 for WB; 2µg for ChIP
NELFA Santa Cruz #sc-365004 1:500 for WB; 2µg for ChIP
NELFE Santa Cruz #sc-377052 1:500 for WB
PAF1 Abcam #ab137519 1:1000 for WB
Pol II Santa Cruz #sc-899; Abcam #ab817; Cell Signaling #2629S 1:1000 for WB; 2µg for ChIP
Pol II (phospho S2) Active motif #61984 1:1000 for WB; 2µg for ChIP
Pol II (phospho S5) Abcam #ab5131 1:1000 for WB; 2µg for ChIP
RBBP5 Bethyl Laboratories #A300-109A 1:1000 for WB; 2µg for ChIP
SPIN1 Abcam #ab118784 1:1000 for WB
SPT16 Santa Cruz #sc-377028 1:500 for WB
SPT6 Cell Signaling #15616S 1:1000 for WB
SSRP1 Cell Signaling  #13421 1:1000 for WB
TAF3 Abcam #ab188332; EMD Millipore #07-1802 1:1000 for WB
INTS11 Atlas Antibodies AB #HPA029025 1:1000 for WB
TFIID (TBP) Santa Cruz #sc-421 1:500 for WB
FITC-conjugated goat α-Rabbit IgG antibody Invitrogen #F-2765 1:500 for Flow

| Validation | Antibodies were validated by manufacturers or validated in previous studies. Statements on antibody validation are present on the manufacturer's websites along with relevant references. Additional validation was done by the use of negative control (control IgG) and control cells for Flow Cytometry analysis. |
|---|---|

## Eukaryotic cell lines

Policy information about cell lines

| Cell line source(s) | E14 mESCs: DPY30-mAID:OsTiR1, DPY30-mAID:OsTiR1_Kdm5dKO, DPY30-mAID:OsTiR1_H3.3dKO, RBBP5-FKBP, DPY30-mAID:OsTiR1_RPB1-APEX2, RBBP5-FKBP_RPB1-APEX2 |
|---|---|
| Authentication | Cells were not authenticated |
| Mycoplasma contamination | Cell lines were tested negative for mycoplasma |
| Commonly misidentified lines (See ICLAC register) | No commonly misidentified cell lines were used |

## ChIP-seq

### Data deposition

☒ Confirm that both raw and final processed data have been deposited in a public database such as GEO.

☒ Confirm that you have deposited or provided access to graph files (e.g. BED files) for the called peaks.

| Data access links  *May remain private before publication.* | All related raw sequencing and related processed data are deposited and available from the Gene Expression Omnibus (GEO) under the accession numbers GSE181714. |
|---|---|
| Files in database submission | GSE181686 mNET-seq on COMPASS-degron cells  GSE181708 Quant-seq on COMPASS-degron cells  GSE181712 TT-seq on COMPASS-degron cells  GSE181892 ChIP-seq on COMPASS-degron cells |
| Genome browser session (e.g. UCSC) | https://www.ncbi.nlm.nih.gov/geo/download/?acc=GSE181714&format=file |

### Methodology

| Replicates | ChIP-seq was performed as individual replicates of two biologically independent degron systerms (Auxin and dTAG) mESCs. Information on reproducibility of the technique and antibody performance as described in the general section on replication of methods. Two replicates for mNET-seq and TT-seq. |
|---|---|
| Sequencing depth | The sequencing was performed in a NextSeq550. The sequencing read information are available as part of the GEO submission (GSE181714). |
| Antibodies | Target Source/Cat No. Application  BRD4 Abcam #ab128874 1:1000 for WB; 2µg for ChIP  CDK9 Santa Cruz #sc-13130 1:500 for WB; 2µg for ChIP |

DPY30 Bethyl Laboratories #A304-296A 1:1000 for WB; 2μg for ChIP
H3K36me3 Active Motif #61022 1:1000 for WB; 1μg for ChIP
H3K4me1 Cell Signalling #5326S 1:1000 for WB; 1μg for ChIP
H3K4me2 Cell Signalling #9725S 1:1000 for WB; 1μg for ChIP
H3K4me3 Cell Signalling #9751S 1:1000 for WB; 1μg for ChIP
HEXIM1 Cell Signaling #12604S 1:1000 for WB; 2μg for ChIP
KDM5A Cell Signaling #3876T 1:1000 for WB; 2μg for ChIP
KDM5B Helin Lab. Home made #Dain78 1:5000 for WB; 2μg for ChIP
NELFA Santa Cruz #sc-365004 1:500 for WB; 2μg for ChIP
Pol II Santa Cruz #sc-899; Abcam #ab817; Cell Signaling #2629S 1:1000 for WB; 2μg for ChIP
Pol II (phospho S2) Active motif #61984 1:1000 for WB; 2μg for ChIP
Pol II (phospho S5) Abcam #ab5131 1:1000 for WB; 2μg for ChIP
RBBP5 Bethyl Laboratories #A300-109A 1:1000 for WB; 2μg for ChIP

| | |
|---|---|
| Peak calling parameters | macs2 with Default parameters |
| Data quality | Data quality was assessed using fastqc |
| Software | bcl2fastq (v2.19.0.316)<br>FastQC (v0.11.8)<br>Bowtie2 (v2.4.1)<br>SAMtools (v1.10)<br>DeepTools (v3.3.0) |

# Flow Cytometry

## Plots

Confirm that:

☒ The axis labels state the marker and fluorochrome used (e.g. CD4-FITC).

☒ The axis scales are clearly visible. Include numbers along axes only for bottom left plot of group (a 'group' is an analysis of identical markers).

☒ All plots are contour plots with outliers or pseudocolor plots.

☒ A numerical value for number of cells or percentage (with statistics) is provided.

## Methodology

| | |
|---|---|
| Sample preparation | mESCs (1 million) were dissociated with Trypsin/EDTA, resuspended in culture medium, spun, and resuspended in PBS. 0.5 mL of cold Fixation Buffer (BioLegend, 420801) was added and then incubated at room temperature for 10 minutes. Subsequently, the cells were labeled with the unconjugated Rabbit DPY30 antibody (Bethyl Laboratories, A304-296A) and subsequently with a FITC-conjugated goat α-Rabbit IgG antibody. |
| Instrument | Beckman Coulter CytoFlex. |
| Software | FlowJo (v10.7.1) |
| Cell population abundance | Over 20,000 cells were counted for each sample |
| Gating strategy | The cells were gated using forward and side scatter parameters (FSC/SSC) for singlets (FSC-A/SSC-A). FL1-A :: B2-510-GFP-A is linked to DPY30 expression. |

☒ Tick this box to confirm that a figure exemplifying the gating strategy is provided in the Supplementary Information.

