## [Peer Review File · Nature]

Manuscript Title: H3K4me3 regulates RNA polymerase II promoter-proximal pause-release

Reviewer Comments & Author Rebuttals

Reviewer Reports on the Initial Version:

Referees' comments:

Referee #1 (Remarks to the Author):

This manuscript addresses the potential role of H3K4me3 in gene regulation. This is an ongoing question, and current reports and models for this vary. Overall, there are some suggestions of roles for H3K4me3 in transcription initiation, and a substantial body of data indicating no general role for H3K4me3 (nicely reviewed in PMID: 28004446). Notably, the H3K4me-complexes have many subunits with different interaction partners; thus perturbation of H3K4me-complexes has pleiotropic effects.

Here, the authors use targeted protein degradation of DPY30 and RBM14, two common subunits of various H3K4me-complexes. Addition of auxin/dTAG causes loss of both proteins, and presumably the associated H3K4me-complexes, from promoters within 1-2 hours and H3K4me3 is found to be lost in the same time frame (with K4me1 and K4me2 turning over more slowly). Using SLAM-seq over a time course of depletion, the authors find that loss of H3K4me3 and H3K4me-complexes for 2h causes a very modest reduction in RNA production (Fig 2) with <400 genes changed. By 8h of auxin treatment, this number grows to ~1000, but this still represents a small fraction of the assayed transcriptome. At this point in reading the manuscript, I expected the authors to come down on the side of H3K4me3 having little overall effect on RNA output. Consistent with this expectation, the authors go on to show convincingly that there is no net effect of losing H3K4me-complexes on promoter levels of factors involved in PIC formation. Thus, this manuscript compellingly demonstrates that loss of H3K4me-complexes causes no defects in transcription initiation.

The authors then go on to study the effect of H3K4me-complexes on pause release. They show a number of CHIP-seq experiments supporting a statistically significant increase in pausing in cells depleted of H3K4me-complexes for 8h. They also include nice data using mNET-seq to provide evidence for a longer half-life of paused Pol II in cells depleted of H3K4me-complexes for 8h. This effect on promoter Pol II levels is not enormous (and is very subtle at 2h) but is intriguing nonetheless. The mechanisms underlying this are not further investigated in this work.

The authors then move on to present data supporting a broader role for H3K4me-complexes in transcription elongation across genes. The conclusion drawn is that loss of H3K4me-complexes reduces Pol II elongation rate. This is curious, since H3K4me3 is located near promoters, but is not present across genes, raising questions about whether the effects observed within gene bodies is due to indirect effects of degrading H3K4me-complexes. The authors don't provide a good explanation for this, leaving me a bit confused.

Throughout the study, the authors work to determine if the reduced transcription observed results from loss of the H3K4me-complexes or the H3K4me3 mark itself. To do so, they repeat several experiments in cells where the H3K4 demethylases have been knocked out (Kdm5ab dKO). In the Kdm5ab dKO cells, auxin mediated depletion of DPY30 still causes rapid loss of H3K4me-complexes but the decay of H3K4me3 is much slower. So, by comparing the kinetics of effects in cells with vs. without Kdm5ab dKO, the authors aim to discriminate between the importance of the complexes vs. the H3K4me3 mark itself. A priori, I like this idea, but am concerned that the DPY-AID+Kdm5ab dKO lines and experiments shown are suboptimal for this.

In summary, this work clearly demonstrates that loss of H3K4me-complexes and H3K4me3 does not impinge on transcription initiation. This study also suggests a role for H3K4me-complexes in stimulation of productive elongation through pause release and elongation rate. But, the mechanisms behind this are unclear, and the effects early on (i.e. within 2h of protein degradation and loss of H3K4me3) are very modest, making it difficult to know which effects are direct vs. indirect.

Major concerns:

1) More information should be provided on the DPY-AID+Kdm5ab dKO lines, including: how many rounds of clonal selection have they gone through? How many different lines were tested here? Since the argument here is strictly kinetic, I would suggest that these results shouldn't rely on a single multiply clonally selected line. Clearly, these line(s) used have established a different baseline for H3K4me3 levels (Fig 1g, h) and intrinsic Pol II behavior (Fig 2g, top).

2) The experiments shown in Figure 2 are difficult to interpret. It is clear from the read counts that these were not side by side experiments sequenced at comparable depth, which would be necessary to allow for direct comparisons in the number of differentially expressed genes. Thus, I would caution against too strong an interpretation of the number of differentially expressed genes at any time point and would focus more on the distributions of Fold changes shown in Figure 2d. Looking at these distributions, however, the 2h time points look really similar between the DPY30-AID and DPY30-AID+Kdm5ab dKO lines, with a very modest but consistent reduction in RNA production across the population. In contrast, the 8h time points look quite different in RNA levels. This is hard for me to conceptualize with the model proposed, since the 2h time points are most direct and relevant comparison, with the greatest difference in the levels of H3K4me3 between DPY30-AID and DPY30-AID+Kdm5ab dKO lines.

In short, my concerns here are two-fold: i) to compare the gene expression changes between DPY30-AID and DPY-AID+Kdm5ab dKO lines, experiments should be performed simultaneously, ii) Based on the data shown, it is not clear that the effects of DPY30 degradation track most closely with the loss of H3K4me3, but instead suggest that dominant effects follow loss of H3K4me-complexes or other indirect phenomenon occurring after 8h.

3) Similarly, were the experiments with DPY30-AID and DPY-AID+Kdm5ab dKO lines shown Fig. 3d and 3g done at the same time? It seems to me that they can't be from the same experiments, since the CDF of Pausing Indices shown for the DPY3-AID line in the absence of auxin is quite different between 3d (left, black line) and 3g (top, black line). It is hard to compare the pausing indices between experiments performed at different times. To best compare Pol II kinetics in DPY30-AID and DPY30-AID+Kdm5ab dKO lines, these experiments should be done in parallel, ideally at 2h of auxin

treatment and with a good nascent RNA assay (PRO-seq or mNET-seq).

4) The results on elongation rates across gene bodies are hard to understand, given the promoter-proximal positioning of H3K4me3. Are the authors proposing that H3K4me2 or me1 within gene bodies, which is also beginning to decrease at this later time point, are involved? Clarifying the proposed mechanisms here would be helpful.

Minor point:

1) Supplemental Figure 8 strikes me as potentially misleading. The Pol II complex is much much larger than shown. There is space for one Pol II between the TSS and first nucleosome (Zeitlinger and Cramer labs), and at most one between a re-wrapped +1 nucleosome and the subsequent +2 nucleosome.

Referee #2 (Remarks to the Author):

I read with great interest the manuscript by Wang et al. entitled “H3K4me3 regulates RNA polymerase II promoter-proximal pause-release”. In this manuscript, the authors generate degron ESC lines in order to acutely deplete major subunits of the SET1/COMPASS methyltransferase complexes, namely DPY30 and RBBP5. This enabled them to address the role of H3K4me3 in transcriptional regulation, overcoming some of the limitations of previous studies (partial knock-downs, redundancy among catalytic components of the complexes, etc.). Using these degron ESC lines and a remarkable battery of genome-wide approaches, the authors show that the depletion of either DPY30 or RBBP5 leads to global defects in transcriptional elongation. This is somehow unexpected, given the preponderant view of H3K4me3 as a histone mark implicated in transcription initiation. Overall, the presented data is of high quality and highly relevant for the field, as it can strongly change the current understanding of a histone mark as profusely investigated as H3K4me3. However, I have some important concerns that I believe the authors should try to address:

Major points:

1. The connection between H3K4me3 and elongation is not completely novel, as it has been reported previously in the context of cancer cells (Chen et al, Nat Genet, 2015) and plants (Ding et al., Plos Genet, 2012). Given the conserved nature of H3K4me3 and of the SET/COMPASS complexes, these previous reports should be acknowledged.
2. Based on the presented data, the authors claim that H3K4me3 mediates the observed defects in transcription elongation. Although the presented data using Kdm5a/b KO supports that interpretation, in my opinion the authors can not dismiss that part or most of the observed transcriptional defects are due to non-enzymatic functions of the SET/COMPASS complexes, as it has been recently shown for MLL3/4 in the context of enhancers (Dorigi et al, Mol Cell). The authors should investigate by CHIP-qPCR whether catalytic (e.g. SET1A/B, MLL1) and non-catalytic (e.g. WDR5) components are bound or not to promoter regions following depletion of DPY30 and/or RBBP5. If complex recruitment is compromised, then the authors should tone down their claims regarding the causative role of H3K4me3. A more conclusive but highly demanding experiment, given the redundancy among SET1 and MLL proteins would be to create catalytic mutants for the

methyltransferases.

3. Previous reports showed that MLL3/4 is important for enhancer function and, in the absence of these proteins, enhancers do not produce eRNAs and gene expression is also severely disrupted (Dorigi et al., *Mol Cell*; Rickels et al., *Nat Genet*). Importantly, the role of these proteins in enhancers seems to be largely independent of their catalytic activity (H3K4me1/2), at least in ESC. Since DPY30 and RBBP5 are common components of all SET1/COMPASS complexes, including those formed by MLL3/4, it is certainly possible that the degnon cell lines used by the authors display defects in enhancer function. Such defects in enhancers could in turn affect gene expression and transcriptional elongation. Moreover, H3K4me3 can also accumulate at active enhancers (Pekowska et al., *Embo J*, 2011). Therefore, the authors should evaluate not only promoter regions but also active enhancers in their different analyses (H3K4me3, eRNAs by NET-seq, etc).

4. The authors claim that the acute loss of H3K4me3 globally affects transcriptional elongation, yet only a subset of the active genes in ESC seems to be downregulated (e.g. 1115 genes after 8 hours of auxin) even after 24 hours of Auxin treatment. Are the downregulated genes somehow different from the unaffected but active genes?. What kind of promoters do the downregulated genes have (CpG-rich, CpG-poor)?. Are the downregulated genes associated with any particular gene ontologies?.

5. Given that not all active genes whose promoters are enriched in H3K4me3 are downregulated upon acute depletion of DPY30/RBBP5, the authors should ideally perform most of their analyses plotting separately the downregulated and the unaffected genes. Moreover, it would be better to always show heatmaps in addition to aggregate plots at least in the supplementary materials. Such analyses will show, even more convincingly, whether PIC formation and transcription initiation are indeed intact even at severely downregulated genes. If transcriptional elongation is globally affected and RNA Pol II pausing increases at most active genes, then why are only a small subset of active genes downregulated?. How many genes show significantly reduced elongation rates and what is their overlap with downregulated genes?.

6. The results section regarding ESC differentiation upon RA treatment is confusing. Gene induction upon RA differentiation is not affected by the loss of H3K4me3, which the authors interpret as supporting evidence for H3K4me3 not being important for transcription initiation. However, for those differentiation genes to be induced, transcriptional elongation needs to take place. Could their results suggest that the acute depletion of DPY30/RBBP5 preferentially affects elongation in ESC but not upon differentiation?. Many genes do not change their expression levels upon ESC differentiation and remain active in RA-treated cells: are those genes downregulated in the Auxin treated cells?. The authors should consider generating DPY30 or RBBP5 degnons in other cell type/s to demonstrate whether their findings can be generalize.

Referee #3 (Remarks to the Author):

The study by Wang et al. investigates the role of H3K4me3 in RNA polymerase II transcription in mouse embryonic stem cells. This histone mark is established by the SET1/COMPASS complex and was so far mainly been linked to transcription initiation. In this study, the authors used CRISPR/Cas9 to generate two knock-in cell lines expressing degnon-tagged versions of the SET1 complex subunits DPY30 and RBBP5. The authors show that both proteins are degraded within less than 2 hours upon

ligand treatment. Both protein depletions led to a loss of H3K4me3 within 2 hours and to a strong depletion of H3K4me2 and of H3K4me within 24 hours. The authors show that the rapid turnover of H3K4me3 was KDM5 dependent. Loss of both SET1 subunits led to a reduction of mRNA synthesis. Interestingly, rapid depletion provoked an accumulation of RNAPII in the promoter-proximal region of genes and an increased half-life of paused RNAPII which both suggest a defect in pause release. The elongation velocity was also reduced upon induced degradation of SET1 subunits. Finally, some evidence is provided that de novo transcription initiation was not affected by the loss of H3K4me3, which challenges prior observations by other laboratories.

The authors provide evidence for an implication of H3K4me3 in promoter-proximal pause release of RNAPII, which is for me the most significant finding of this work. A particular strength is the use of spike-in controls in most of the genome-wide methods that are applied in this study which allow quantitative comparisons between conditions. However, the main limitation of this study is the lack of mechanism of how H3K4me3 regulates promoter-proximal pause release and transcriptional elongation. Furthermore, I am not fully convinced by the claim that H3K4me3 is not required for transcription initiation based on the data that is provided. This work will be of interest for researchers in the transcription field.

Main comments:

(1) One main limitation of this study is that the mechanism of how H3K4me3 is involved in the regulation of RNAPII pause release and transcription elongation remains unknown. How is the decrease in H3K4me3 translated into the pause release defect? What factors play a role in this context? More insights into the molecular mechanism should be provided.

(2) A central claim of the study is that H3K4me3 is not required for transcription initiation. I am not convinced by this claim based on the data that is provided. To support this claim, the authors mainly use a retinoic acid inducible differentiation system -which is not properly described- and measure the steady-state RNA level (RNA-seq) and RNAPII density (ChIP-seq) in the presence and absence of Auxin. My concern is that with this system and the methods that were used mechanisms of transcription initiation and pause release cannot be distinguished. It is not clear whether initiation or pause release mechanisms or both underlie the observation that the induction of genes was not altered between control and Auxin-treated cells. Furthermore, RNA-seq measures the steady-state level of transcripts and is therefore a very indirect way to look at transcription initiation. Along the same line, due to the limited spatial resolution of ChIP-seq data (S7G,H) with RNAPII peaks spanning the entire promoter and promoter-proximal regions, the specific impact of H3K4me3 loss on both regions (promoter and promoter-proximal) cannot be resolved.

In addition, the finding that the induction of genes was not altered upon loss of H3K4me3 is perplexing and not in line with the model that the authors propose. If H3K4me3 plays a general role in pause release, I would have expected a reduction in expression of RA induced genes upon Auxin treatment. However, this seems not be the case. An explanation is needed.

This claim also confuses me given the previous studies by the Patrick Cramer lab and others that have convincingly shown that promoter-proximal pausing has an impact on transcription initiation and the initiation frequency. How can a defect in pause release as observed by the decrease of

H3K4me3 can have no impact on transcription initiation?

(3) The authors performed SLAM-seq time course experiments to show that the mRNA synthesis is reduced by SET1 subunit degradation (Fig. 2). What fraction of genes show a reduction in mRNA synthesis at the different time points? What is the overlap of altered genes per time point between DPY30 and RBBP5? This comparative analysis is important since it is not clear, if replicate measurements were performed. Are there any gene classes that are enriched at different times following the treatment?

The authors also provide evidence by ChIP-seq and NET-seq that RNAPII accumulates in the promoter-proximal region of genes following SET1 subunit ablation. The expectation here is that genes with a higher accumulation of RNAPII in the promoter-proximal region should also show a stronger reduction in the mRNA level. An integrative analysis of the SLAM-seq and ChIP-seq/NET-seq data would be informative.

The accumulation of RNAPII in the promoter-proximal region should be better quantified. The increase in RNAPII can hardly be seen in Figure 3C.

(4) Since the study is based on knock-in cell lines for DPY30 and RBBP5 more quality controls should be provided. Do the degron tags interfere with the cell function? Any impact on cell growth and cell viability? Recent studies have shown that degron tags can lead to a significant reduction of target protein levels already prior to the degrader treatment. Is this also the case for DPY30- and RBBP5-tagged proteins? These key control measures should be included and clearly mentioned in the text.

Minor comments:

(1) It needs to be clarified why DPY30 and RBBP5 were chosen for induced degradation and not the methyltransferase subunit.

(2) Figure 3B: please clarify if the ChIP-seq experiments included spike-ins and if data was normalized to spike-ins. Otherwise, quantitative comparisons between conditions are difficult.

(3) Figure 3C: is the increase in RNAPII density upon degradation significant?

(4) Correlation plots (with correlation coefficients) for biological replicates should be shown for all genome-wide data including ChIP-seq and SLAM-seq data.

(5) Figure S5A: please add correlation coefficients and please also mention which type of correlation analysis was performed.

Author Rebuttals to Initial Comments:

Point-by-point response the Referees' comments

We would like to thank all Referees for the positive and thoughtful comments that has helped us improving the quality of our manuscript. We are encouraged that they found our results exciting and important. Following the Referees' comments and suggestions, we have conducted a series of new experiments and done further analyses to address their concerns.

We have added new data addressing the mechanism by which H3K4me3 regulates RNAPII pause-release. To do this, we have utilized APEX2, a proximity labeling approach (RPB1-APEX2 knock-in cell line) in combination with SILAC-LC-MS/MS to identify proteins associated with RNAPII, depending on the presence of H3K4me3 (new Figure 5). This and additional experiments have led to the identification of the Integrator Complex Subunit 11 (INTS11) protein as interacting with RNAPII depending on H3K4me3. We have further demonstrated that INTS11 is required for promoter-proximal pause-release, which in in agreement with recent published data (new Figure 6).

Below, we provide a point-by-point response to the Referees' comments. We hope that the Referees based on these responses can recommend the manuscript for publication.

Referee #1 (Remarks to the Author):

This manuscript addresses the potential role of H3K4me3 in gene regulation. This is an ongoing question, and current reports and models for this vary. Overall, there are some suggestions of roles for H3K4me3 in transcription initiation, and a substantial body of data indicating no general role for H3K4me3 (nicely reviewed in PMID: 28004446). Notably, the H3K4me-complexes have many subunits with different interaction partners; thus perturbation of H3K4me-complexes has pleiotropic effects.

We thank the Referee for pointing out this important reference. It was added to the revised manuscript.

Here, the authors use targeted protein degradation of DPY30 and RBBP5, two common subunits of various H3K4me-complexes. Addition of auxin/dTAG causes loss of both proteins, and presumably the associated H3K4me-complexes, from promoters within 1-2 hours and H3K4me3 is found to be lost in the same time frame (with K4me1 and K4me2 turning over more slowly). Using SLAM-seq over a time course of depletion, the authors find that loss of H3K4me3 and H3K4me-complexes for 2h causes a very modest reduction in RNA production (Fig 2) with <400 genes changed. By 8h of auxin treatment, this number grows to ~1000, but this still represents a small fraction of the assayed transcriptome. At this point in reading the manuscript, I expected the authors to come down on the side of H3K4me3 having little overall effect on RNA output. Consistent with this expectation, the authors go on to show convincingly that there is no net effect of losing H3K4me-complexes on promoter levels of factors involved in PIC formation. Thus, this manuscript compellingly demonstrates that loss of H3K4me-complexes causes no defects in transcription initiation.

The authors then go on to study the effect of H3K4me-complexes on pause release. They show a number of ChIP-seq experiments supporting a statistically significant increase in pausing in cells depleted of H3K4me-complexes for 8h. They also include nice data using mNET-seq to provide evidence for a longer half-life of paused Pol II in cells depleted of H3K4me-complexes for 8h. This effect on promoter Pol II levels is not enormous (and is very subtle at 2h) but is intriguing nonetheless. The mechanisms underlying this are not further investigated in this work.

The authors then move on to present data supporting a broader role for H3K4me-complexes in transcription elongation across genes. The conclusion drawn is that loss of H3K4me-complexes reduces Pol II elongation rate. This is curious, since H3K4me3 is located near promoters, but is not present across genes, raising questions about whether the effects observed within gene bodies is due to indirect effects of degrading H3K4me-complexes. The authors don't provide a good explanation for this, leaving me a bit confused.

We thank the Referee for the generally positive comments and for bringing up these important points. We agree that it is important to understand the mechanism by which H3K4me3 can regulate RNAPII pause release and how H3K4me3 contributes to elongation. Indeed, this has been our major focus prior to and after the submission of the manuscript. In one of our approaches, we have employed a technique called APEX2-SILAC to study how H3K4me3 can impact the role of RNAPII beyond transcriptional initiation. Our new data show that the RNA endonuclease of integrator subunit 11 (INTS11) is recruited to sites of ongoing transcription (RNAPII), dependent on H3K4me3. INTS11 has recently been shown to play an important role in RNAPII pause release and elongation¹, but little was

known about how it is recruited to chromatin. Our new data have been included in the revised manuscript (p15-p19, new Fig. 5 and 6, and new Extended Data Fig. 12 and 13). These new results couples H3K4me3 to the recruitment of integrator and links it to the regulation of pause-release and transcriptional elongation.

Throughout the study, the authors work to determine if the reduced transcription observed results from loss of the H3K4me-complexes or the H3K4me3 mark itself. To do so, they repeat several experiments in cells where the H3K4 demethylases have been knocked out (*Kdm5ab* dKO). In the *Kdm5ab* dKO cells, auxin mediated depletion of DPY30 still causes rapid loss of H3K4me-complexes but the decay of H3K4me3 is much slower. So, by comparing the kinetics of effects in cells with vs. without *Kdm5ab* dKO, the authors aim to discriminate between the importance of the complexes vs. the H3K4me3 mark itself. A priori, I like this idea, but am concerned that the DPY-AID+*Kdm5ab* dKO lines and experiments shown are suboptimal for this.

In summary, this work clearly demonstrates that loss of H3K4me-complexes and H3K4me3 does not impinge on transcription initiation. This study also suggests a role for H3K4me-complexes in stimulation of productive elongation through pause release and elongation rate. But, the mechanisms behind this are unclear, and the effects early on (i.e. within 2h of protein degradation and loss of H3K4me3) are very modest, making it difficult to know which effects are direct vs. indirect.

We would like to thank the Referee for the critical and constructive comments. In terms of the *Kdm5ab* dKO cells that were used in our study, we were first in fact positively surprised to observe the very fast turnover of H3K4me3 (in response to DPY30 and RBBP5 degradation) and the difference between the turnover of H3K4me3 and that of H3K4me2/H3K4me1. Initially, we thought that the fast H3K4me3 turnover was due to histone exchange (H3.3 at TSS versus H3.1/H3.2 throughout the genome), however, we could demonstrate that the KDM5 demethylases were responsible. By using the *Kdm5ab* dKO cell lines we show a significant delay in H3K4me3 turnover, and this leads to a significant delay in the transcriptional impact of removing DPY30 in the mESCs. In other words, we have established a very strong correlation between the levels of H3K4me3 and the regulation of transcriptional pause-release. We believe that this is probably the only way of studying the distinct and direct role of H3K4me3 in transcriptional regulation. The Referee may suggest catalytic mutants of the MLL/SET proteins; however, the catalytic mutants will impact all the three methylated forms of H3K4 and therefore not be specific to H3K4me3. In addition, we now show that INTS11 is directly involved in the RNAPII pause-release process after H3K4me3 loss as described above.

Major concerns:

1) More information should be provided on the DPY-AID+*Kdm5ab* dKO lines, including: how many rounds of clonal selection have they gone through? How many different lines were tested here? Since the argument here is strictly kinetic, I would suggest that these results shouldn't rely on a single multiply clonally selected line. Clearly, these line(s) used have established a different baseline for H3K4me3 levels (Fig 1g, h) and intrinsic Pol II behavior (Fig 2g, top).

We apologize for the lack of clarity in this part of the manuscript, and we have now provided more details in the methods part of the manuscript. To generate the *Kdm5ab* dKO cells based on DPY30-mAID background, we first generated the *Kdm5a* KO in the DPY30-mAID line (*Kdm5a*^{-/-}) by Cas9 (sgRNAs are listed in Table S1), then we generated the *Kdm5b* knock out in the *Kdm5a*^{-/-} line (at passage three of *Kdm5a*^{-/-} line). Subsequently, two clones with both *Kdm5a* and *Kdm5b* knock out (hereafter termed DPY30-mAID; *Kdm5ab* dKO) were picked up for the downstream analysis. Regarding the different baseline for H3K4me3 in the dKO cell lines, this is indeed what we would expect based on previous publications. To address the concern about the clonally selected lines, we have also evaluated the cell growth analysis in the dKO cell lines, and the data show that there are no detectable differences in cell proliferation. In addition, these clones showed comparable expression levels of pluripotent genes. Thus, the *Kdm5ab* dKO cell lines were relatively easy to generate, and we do not have any evidence for strong clonal selection. In response to the Referee's comment, we have performed RNAPII ChIP-seq in another dKO clone (#2) and these data are consistent in both dKO clones (Revised Extended Data Fig. 2, d and e, and Revised Extended Data Fig. 8a).

2) The experiments shown in Figure 2 are difficult to interpret. It is clear from the read counts that these were not side by side experiments sequenced at comparable depth, which would be necessary to allow for direct comparisons in the number of differentially expressed genes. Thus, I would caution against too strong an interpretation of the number of differentially expressed genes at any time point and would focus more on the distributions of Fold changes shown in Figure 2d. Looking at these distributions, however, the 2h time points look really similar between the DPY30-AID and DPY30-AID+*Kdm5ab* dKO lines, with a very modest but consistent reduction in RNA production across the population. In contrast, the 8h time points look quite different in RNA levels. This is hard for me to conceptualize with the model proposed, since the 2h time points are most direct and relevant comparison, with the greatest difference in the levels of H3K4me3 between DPY30-AID and DPY30-AID+*Kdm5ab* dKO lines.

In short, my concerns here are two-fold: i) to compare the gene expression changes between DPY30-AID and DPY-AID+*Kdm5ab* dKO lines, experiments should be performed simultaneously, ii) Based on the data shown, it is not clear that the effects of DPY30 degradation track most closely with the loss of H3K4me3, but instead suggest that dominant effects follow loss of H3K4me-complexes or other indirect phenomenon occurring after 8h.

We thank the Referee for the insightful comments.

- i. As noticed by the Referee, we did not perform all the SLAM-seq in these cells simultaneously. However, as we mentioned in the main text and Method parts, we performed the SLAM-seq experiments with ERCC spike-in controls. The data analyses in Figure 2 are based on the normalization data to ERCC controls. We would like to argue that experiments should be reproducible and should not be dependent on whether they are done simultaneously if they are properly controlled. The ERCC spike-in controls allow us to compare experiments done at different times. We agree that the MA plots presented in Figure 2a,b,c may be too complicated when comparing different groups. Therefore, we have provided a new Figure 2e (see also below), in

which we plotted the Log2Foldchange of the samples in a boxplot for all the conditions tested. We believe that these new plots better present a comparison between the experiments and clearly show the delayed effect on transcription in the dKO cells.

Revised Fig. 2 (e) Boxplots showing the nascent transcriptional changes (\log_2) for genes in indicated samples across time points with DPY30 or RBBP5 degradation kinetics. Center-lines show median values where box limits represent upper and lower quartiles. P values were calculated with a two-sided Wilcoxon test.

- ii. Regarding the second point raised by the Referee, we would like to highlight some key points. 1) To avoid a biased approach based on one subunit of the SET1/MLL complexes, we decided to target two different core components, DPY30 and RBBP5 using two different degron systems. Subsequently, we used the DPY30-mAID and RBBP5-FKBP degron cells for most of the analyses in our study, and importantly the results obtained from these two independent systems are very consistent, suggesting that these observations are indeed correlated with the loss of H3K4me3. 2) The fact that we can delay the effects on transcription in response to the loss of DPY30 by knocking out Kdm5ab further supports the notion that the effects are due to loss function of SET1/COMPASS complexes – dependent H3K4 trimethylation and not due to indirect function (i.e. a non-enzymatic function). 3) We expected to observe a gradual effect in the SLAM-seq experiments, as both the degradation of the protein in the degron systems and the turnover of H3K4me3 are time dependent. Therefore, we believe these dominant effects were caused by the loss of H3K4me3 itself.

3) Similarly, were the experiments with DPY30-AID and DPY-AID+Kdm5ab dKO lines shown Fig. 3d and 3g done at the same time? It seems to me that they can't be from the same experiments, since the CDF of Pausing Indices shown for the DPY3-AID line in the absence of auxin is quite different between 3d (left, black line) and 3g (top, black line). It is hard to compare the pausing indices between experiments performed at different times. To best compare Pol II kinetics in DPY30-AID and DPY30-AID+Kdm5ab dKO lines, these experiments should be done in parallel, ideally at 2h of auxin treatment and with a good nascent RNA assay (PRO-seq or mNET-seq).

In this case, we agree with the Referee that the analysis should be performed simultaneously. To address this question and as suggested by this Referee, we now performed additional mNET-seq

(single-nucleotide resolution and nascent transcription profiles) experiments in both DPY30-AID and DPY30-AID+Kdm5ab dKO lines in parallel at the same time points (0h, 2h, 8h). Specifically, we performed duplicate mNET-seq experiments (Revised Figure 3f) that confirm the findings. A visible delayed effect on RNAPII pausing is observed in these new mNET-seq data, providing further support that H3K4me3 is essential for the regulation of RNAPII pause-release.

Revised Fig. 3 (f) Boxplots showing the RNAPII pausing index by mNET-seq in DPY30-mAID, DPY30-mAID:Kdm5ab dKO and Auxin-treated cells. Center-lines show median values where box limits represent upper and lower quartiles. P values were calculated with a two-sided Wilcoxon test.

4) The results on elongation rates across gene bodies are hard to understand, given the promoter-proximal positioning of H3K4me3. Are the authors proposing that H3K4me2 or me1 within gene bodies, which is also beginning to decrease at this later time point, are involved? Clarifying the proposed mechanisms here would be helpful.

We apologize for the lack of clarity and have now provided additional experiments as mentioned above, in which we provide more mechanistic insights into how H3K4me3 regulates RNAPII pause release (p15-p19, Fig. 5 and 6, and Extended Data Fig. 12 and 13). INTS11 has recently been shown to play an essential role in RNAPII pause-release and elongation¹, and we discuss the proposed mechanism in the revised manuscript (pages 19-20). Although our study focuses on H3K4me3, it's an interesting question whether H3K4me1 and H3K4me2 are involved as well. As the Referee know, H3K4me1 has been suggested to have a specific role in the regulation of enhancer activity, however, recent publications have questioned such a role. To address the specific role of H3K4me1 and H3K4me2 in transcriptional regulation, we would need other experimental systems, and the question is whether it is feasible to develop such systems. As suggested by the Referee, we have discussed this key point in our revised text (page 19).

Minor point:

1) Supplemental Figure 8 strikes me as potentially misleading. The Pol II complex is much much larger than shown. There is space for one Pol II between the TSS and first nucleosome (Zeitlinger and Cramer labs), and at most one between a re-wrapped +1 nucleosome and the subsequent +2 nucleosome.

We thank the Referee for raising this important point and now we changed the figure accordingly (Extended Data Fig. 14).

Referee #2 (Remarks to the Author):

I read with great interest the manuscript by Wang et al. entitled “H3K4me3 regulates RNA polymerase II promoter-proximal pause-release”. In this manuscript, the authors generate degron ESC lines in order to acutely deplete major subunits of the SET1/COMPASS methyltransferase complexes, namely DPY30 and RBBP5. This enabled them to address the role of H3K4me3 in transcriptional regulation, overcoming some of the limitations of previous studies (partial knock-downs, redundancy among catalytic components of the complexes, etc.). Using these degron ESC lines and a remarkable battery of genome-wide approaches, the authors show that the depletion of either DPY30 or RBBP5 leads to global defects in transcriptional elongation. This is somehow unexpected, given the preponderant view of H3K4me3 as a histone mark implicated in transcription initiation. Overall, the presented data is of high quality and highly relevant for the field, as it can strongly change the current understanding of a histone mark as profusely investigated as H3K4me3. However, I have some important concerns that I believe the authors should try to address:

We thank the Referee for the positive remarks on the importance and relevance of our work. In the following, we provide our response to the Referee’s comments point-by-point.

Major points:

1. The connection between H3K4me3 and elongation is not completely novel, as it has been reported previously in the context of cancer cells (Chen et al, Nat Genet, 2015) and plants (Ding et al., Plos Genet, 2012). Given the conserved nature of H3K4me3 and of the SET/COMPASS complexes, these previous reports should be acknowledged.

We thank the Referee for pointing out these studies and have now included references to them in our revised manuscript.

2. Based on the presented data, the authors claim that H3K4me3 mediates the observed defects in transcription elongation. Although the presented data using Kdm5a/b KO supports that interpretation, in my opinion the authors can not dismiss that part or most of the observed transcriptional defects are due to non-enzymatic functions of the SET/COMPASS complexes, as it has been recently shown for MLL3/4 in the context of enhancers (Dorigi et al, Mol Cell). The authors should investigate by CHIP-qPCR whether catalytic (e.g. SET1A/B, MLL1) and non-catalytic (e.g. WDR5) components are bound or not to promoter regions following depletion of DPY30 and/or RBBP5. If complex recruitment is compromised, then the authors should tone down their claims regarding the causative role of H3K4me3. A more conclusive but highly demanding experiment, given the redundancy among SET1 and MLL proteins would be to create catalytic mutants for the methyltransferases.

The Referee raises an interesting point about the non-enzymatic functions of the SET/COMPASS complexes, and this is indeed the key question. 1) We have followed the Referee’s suggestion and performed additional CHIP-qPCR for WDR5, SETD1B and MLL1 in the degron cells. These are shown in the new Extended Data Fig. 3. We found loss of DPY30 did not lead to a decrease of the binding of any

of these components to chromatin. 2) In addition, we also did not observe any changes in the association of these proteins with RNAPII by APEX2-SILAC/MS (new Figure 5), further providing evidence that their recruitment is uncompromised in the absence of DPY30. 3) For the third point, we must disagree with the Referee that the catalytic mutants of the SET and MLL proteins will give more conclusive results. We do not think so, because the loss of the catalytic activity will affect all three methylation states of H3K4. Moreover, it would be very difficult to develop a time-dependent degradation system by using the catalytic mutants of all six SET and MLL proteins.

3. Previous reports showed that MLL3/4 is important for enhancer function and, in the absence of these proteins, enhancers do not produce eRNAs and gene expression is also severely disrupted (Dorigi et al., *Mol Cell*; Rickels et al., *Nat Genet*). Importantly, the role of these proteins in enhancers seems to be largely independent of their catalytic activity (H3K4me1/2), at least in ESC. Since DPY30 and RBBP5 are common components of all SET1/COMPASS complexes, including those formed by MLL3/4, it is certainly possible that the degon cell lines used by the authors display defects in enhancer function. Such defects in enhancers could in turn affect gene expression and transcriptional elongation. Moreover, H3K4me3 can also accumulate at active enhancers (Pekowska et al., *Embo J*, 2011). Therefore, the authors should evaluate not only promoter regions but also active enhancers in their different analyses (H3K4me3, eRNAs by NET-seq, etc).

The Referee raises another interesting point as to whether the effects also can be caused by the enhancer function after acute loss of DPY30 or RBBP5. Our results do not support such mechanism, because: 1) Our new ChIP-qPCR data on WDR5, SETD1B and MLL1 suggest that the recruitment of these components from COMPASS complexes are uncompromised in the acute degradation cells (see also above response to point 1). 2) We have included an analysis of the enhancer regions as suggested by the Referee in the revised version of the manuscript. This analysis shows that H3K4me3 is at relative higher levels in active enhancers compared with inactive enhancers, however the abundance of H3K4me3 in enhancer regions are significantly lower than the promoter regions (new Extended Data Fig. 9, e and f). In addition, we did not any find significant changes in RNAPII occupancy on enhancers after auxin-treatment by mNET-seq.

4. The authors claim that the acute loss of H3K4me3 globally affects transcriptional elongation, yet only a subset of the active genes in ESC seems to be downregulated (e.g. 1115 genes after 8 hours of auxin) even after 24 hours of Auxin treatment. Are the downregulated genes somehow different from the unaffected but active genes?. What kind of promoters do the downregulated genes have (CpG-rich, CpG-poor)?. Are the downregulated genes associated with any particular gene ontologies?.

We thank the Referee for the helpful comments. We performed such analysis in the revised manuscript. These are shown in the new Extended Data Fig. 5. In brief, we found relatively higher H3K4me3 levels were enriched at the rapidly downregulated genes compared with the unaffected genes. In addition, the rapidly downregulated genes have more broad peaks of H3K4me3 and the nascent RNA were significantly decreased in CGI genes compared with non-CGI genes. GO analyses

show that the downregulated genes are enriched for genes involved in processes such as ribosome biogenesis, translation and cell cycle progress.

5. Given that not all active genes whose promoters are enriched in H3K4me3 are downregulated upon acute depletion of DPY30/RBBP5, the authors should ideally perform most of their analyses plotting separately the downregulated and the unaffected genes. Moreover, it would be better to always show heatmaps in addition to aggregate plots at least in the supplementary materials. Such analyses will show, even more convincingly, whether PIC formation and transcription initiation are indeed intact even at severely downregulated genes. If transcriptional elongation is globally affected and RNA Pol II pausing increases at most active genes, then why are only a small subset of active genes downregulated?. How many genes show significantly reduced elongation rates and what is their overlap with downregulated genes?.

We thank the Referee for these insightful comments. Following the suggestions, 1) we now included separate heatmaps of the downregulated and unaffected genes from the existing ChIP-seq, mNET-seq and TT-seq data (revised Extended Data Fig. 10c). We do see increased RNAPII pausing around promoter regions in both group genes, suggesting a general role of H3K4m3 in transcriptional regulation. 2) Because only a set of non-overlapping protein-coding genes of >60 kb and <300 kb was selected for elongation analysis (see Methods), the overlap analysis is difficult, instead, we replotted the elongation rates by splitting the downregulated and unaffected genes (revised Extended Data Fig. 10d). As we mentioned above, we indeed found that the effects in both degradation systems are time-dependent, and that the rapid response genes (SLAM-seq) have higher basic levels of H3K4me3 with broad peaks, these may result in the various responses in the degron systems. Based on our additional data on INTS11, we believe there is a general role of H3K4m3 in transcriptional regulation.

6. The results section regarding ESC differentiation upon RA treatment is confusing. Gene induction upon RA differentiation is not affected by the loss of H3K4me3, which the authors interpret as supporting evidence for H3K4me3 not being important for transcription initiation. However, for those differentiation genes to be induced, transcriptional elongation needs to take place. Could their results suggest that the acute depletion of DPY30/RBBP5 preferentially affects elongation in ESC but not upon differentiation?. Many genes do not change their expression levels upon ESC differentiation and remain active in RA-treated cells: are those genes downregulated in the Auxin treated cells?. The authors should consider generating DPY30 or RBBP5 degrons in other cell type/s to demonstrate whether their findings can be generalized.

We are sorry that the Referee finds the differentiation experiments confusing. The aim of these experiments was to understand whether H3K4me3 is required for transcriptional initiation. As shown in Figure 4i, RA-induced genes are indeed induced (RNA-seq) in H3K4me3-depleted cells, and RNAPII enrichment (ChIP-seq) is still increased in response to RA-treatment in the absence of H3K4me3. Our point is that RA-induced genes can still be induced (although not to the same extent) after H3K4me3 loss, and that H3K4me3 loss still allows the recruitment of RNAPII to the TSS of RA-regulated genes.

However, we agree that these assays cannot distinguish between transcriptional initiation or elongation. And in response to the Referee's concern, we performed mNET-seq to obtain single-nucleotide resolution in the RA-differentiation experiments. The correlation analysis of the mNET-seq showed that loss of H3K4me3 did not impair the rewiring of RNAPII at promoter-proximal region of genes induced by RA treatment (Figure 4j). Taken together, our multi-omics results suggest that H3K4me3 is not required for RNAPII loading and therefore indirectly not for transcriptional initiation.

As suggested by the Referee, we also tried to generate DPY30 and RBBP5 degrons in other cell types. We tried to do this in immortalized MEFs and B16-F10 melanoma cells, however, even though we screened more than one thousand single-cell clones of each cell line, we were unable to identify cells with homozygous insertion of the degron-tags.

Referee #3 (Remarks to the Author):

The study by Wang et al. investigates the role of H3K4me3 in RNA polymerase II transcription in mouse embryonic stem cells. This histone mark is established by the SET1/COMPASS complex and was so far mainly linked to transcription initiation. In this study, the authors used CRISPR/Cas9 to generate two knock-in cell lines expressing degron-tagged versions of the SET1 complex subunits DPY30 and RBBP5. The authors show that both proteins are degraded within less than 2 hours upon ligand treatment. Both protein depletions led to a loss of H3K4me3 within 2 hours and to a strong depletion of H3K4me2 and of H3K4me within 24 hours. The authors show that the rapid turnover of H3K4me3 was KDM5 dependent. Loss of both SET1 subunits led to a reduction of mRNA synthesis. Interestingly, rapid depletion provoked an accumulation of RNAPII in the promoter-proximal region of genes and an increased half-life of paused RNAPII which both suggest a defect in pause release. The elongation velocity was also reduced upon induced degradation of SET1 subunits. Finally, some evidence is provided that de novo transcription initiation was not affected by the loss of H3K4me3, which challenges prior observations by other laboratories.

The authors provide evidence for an implication of H3K4me3 in promoter-proximal pause release of RNAPII, which is for me the most significant finding of this work. A particular strength is the use of spike-in controls in most of the genome-wide methods that are applied in this study which allow quantitative comparisons between conditions. However, the main limitation of this study is the lack of mechanism of how H3K4me3 regulates promoter-proximal pause release and transcriptional elongation. Furthermore, I am not fully convinced by the claim that H3K4me3 is not required for transcription initiation based on the data that is provided. This work will be of interest for researchers in the transcription field.

We thank the Referee for the helpful and complimentary comments. We have included mechanistic data in the manuscript as outlined in response to Referee #1 showing that H3K4me3 is required for INTS11 recruitment and promoter-proximal pause release (p15-p19, Fig. 5 and 6, and Extended Data Fig. 12 and 13).

Main comments:

(1) One main limitation of this study is that the mechanism of how H3K4me3 is involved in the regulation of RNAPII pause release and transcription elongation remains unknown. How is the decrease in H3K4me3 translated into the pause release defect? What factors play a role in this context? More insights into the molecular mechanism should be provided.

Please see our responses to Referee 1 above. As mentioned above, we have included new results showing the requirement of H3K4me3 for the recruitment of INTS11. Moreover, we also included data using newly constructed INTS11-degron cell lines to further validate that H3K4me3-dependent recruitment of the Integrator Complex Subunit 11 (INTS11) is essential for the regulation of promoter proximal pausing. These are shown in the new Fig. 5 and Fig. 6, and Extended Data Fig. 12-13.

(2) A central claim of the study is that H3K4me3 is not required for transcription initiation. I am not convinced by this claim based on the data that is provided. To support this claim, the authors mainly

use a retinoic acid inducible differentiation system -which is not properly described- and measure the steady-state RNA level (RNA-seq) and RNAPII density (ChIP-seq) in the presence and absence of Auxin. My concern is that with this system and the methods that were used mechanisms of transcription initiation and pause release cannot be distinguished. It is not clear whether initiation or pause release mechanisms or both underlie the observation that the induction of genes was not altered between control and Auxin-treated cells. Furthermore, RNA-seq measures the steady-state level of transcripts and is therefore a very indirect way to look at transcription initiation. Along the same line, due to the limited spatial resolution of ChIP-seq data (S7G,H) with RNAPII peaks spanning the entire promoter and promoter-proximal regions, the specific impact of H3K4me3 loss on both regions (promoter and promoter-proximal) cannot be resolved.

In addition, the finding that the induction of genes was not altered upon loss of H3K4me3 is perplexing and not in line with the model that the authors propose. If H3K4me3 plays a general role in pause release, I would have expected a reduction in expression of RA induced genes upon Auxin treatment. However, this seems not be the case. An explanation is needed.

We thank the Referee for these insightful comments. We agree that it is difficult to distinguish between transcription initiation and elongation simultaneously by the steady-state RNA-seq and ChIP-seq. As we agree these data (RNA-seq and ChIP-seq) do not rule out an effect on transcriptional initiation, and in response to the Referee's suggestion, we have performed mNET-seq to obtain single-nucleotide resolution in the RA-differentiation experiments. The correlation analysis of the mNET-seq showed that loss of H3K4me3 did not impair the rewiring of RNAPII at promoter-proximal region of genes induced by RA treatment (Fig. 4j). Therefore, we conclude that H3K4me3 is not required for RNAPII loading and therefore indirectly not for transcriptional initiation.

We apologize for the lack of clarity leading to the Referee's misunderstanding of the RA experiments. Gene expression is indeed induced in the absence of H3K4me3 by RA-treatment. However, they are not induced to the same extent (Figure 4, h and i), which is consistent with the model we propose. Since this does not appear to be clear from the previous text, we now clarify and explain this in the revised version of the manuscript (page 15).

This claim also confuses me given the previous studies by the Patrick Cramer lab and others that have convincingly shown that promoter-proximal pausing has an impact on transcription initiation and the initiation frequency. How can a defect in pause release as observed by the decrease of H3K4me3 can have no impact on transcription initiation?

This is an interesting point. Although we do not disagree with the 'pause-initiation limit' concept, our understanding is that it has only been observed when transcription is blocked by very strong transcriptional inhibitors, which could potentially introduce artifactual changes. Based on our results in native, instead the words "no impact", we would only like to highlight that we do not observe a dramatic change in transcriptional initiation after H3K4me3 loss.

(3) The authors performed SLAM-seq time course experiments to show that the mRNA synthesis is reduced by SET1 subunit degradation (Fig. 2). What fraction of genes show a reduction in mRNA synthesis at the different time points? What is the overlap of altered genes per time point between DPY30 and RBBP5? This comparative analysis is important since it is not clear, if replicate measurements were performed. Are there any gene classes that are enriched at different times following the treatment?

We thank the Referee for these questions, which are similar to the ones asked by Referee 2 point #4, and we now addressed them in the revised version of the manuscript (Extended Data Fig. 5a-d).

The authors also provide evidence by CHIP-seq and NET-seq that RNAPII accumulates in the promoter-proximal region of genes following SET1 subunit ablation. The expectation here is that genes with a higher accumulation of RNAPII in the promoter-proximal region should also show a stronger reduction in the mRNA level. An integrative analysis of the SLAM-seq and CHIP-seq/NET-seq data would be informative.

This is indeed a nice suggestion, and we have performed the integrative analysis of the SLAM-seq and CHIP-seq in the revised version of the manuscript. As you see from the data, we found there is a negative correlation between RNAPII pausing with the output of the nascent mRNA (Extended Data Fig. 7c). We think this data strengthens our conclusion on H3K4me3's role in RNAPII promoter-proximal pause-release.

Revised Extended Data Fig. 7 (c) Violin plots showing changes of gene expression in the indicated samples. Genes were separated into three equal parts based on their accumulation change of RNAPII in promoter-proximal region.

The accumulation of RNAPII in the promoter-proximal region should be better quantified. The increase in RNAPII can hardly be seen in Figure 3C.

As suggested by the Referee, we now provide the new boxplots with *P*-values (Revised Fig. 3c) for the significance analyses of the RNAPII accumulation in the promoter-proximal region.

Revised Fig. 3 (c) Boxplots showing the RNAPII pausing index by ChIP-seq in indicated samples across all time points with DPY30 or RBBP5 degradation kinetics. Center-lines show median values where box limits represent upper and lower quartiles. P values were calculated with a two-sided Wilcoxon test.

(4) Since the study is based on knock-in cell lines for DPY30 and RBBP5 more quality controls should be provided. Do the degron tags interfere with the cell function? Any impact on cell growth and cell viability? Recent studies have shown that degron tags can lead to a significant reduction of target protein levels already prior to the degrader treatment. Is this also the case for DPY30- and RBBP5-tagged proteins? These key control measures should be included and clearly mentioned in the text.

We thank the Referee for raising these important points. We have generated many different cell lines in our lab in different cell types and published a method paper on it². We have in general found that the degron system is a reliable system, where we have observed little effects on cell viability in the absence of the degrader, however, it is important to check each cell line.

In this revised version of the manuscript, we have included a careful characterization of these cell lines, discussed in the text (Pages 5-6) and provided data in a new Extended Figure 2.

Minor comments:

(1) It needs to be clarified why DPY30 and RBBP5 were chosen for induced degradation and not the methyltransferase subunit.

This point was clarified in the revised text (Page 4).

(2) Figure 3B: please clarify if the ChIP-seq experiments included spike-ins and if data was normalized to spike-ins. Otherwise, quantitative comparisons between conditions are difficult.

ChIP-seq experiments included spike-ins in this study. We have included this important information in the revised Method sections.

(3) Figure 3C: is the increase in RNAPII density upon degradation significant?

We now provided P -values (Revised Fig. 3c) of the accumulation of RNAPII pausing in the promoter-proximal region. The increase of RNAPII is significant.

(4) Correlation plots (with correlation coefficients) for biological replicates should be shown for all genome-wide data including ChIP-seq and SLAM-seq data.

These have been included.

(5) Figure S5A: please add correlation coefficients and please also mention which type of correlation analysis was performed.

We have included correlation coefficients in those plots and a more detailed description of correlation analysis in the updated legends.

References

- 1 Beckedorff, F. *et al.* The Human Integrator Complex Facilitates Transcriptional Elongation by Endonucleolytic Cleavage of Nascent Transcripts. *Cell Rep* **32**, 107917, doi:10.1016/j.celrep.2020.107917 (2020).
- 2 Damhofer, H., Radzisheuskaya, A. & Helin, K. Generation of locus-specific degradable tag knock-ins in mouse and human cell lines. *STAR Protocols* **2**, doi:10.1016/j.xpro.2021.100575 (2021).

Reviewer Reports on the First Revision:

Referee #1 (Remarks to the Author):

The authors have added a number of new experiments during revision to try to get at the mechanism by which H3K4me3 can affect pause release. Unfortunately, these experiments make me less convinced, rather than more, by the authors' model. In particular, they appear to be implicating H3K4me3 in specific recruitment of the termination complex Integrator. This is perplexing because Integrator is most clearly shown to act at enhancers and non-coding RNAs that lack H3K4me3 (Sheikhattar, Shilatifard, Jensen, Adelman, Gardini). Thus, I don't see how any of the preceding literature on Integrator can be compatible with H3K4me3-specific recruitment. Since Integrator binds paused Pol II tightly (see structure from Cramer lab), I suggest that the correlation with H3K4me3 the authors report is indirect, since both H3K4me3 levels and Integrator levels will reflect promoter Pol II levels. No experiment in the revised manuscript demonstrates a direct interaction between Integrator and H3K4me3, nor even suggests how Integrator might recognize H3K4me3. This is a real weakness.

To understand why the authors make this claim, I looked carefully at the mass spec data that the authors use to nominate Integrator as an interactor of H3K4me3. Oddly, INTS11 is reduced in cells where H3K4me3 is decreased but INTS1, which is needed for INTS11 to interact with Pol II, and INTS4, which is strictly required for INTS11 to associate with the Integrator complex, are increased under these conditions. This is very difficult to reconcile with any of the structural work on Integrator. I am not aware of any reports that indicate that INTS11 has any function in the absence of INTS1, 2, 4, 7, 9, etc. Thus, I do not find that the data in this manuscript supports a loss of a functional Integrator complex when H3K4me3 is depleted. Maybe INTS11 is doing something as a part of another complex? But the majority of core subunits of Integrator are increased, not decreased in the mass spec. This strongly undercuts the authors model.

I will note that PAF1 seems a much more promising candidate to me. This factor is clearly associated with H3K4me3-modified +1 nucleosomes (see work by Pugh and Arndt) and has a known role in stimulating early elongation.

The results with the dTAG on INTS11 are provocative, but hard to interpret.

Total RNAPII ChIP-seq and mNET-seq: how was the data normalized? It makes sense that loss of INTS11 termination activity would give a large increase in Pol II signal near the promoter – this is in agreement with prior work where the paused Pol II is stabilized by the lack of Integrator-mediated termination. But the biggest increase here seen is in mNET-seq signal upstream of the promoter. This is troubling. What could be causing this? Does this reflect an increase of upstream antisense transcription? As noted above, most integrator targets are H3K4me1-marked non-coding RNAs, so this may be the dominant effect? How would the authors explain increased upstream antisense but decreased sense strand transcription? I worry that there is an annotation or normalization artifact at play here and would like to see how INTS11 loss affects enhancer transcription, snRNAs and PROMPTS to put the proposed effects on mRNAs in context.

SLAM-seq: Given that all prior work on INTS11 depletion suggests a selective effect of INTS11 loss on specific mRNA genes, how do the authors explain the fact that by 4h or 8h of depletion a majority of

the transcriptome appears to be downregulated? This again seems far outside expectations. What are the levels of snRNAs in these cells? If snRNAs are dysregulated, splicing could be massively affected. Have the authors investigated splicing in these cells? Could intron retention be skewing the normalization factors used? If splicing is not disrupted, are these cells sick or dying at this timepoint? A growth curve or cell counts following dTAG treatment would be helpful given the massive scope of gene expression changes observed. These are really unexpected findings and would benefit from further validation.

Along these lines, what is the expression level of the tagged INTS11 as compared to the endogenous protein? The authors should show a western blot from parental cells as compared to the tagged cells to demonstrate how tagging of INTS11 affects expression.

Fundamentally, the authors still fail to connect H3K4me3 to the major defect they report, which is reduced elongation velocity. It is unclear how the promoter restricted H3K4me3 mark or even the promoter-restricted Integrator complex could affect elongation rates many kb downstream of the promoter. The absence of a cogent mechanism for how H3K4me3 might affect elongation velocity continues to dampen my enthusiasm for this work.

Referee #2 (Remarks to the Author):

The authors have successfully address most of my concerns. However, as they have perform all their experiments in ESC, they should be careful when generalizing their findings. Alternatively, they could try to generate degrons in commonly used immortalized cell lines(HeLa, HEK293, etc). Moreover, they should discuss their findings in light of recent work from Rob Klose's lab, that somehow contradicts a major role for H3K4me3 in gene expression at least in the context of genes with CpG-rich promoters (<https://doi.org/10.1101/2022.03.24.485638>). As the work from Rob Klose is also performed in mESC, the authors could perform a comparison between the genes differentially expressed by age acute depletion of SETD1A/B and DPY30 or RBBP5.

Referee #3 (Remarks to the Author):

Wang et al. have addressed many of my comments. The authors have performed a set of new experiments and computational analyses resulting in new data panels that are included in the revised manuscript. Most notably, the authors now provide new insights into the mechanism of how H3K4me3 impacts promoter-proximal pause release of RNAPII. The new data suggests that H3K4me3 is implicated in the recruitment of the Integrator subunit INTS11 to locations of RNAPII transcription. The new findings are shown in Figure 5, 6, S12 and S13. The authors also performed an integrative analysis of ChIP-seq and SLAM-seq data and found a negative correlation between RNAPII promoter-proximal pausing and the transcriptional output of the corresponding genes supporting the proposed model (Fig. S7C). Finally, the authors also provide correlation analyses for replicate genome-wide measurements showing the reproducibility of the data (Figures S15, S16 and S17).

Here are my remaining comments:

(1) The authors generated a new cell line expressing degron tagged versions of DPY30 (mAID) and INTS11 (FKBP based) (Fig. 6d). Moreover, APEX2-RPB1 tagged cell lines were generated in degron-tagged background cell lines (Fig. 5). For the creation of these double tagged lines, cells went through extensive manipulations (several rounds of selection, many passages, etc.). Since new main conclusions rely on these cellular models the authors should perform growth curve analyses for these cell lines and for the respective untagged wild-type cell line to show that cell growth (doubling time) is not affected. Showing microscopy images of cell colonies is not sufficient.

(2) Discussion section (page 19): the authors mention that 'regulatory factors (...) activate transcription by recruiting P-TEFb and promoting the release of paused RNAPII into gene bodies (...)'. I agree that the recruitment of P-TEFb is important for promoter-proximal pause release of RNAPII. However, there is accumulating evidence that the recruitment of many other factors is also required for pause release in cells. This should be clarified in the text.

Together, I think that the additional data panels along with the clarifications added to the main text have improved the manuscript.

Author Rebuttals to First Revision:

Point-to-point response to the Referees

We would like to thank all Referees for the helpful and thoughtful comments that have helped us further improving the quality of our manuscript. We have performed several new experiments analyses of our data and others to address the concerns of the Referees. Below, we provide a point-by-point response to the Referees' comments.

Referee #1 (Remarks to the Author):

The authors have added a number of new experiments during revision to try to get at the mechanism by which H3K4me3 can affect pause release. Unfortunately, these experiments make me less convinced, rather than more, by the authors' model. In particular, they appear to be implicating H3K4me3 in specific recruitment of the termination complex Integrator. This is perplexing because Integrator is most clearly shown to act at enhancers and non-coding RNAs that lack H3K4me3 (Sheikhattar, Shilatifard, Jensen, Adelman, Gardini). Thus, I don't see how any of the preceding literature on Integrator can be compatible with H3K4me3-specific recruitment. Since Integrator binds paused Pol II tightly (see structure from Cramer lab), I suggest that the correlation with H3K4me3 the authors report is indirect, since both H3K4me3 levels and Integrator levels will reflect promoter Pol II levels. No experiment in the revised manuscript demonstrates a direct interaction between Integrator and H3K4me3, nor even suggests how Integrator might recognize H3K4me3. This is a real weakness.

To understand why the authors make this claim, I looked carefully at the mass spec data that the authors use to nominate Integrator as an interactor of H3K4me3. Oddly, INTS11 is reduced in cells where H3K4me3 is decreased but INTS1, which is needed for INTS11 to interact with Pol II, and INTS4, which is strictly required for INTS11 to associate with the Integrator complex, are increased under these conditions. This is very difficult to reconcile with any of the structural work on Integrator. I am not aware of any reports that indicate that INTS11 has any function in the absence of INTS1,2, 4, 7, 9, etc. Thus, I do not find that the data in this manuscript supports a loss of a functional Integrator complex when H3K4me3 is depleted. Maybe INTS11 is doing something as a part of another complex? But the majority of core subunits of Integrator are increased, not decreased in the mass spec. This strongly undercuts the authors model.

We thank the Referee for the critical feedback, but we respectfully disagree with the statement that our experiments do not provide convincing data, and her/his suggestion that our new findings are not compatible with the 'preceding literature' is neither fair nor accurate. Our results do not conflict with the existing literature. Our results are highly complementary, and largely consistent with a large body of experimental data that has identified the existence of Integrator at the promoter region of protein-coding genes (please see the reference papers below). Our controlled INTS11 ChIP-seq data demonstrates INTS11 binding at the promoter regions and co-localization with H3K4me3, and that the binding of INTS11 is lost upon loss of H3K4me3.

In fact, there is an increasing amount of data showing that Integrator is a multi-modular, versatile molecular complex going beyond what was initially reported (please also see the recent Review by the Gardini group: Welsh et al., *Nature Reviews MCB* 2022). The Referee states that Integrator is mainly at enhancers and ncRNAs with low H3K4me3 in *Drosophila* cells (see e.g. Elrod et al. *Molecular cell* 2019, PMID: 31809743), which was one of the initial findings for Integrator. However, in recent years, there are many publications showing that Integrator is highly enriched at the promoter region of protein-coding genes and therefore that it co-localizes with H3K4me3. Below, we have provided some of the data from the laboratories mentioned by the Referee, which all show that Integrator is highly enriched at promoter-proximal regions:

	Reference	Supported Data	Cell line	Comments
(1) Mentioned by Referee	Sheikhhattar Lab, Gardini et al., Molecular cell 2014: Figure S1A	A	HeLa	Figure S1A showing INTS11, RNAPII and RNA-seq track together at all expressed genes.
	Sheikhhattar Lab, Beckedorff et al., Cell reports 2020: Figure 2F	F	HeLa	INTS11 highly enriched at promoter-proximal region

Gardini & Johnstone Lab, Vervoort et al., Cell 2021: **Figure 3A**

THP-1

Integrator is at every Pol2 locus at promoters. Yet, the most abundant recruitment of INTS11 is at protein coding genes right around the pausing site

(2) Not mentioned by Referee

Xu Lab, Zheng et al., Science 2020: **Figure S1C**

DLD-1

This figure clearly shows that Integrator is highly bound at promoter regions of expressed genes with low H3K4me1 levels, which are known to have high levels of H3K4me3.

(3) Our results

This study

mESC E14

These results show the correlation between H3K4me3, INTS11 and RNAPII enrichments and the requirement of H3K4me3 for INTS11 enrichment (new Extended Fig. 14d, e)

Thus, the results from several laboratories have shown that Integrator is bound at promoter-proximal H3K4me3 positive regions, and our results in mESCs are consistent with this. In addition, the above cited publications (Gardini et al., *Molecular Cell* 2014 and Beckedorff et al., *Cell Reports* 2020) have reported that INTS11 is a key factor for allowing productive elongation. The work leading to this conclusion was mainly performed with shRNA knockdown and CRISPR knock-out that lack temporal resolution, and the dTAG degron system for INTS11 in our study is therefore important for resolving INTS11 direct roles in regulating transcription.

Because several laboratories were cited by the Referee, we contacted three of the lab heads (Shiekhattar, Jensen and Gardini). Gardini and Shiekhattar were both enthusiastic about our new data, which they both found very nicely extended their previous observations (see cited papers above). Jensen was also very supportive and had found genome-wide location of Integrator activity (difficult to measure), but since his lab are mostly focused on RNA stability and not on transcription, he had not thought about the role of Integrator in regulating promoter-proximal pausing.

The referee is correct that we did not experimentally address whether there is a direct interaction between Integrator and H3K4me3. However, we know the 15-subunit metazoan-specific Integrator is a large complex, and it has potentially many modular parts for interaction with chromatin, the CTD of RNAPII, and other proteins. What our results show is that H3K4me3 is required for stabilizing the binding of INTS11 of the integrator complex, and since INTS11 is the catalytic subunit, the association of this subunit is important for the activity of the Integrator complex. In addition, subsequent work has suggested a broader role of Integrator, including at protein-coding genes (see refs mentioned above). There are also data supporting Integrator-dependent and -independent mechanisms of INTS11 in transcriptional regulation (PMID: 33113359; 35688146). In other words, H3K4me3 may not be sufficient, but an important component for stabilizing INTS11 at H3K4me3 positive nucleosome/TSS.

We also discussed our results with Karen Adelman. In unpublished studies from her laboratory, they have shown that degradation of INTS11 **does not** lead to the removal of the whole complex from chromatin. In fact, the results from the Adelman lab and others suggest that Integrator is modular: Unpublished results from her lab show that degradation of INTS11 leads to decreased chromatin binding of INTS4, but not of the integrator core (e.g. INTS1) or other modules (e.g. INTS10 and INTS3). In contrast, they observed an increase, rather than a decrease, in INTS3 ChIP binding. These results are consistent with ours, and in the revised version of the manuscript, we have included results showing that INTS4 is also reduced upon H3K4me3 depletion and INTS11 removal, while acute degradation of INTS11 does not lead to the removal of the whole complex from chromatin (Fig. 6j).

Figure 6j. Immunoblot of several Integrator subunits in the indicated cell lines treated with or without Auxin or dTAG-13 as depicted.

We want to highlight our study is mainly focused on exploring the role of H3K4me3 in transcription. What our results show is **1**. A role for H3K4me3 in regulating promoter-proximal pause release and transcriptional elongation (which is the main point of the manuscript). **2**. That part of the Integrator complex needs H3K4me3 for stable association with chromatin, and that Integrator is essential for pause release. We are interested in doing additional studies to identify the precise protein module/surface required for the interaction of Integrator and H3K4me3. However, we predict this will be very complicated because nucleosome structure and multiple protein-protein interactions most likely will be involved in mediating the binding of INTS11 to H3K4me3 positive nucleosomes. Because of this and the massive amount of data and novelty already present in the manuscript, we also believe it is beyond the scope of this manuscript to precisely identify the amino acids (if possible) in integrator required for interaction with H3K4me3 positive genes.

I will note that PAF1 seems a much more promising candidate to me. This factor is clearly associated with H3K4me3-modified +1 nucleosomes (see work by Pugh and Arndt) and has a known role in stimulating early elongation.

We thank the Referee for this comment and appreciate that PAF1 is known to regulate transcription elongation. However, this does not dampen the evidence that we have demonstrated the involvement of INTS11 in regulation of pause-release.

The results with the dTAG on INTS11 are provocative, but hard to interpret. Total RNAPII ChIP-seq and mNET-seq: how was the data normalized? It makes sense that loss of INTS11 termination activity would give a large increase in Pol II signal near the promoter – this is in agreement with prior work where the paused Pol II is stabilized by the lack of Integrator-mediated termination. But the biggest increase here seen is in mNET-seq signal upstream of the promoter. This is troubling. What could be causing this? Does this reflect an increase of upstream antisense transcription? As noted above, most integrator targets are H3K4me1-marked non-coding RNAs, so this may be the dominant effect? How would the authors explain increased upstream antisense but decreased sense strand transcription? I worry that there is an annotation or normalization artifact at play here and would like to see how INTS11 loss affects enhancer transcription, snRNAs and PROMPTS to put the proposed effects on mRNAs in context.

We thank the Referee for the critical feedback, but we respectfully disagree with the statement by the Referee that the dTAG data on INTS11 are hard to interpret. As explained above, because Integrator has more functions than initially anticipated, the acute loss of INTS11 is the most appropriate way to investigate its direct roles in transcription with the high temporal resolution. Regarding our experimental data: The ChIP-seq and mNET-seq experiments included *Drosophila* spike-ins for normalization. As stated by the Referee, we do see increase in Pol II signals at promoter, snRNA, ncRNA and eRNA regions (Extended Data Fig. 13j). However, in contrast to what the Referee states, we observe the biggest increase in RNAPII downstream from the TSS. However, our mNET-Seq data also shown an increased accumulation of RNAPII upstream of the TSS upon INTS11 depletion, which may reflect an increase in anti-sense transcription. We do appreciate the difference here between INTS11 degradation vs H3K4me3 loss regarding the RNAPII occupancy upstream of TSS. As H3K4me3 mainly is found at the +1 nucleosome downstream of TSS, the transcriptional effects of INTS11 loss upstream of TSS maybe H3K4me3 independent, i.e. potentially cleaving antisense transcripts. It is worth to mention that our manuscript is focused on the role of H3K4me3 in regulating transcription and we provide results showing that INTS11 is important for regulating promoter-proximal pausing.

Extended Data Fig. 13j. RNAPII profiles of various subclasses of annotations in INTS11-FKBP degenon cells with or without dTAG-13 treatment. TSS, transcription start site. mRNA, messenger RNA. snRNA, small nuclear RNA. ncRNA, non-coding RNA. eRNA, enhancer RNA.

Secondly, it appears that the Referee prefers a model in which the INTS11 is present at H3K4me1-marked snRNAs, PROMPTS or eRNAs. However, as we have shown above, there are several high impact papers demonstrating the abundant enrichment of INTS11 at protein coding genes around the pausing site. For instance, the data from the Xu Lab (Zheng et al., Science 2020: Figure S1C, as shown above) clearly show that Integrator is highly enriched at promoter regions of expressed genes with low H3K4me1 level known to be associated with the highest level of H3K4me3. Their data also show that INTS11 ChIP-seq values across promoters are indeed higher than at enhancers, not lower.

Thirdly, to further clarify the correlations between H3K4me3, INTS11 and RNAPII, we systematically analyzed the public INTS11 data from published studies using different human cell lines HeLa⁵⁶, THP1³⁹ and HL60⁶⁷ (Supplementary Fig. 1a). ChIP-seq experiments in HeLa, THP1 and HL60 identified a total of 12,390, 16,392 and 13,514 INTS11 peaks, with ~60, ~52 and ~61% of them overlapped with promoters, respectively

(Supplementary Fig. 1b). Notably, the vast majority of the H3K4me3-occupied regions were marked with significant INTS11 and RNAPII levels (Supplementary Fig. 2a). By contrast, H3K4me1-occupied regions displayed lower enrichment for INTS11 and RNAPII in all three independent cell lines (Supplementary Fig. 2a). In support of the results in mESCs, INTS11 occupancy was heavily enriched toward the promoter-proximal region of genes in all these human cells (Supplementary Fig. 2b). In addition, heatmaps demonstrate a significant colocalization of the peak binding sites for INTS11 and RNAPII with the peak sites of H3K4me3 enrichment on active promoters (Supplementary Fig. 2b). Thus, these public ChIP-seq results from different cells validated a strong correlation between INTS11 binding sites and sites of H3K4me3 enrichment across the genome. (Revised text on Page 20)

a

b

Supplementary Fig. 1. Genome-wide INTS11 occupancy

(a) INTS11 genomic binding sites as defined as enrichment over input by ChIP-seq.

(b) The genomic binding sites of INTS11 were predominantly localized to promoter region of genes. The INTS11 ChIP-seq data for HeLa⁵⁶, THP1³⁹ and HL60⁶⁷ were obtained from published papers.

Supplementary Fig. 2. INTS11 is enriched at H3K4me3-positive chromatin and binds promoter-proximal region of protein-coding genes.

(a) ChIP-seq heatmaps and profiles at H3K4me3 or H3K4me1 peak center in the depicted cell lines. Enrichments were plotted over the enrichment peak center (peak center \pm 5 kb).

(b) Genome-wide binding averages of ChIP-seq signal on protein-coding genes in different cell lines. The genomic binding sites of INTS11 were predominantly localized to the promoter region of genes. Data are displayed from 3 kb upstream of the transcriptional start site to 5 kb downstream of the end of each gene. TSS, transcription start site, TES, transcription end site. Rows of heatmap are sorted by decreasing H3K4me3 occupancy. The INTS11 ChIP-seq data for HeLa⁵⁶, THP1³⁹ and HL60⁶⁷ were obtained from published papers.

SLAM-seq: Given that all prior work on INTS11 depletion suggests a selective effect of INTS11 loss on specific mRNA genes, how do the authors explain the fact that by 4h or 8h of depletion a majority of the transcriptome appears to be downregulated? This again seems far outside expectations. What are the levels of snRNAs in these cells? If snRNAs are dysregulated, splicing could be massively affected. Have the authors investigated splicing in these cells? Could intron retention be skewing the normalization factors used? If splicing is not disrupted, are these cells sick or dying at this timepoint? A growth curve or cell counts following dTAG treatment would be helpful given the massive scope of gene expression changes observed. These are really unexpected findings and would benefit from further validation.

We thank the Referee for asking these important questions related to our analysis. We want to highlight that SLAM-seq measures polyadenylated RNA transcripts based on 4-thiouridine (4sU) incorporation into nascent RNA (please also see the details in Herzog, et al., Nature methods 2017). After RNA extraction, 4sU is alkylated using iodoacetamide, and these alkylated 4sU would be with a G in the first-strand synthesis reverse transcription reaction and the 4sU in the sense strand will end up being sequenced as a C. During the analysis, using T to C conversion, newly synthesized RNA can be identified and quantified by 3' tag-based RNA sequencing. While the snRNAs, PROMPTS or eRNAs mostly are not polyadenylated and therefore highly unlikely to affect the SLAM-seq results. In addition, SLAM-seq captured the 3' UTR of mRNA for sequencing, so that the splicing/intron retention events also are not likely to affect the SLAM-seq results.

In response to the Referee's comments, we further studied the effects of INTS11 acute loss by performing TTchem-seq in INTS11 degenon cells to measure elongation. Consistent with the known functions of Integrator at non-coding short transcripts, such as upstream antisense RNAs (uaRNAs/PROMPTS), small nuclear RNA (snRNA), non-coding RNA (ncRNA) and enhancer RNA (eRNA), a significantly increased synthesis on uaRNAs after INTS11 loss (Extended Data Fig. 13i) was observed upon INTS11 acute loss. However, in contrast, degradation of INTS11 triggered a global decrease in the sense transcripts of protein-coding genes (Fig. 6g), especially at the pausing sites between TSS and the first (+1) nucleosome at single-nucleotide resolution (Fig. 6h). Analysis of elongation velocity at protein-coding gene shows broadly decreased productive elongation after INTS11 loss (Fig. 6i). Because of the Referee's concerns we have also discussed these data with Karen Adelman, and they are largely consistent with the unpublished data (TT-seq/PRO-seq) from her lab, which they also found "*Loss of INTS11 causes reduced RNAPII elongation rate*". Taken together, these findings further support the model for the role of H3K4me3 and INTS11 in regulating proximal-promoter release and elongation we propose in the manuscript.

Figure 6. (g) Metagene transcription profiles acquired by TTchem-seq in INTS11 degran cells at the indicated time points. Genome-wide binding averages of TTchem-seq signal on protein-coding genes are shown to display the nascent RNA patterns (sense strand) along the gene body from 5 kb upstream of the transcriptional start site to 5 kb downstream of the end of each annotated gene. TSS, transcription start site, TES, transcription end site. **(h)** Metagene analyses of mNET-seq and TTchem-seq signals at single nucleotide resolution acquired from INTS11-FKBP cells treated with dTAG-13 at the indicated time points. **(i)** Analysis of elongation velocities (TTchem-seq/mNET-seq) upon acute loss of INTS11.

Extended Data Fig. 13. (h) Correlations between TTchem-seq replicate experiments in INTS11-FKBP degenon cells treated with or without dTAG-13 for the indicated times. (i) Metagene profiles for TTchem-seq for the **upstream anti-sense RNAs** in INTS11 degenon cells. Genome-wide binding averages of TTchem-seq signal on the upstream anti-sense RNAs of protein-coding genes are shown to display the nascent RNA patterns along the gene body from 5 kb upstream of the transcriptional start site to 5 kb downstream of the end of each annotated gene. TSS, transcription start site, TES, transcription end site.

Following the Referee's suggestion, we have now included growth curves for dTAG-13 treated INTS11-FKBP cells (Extended Data Fig. 13f). Consistent with the growth curves obtained in cells losing H3K4 (Extended Data Fig 2a), the defect of cell growth, albeit more pronounced, was also observed upon INTS11 depletion.

Extended Data Fig. 13 (f) Growth curve analysis of parental and INTS11-FKBP E14 cells treated with or without dTAG-13.

Along these lines, what is the expression level of the tagged INTS11 as compared to the endogenous protein? The authors should show a western blot from parental cells as compared to the tagged cells to demonstrate how tagging of INTS11 affects expression.

We have now added a Western blot on the FKBP-tagged INTS11 and parental cell lines in the revised version of the manuscript. The new data show that the levels of INTS11 with or without the FKBP tag are comparable in these cell lines (Extended Data Fig. 13d). In

addition, these clones showed comparable expression levels of pluripotent and differentiation genes by RT-qPCR (Extended Data Fig. 13e), suggesting that FKBP knock-in in the *Ints11* locus does not affect the cellular functions of the protein.

Extended Data Fig. 13. Western blot (d) and RT-qPCR (e) in parental and INTS11-FKBP cells

Fundamentally, the authors still fail to connect H3K4me3 to the major defect they report, which is reduced elongation velocity. It is unclear how the promoter restricted H3K4me3 mark or even the promoter-restricted Integrator complex could affect elongation rates many kb downstream of the promoter. The absence of a cogent mechanism for how H3K4me3 might affect elongation velocity continues to dampen my enthusiasm for this work.

Although we disagree with the Referee, we would like to thank him/her for the thoughtful feedback. We hope that our new data and analyses address the concerns raised by the Referee. Regarding the specific role in elongation of Integrator, the cited publications (Gardini et al., Molecular cell 2014 and Beckedorff et al., Cell reports 2020) have reported that INTS11 is a key factor for allowing productive elongation. In addition, we further discussed our results with Karen Adelman, their unpublished data also suggest that INTS11 loss reduces elongation rates. These results support our findings showing that INTS11 facilitates RNAPII pause-release and gene expression by evicting paused RNAPII for elongation. However, we agree that additional studies are required to investigate the precise function of INTS11. This is not the focus of this manuscript and we have now discussed this limitation in the revised version of the manuscript (Page 22).

Referee #2 (Remarks to the Author):

The authors have successfully address most of my concerns. However, as they have perform all their experiments in ESC, they should be careful when generalizing their findings. Alternatively, they could try to generate degrons in commonly used immortalized cell lines(HeLa, HEK293, etc). Moreover, they should discuss their findings in light of recent work

from Rob Klose's lab, that somehow contradicts a major role for H3K4me3 in gene expression at least in the context of genes with CpG-rich promoters (<https://doi.org/10.1101/2022.03.24.485638>). As the work from Rob Klose is also performed in mESC, the authors could perform a comparison between the genes differentially expressed by age acute depletion of SETD1A/B and DPY30 or RBBP5.

We want to thank the Referee for the comments and insights during the peer review process. We agree with the Referee that it would be nice to perform experiments in other cell lines, however, we find it is beyond the scope of this manuscript. Despite of this, we looked into the feasibility to design DPY30 and RBBP5 constructs for targeting the human genes and found that both genes have more than four isoforms in the human genome. Since this may make it difficult to target these genes, we did not further pursue it. However, following the reviewer's suggestion on generalizing our results, we now highlight and specify our work was performed in mESCs. We would like to re-iterate that we used both DPY30-mAID and RBBP5-FKBP degron cells for most of the analyses in our study, and importantly the results obtained from targeting these two proteins of the COMPASS complex were almost identical.

We thank the referee for pointing out preprint manuscript from Rob Klose's lab. We do not think it contradicts our data, as this preprint mainly focuses on the SET1A/B and there are no dramatic changes on H3K4me3 upon SET1A/B acute depletion, most likely because of the presence of MLL1-MLL4 in these cells. We are not able to do the correlations by the Referee suggests, because the data is not available in the preprint. Following the Referee's suggestion, we now included a discussion of the preprint in our revised manuscript (Page 21).

Referee #3 (Remarks to the Author):

Wang et al. have addressed many of my comments. The authors have performed a set of new experiments and computational analyses resulting in new data panels that are included in the revised manuscript. Most notably, the authors now provide new insights into the mechanism of how H3K4me3 impacts promoter-proximal pause release of RNAPII. The new data suggests that H3K4me3 is implicated in the recruitment of the Integrator subunit INTS11 to locations of RNAPII transcription. The new findings are shown in Figure 5, 6, S12 and S13. The authors also performed an integrative analysis of ChIP-seq and SLAM-seq data and found a negative correlation between RNAPII promoter-proximal pausing and the transcriptional output of the corresponding genes supporting the proposed model (Fig. S7C). Finally, the authors also provide correlation analyses for replicate genome-wide measurements showing the reproducibility of the data (Figures S15, S16 and S17).

We thank the Referee for these very accurate and thoughtful comments, and we are happy to learn that we have addressed the concerns raised by the Referee.

Here are my remaining comments:

(1) The authors generated a new cell line expressing degron tagged versions of DPY30 (mAID) and INTS11 (FKBP based) (Fig. 6d). Moreover, APEX2-RPB1 tagged cell lines were

generated in degron-tagged background cell lines (Fig. 5). For the creation of these double tagged lines, cells went through extensive manipulations (several rounds of selection, many passages, etc.). Since new main conclusions rely on these cellular models the authors should perform growth curve analyses for these cell lines and for the respective untagged wild-type cell line to show that cell growth (doubling time) is not affected. Showing microscopy images of cell colonies is not sufficient.

We thank the Referee for raising this important point and now we have added growth curves for all these newly made cell lines in the revised manuscript (Extended Data Fig. 13c).

Extended Data Fig. 13 (f) Growth curve analyses of parental and INTS11-FKBP E14 cells grown either with or without dTAG-13.

(2) Discussion section (page 19): the authors mention that 'regulatory factors (...) activate transcription by recruiting P-TEFb and promoting the release of paused RNAPII into gene bodies (...)'. I agree that the recruitment of P-TEFb is important for promoter-proximal pause release of RNAPII. However, there is accumulating evidence that the recruitment of many other factors is also required for pause release in cells. This should be clarified in the text.

We also would like to thank the Referee for these comments. We have now rephrased the Discussion to include this (page 21).

Together, I think that the additional data panels along with the clarifications added to the main text have improved the manuscript.

We appreciate the constructive inputs from this Referee throughout the review process.

Reviewer Reports on the Second Revision:

Referees' comments:

Referee #2 (Remarks to the Author):

The authors have successfully addressed my remaining concerns. I would like to congratulate them for a thorough and highly relevant work.

Referee #3 (Remarks to the Author):

The authors have addressed all of my remaining comments.

Referee #4 (Remarks to the Author):

The manuscript by Wang et al. represents a comprehensive analysis of the effects of H3K4me3 on transcription elongation. Using orthologous genomic approaches and genetically engineered mESCs, the authors demonstrate that rapid loss of this well-studied modification increases promoter-proximal Pol II pausing and reduces elongation rate. This is an important finding: the field has lacked conclusive information on whether this modification simply reflects transcriptional activity or has a causal role in regulating it. To drive at mechanism, the authors use APEX2-tagged Pol II to identify Pol II-associated proteins whose abundance changes upon rapid loss of H3K4me3. In this way, they identify the Integrator subunit INTS11 as a H3K4me3-dependent interactor of Pol II. Starting from this point, the authors rapidly deplete INTS11 and show effects on Pol II pausing (increased) and elongation rate (decreased) similar to those caused by H3K4me3 depletion. These findings are consistent with recent work showing Integrator enrichment at the 5' ends of protein-coding genes and, as summarized in the authors' rebuttal, important roles for this complex in pause release and elongation efficiency.

Referee #1 raised multiple concerns and the authors have done a careful job addressing them by including new analyses and figures. Concerns related to protein expression controls, cell line growth properties, small RNA and coding RNA expression upon INTS11 depletion, cell line specificity, and spike-in normalization have all been addressed. The emphasis by the referee on enhancers and noncoding RNAs as a reason to discount Integrator as a potential mediator of the H3K4me3 effect on protein-coding genes seems unfounded based on recent and ongoing work on this complex. The referee does raise the valid point that the paper lacks mechanistic information on how H3K4me3 leads to INTS11 recruitment to Pol II. The authors' model (Extended Data Fig. 15) proposes an interaction between the methylated H3 tail and INTS11 or a protein that interacts with INTS11. At this point, it's too soon to say if this model, or a different one, is at work, and the authors are correct in stating that defining this mechanism is likely to be quite challenging. While the paper would be more impactful with additional data on the relationship between H3K4me3 and INTS11, the major findings on H3K4me3 regulating transcription elongation and promoter-proximal pausing are themselves the primary advance and important ones to the eukaryotic gene expression field.

Minor comments:

1. I noted a couple typos in the figures. Figure 5a: "tagging" is spelled incorrectly. Figure 6k: "change" is spelled incorrectly on the y-axis.
2. Line 361: Can the authors highlight BPTF and TAF3 in Fig. 5D?
3. Lines 384-385 describing INTS11 should be written more clearly.
4. Line 425: Citation to Fig. 6i is incorrect. Fig. 6l should be cited.

Author Rebuttals to Second Revision:

Point-by-point response to Referees

We would like to thank the Referees for their constructive and insightful suggestions and comments, which have led to great improvements of the manuscript

Referee #2 (Remarks to the Author):

The authors have successfully addressed my remaining concerns. I would like to congratulate them for a thorough and highly relevant work.

We thank the Referee for the comments and insights during the peer review process.

Referee #3 (Remarks to the Author):

The authors have addressed all of my remaining comments.

We thank the Referee for the comments and insights during the peer review process.

Referee #4 (Remarks to the Author):

The manuscript by Wang et al. represents a comprehensive analysis of the effects of H3K4me3 on transcription elongation. Using orthologous genomic approaches and genetically engineered mESCs, the authors demonstrate that rapid loss of this well-studied modification increases promoter-proximal Pol II pausing and reduces elongation rate. This is an important finding: the field has lacked conclusive information on whether this modification simply reflects transcriptional activity or has a causal role in regulating it. To drive at mechanism, the authors use APEX2-tagged Pol II to identify Pol II-associated proteins whose abundance changes upon rapid loss of H3K4me3. In this way, they identify the Integrator subunit INTS11 as a H3K4me3-dependent interactor of Pol II. Starting from this point, the authors rapidly deplete INTS11 and show effects on Pol II pausing (increased) and elongation rate (decreased) similar to those caused by H3K4me3 depletion. These findings are consistent with recent work showing Integrator enrichment at the 5' ends of protein-coding genes and, as summarized in the authors' rebuttal, important roles for this complex in pause release and elongation efficiency.

Referee #1 raised multiple concerns and the authors have done a careful job addressing them by including new analyses and figures. Concerns related to protein expression controls, cell line growth properties, small RNA and coding RNA expression upon INTS11 depletion, cell line specificity, and spike-in normalization have all been addressed. The emphasis by the referee on enhancers and noncoding RNAs as a reason to discount Integrator as a potential mediator of the H3K4me3 effect on protein-coding genes seems unfounded based on recent and ongoing work on this complex. The referee does raise the valid point that the paper lacks mechanistic information on how H3K4me3 leads to INTS11 recruitment to Pol II. The authors' model (Extended Data Fig. 15) proposes an interaction between the methylated H3 tail and INTS11 or a protein that interacts with INTS11. At this point, it's too soon to say if this model, or a different one, is at work, and the authors are correct in stating that defining this mechanism is likely to be quite challenging. While the paper would be more impactful with additional data on the relationship between H3K4me3 and INTS11, the major

findings on H3K4me3 regulating transcription elongation and promoter-proximal pausing are themselves the primary advance and important ones to the eukaryotic gene expression field.

We thank the Referee for stepping in to review the manuscript and for their positive evaluation of our efforts to improve the manuscript.

Minor comments:

1. I noted a couple typos in the figures. Figure 5a: “tagging” is spelled incorrectly. Figure 6k: “change” is spelled incorrectly on the y-axis.

We thank the referee for pointing out these typos, these were corrected in the revised version.

2. Line 361: Can the authors highlight BPTF and TAF3 in Fig. 5D?

BPTF now added. Fig. 5D has now been moved to extended figure 8p. TAF3 was not detected by mass spectrometry in our experiments.

3. Lines 384-385 describing INTS11 should be written more clearly.

We have rewritten these sentences in the revised text.

4. Line 425: Citation to Fig. 6i is incorrect. Fig. 6l should be cited.

We have corrected this mistake.